# Symmetry-enforced topological nodal planes at the Fermi surface of a chiral magnet

Marc A. Wilde[1,2 ✉], Matthias Dodenhöft[1], Arthur Niedermayr[1], Andreas Bauer[1,2], Moritz M. Hirschmann[3], Kirill Alpin[3], Andreas P. Schnyder[3 ✉] & Christian Pfleiderer[1,2,4 ✉]

Despite recent efforts to advance spintronics devices and quantum information technology using materials with non-trivial topological properties, three key challenges are still unresolved[1–9]. First, the identification of topological band degeneracies that are generically rather than accidentally located at the Fermi level. Second, the ability to easily control such topological degeneracies. And third, the identification of generic topological degeneracies in large, multisheeted Fermi surfaces. By combining de Haas–van Alphen spectroscopy with density functional theory and band-topology calculations, here we show that the non-symmorphic symmetries[10–17] in chiral, ferromagnetic manganese silicide (MnSi) generate nodal planes (NPs)[11,12], which enforce topological protectorates (TPs) with substantial Berry curvatures at the intersection of the NPs with the Fermi surface (FS) regardless of the complexity of the FS. We predict that these TPs will be accompanied by sizeable Fermi arcs subject to the direction of the magnetization. Deriving the symmetry conditions underlying topological NPs, we show that the 1,651 magnetic space groups comprise 7 grey groups and 26 black-and-white groups with topological NPs, including the space group of ferromagnetic MnSi. Thus, the identification of symmetry-enforced TPs, which can be controlled with a magnetic field, on the FS of MnSi suggests the existence of similar properties—amenable for technological exploitation—in a large number of materials.

Nearly a century ago Wigner, von Neumann and Herring[1,2] addressed the conditions under which Bloch states form degenerate band crossings, but their topological character and technological relevance has been recognized only recently[3–5]. To be useful[4–9], tiny changes of a control parameter must generate a large response, underscoring the lack of control over the band filling as the unresolved key challenge in materials with band crossings known so far. This raises the question whether topological band crossings exist that are (1) generically located at the Fermi level, (2) separated sufficiently in the Brillouin zone (BZ) and (3) easy to control.

Natural candidates are systems with non-symmorphic symmetries—for example, screw rotations—that generate positions in reciprocal space at which band-crossings are symmetry-enforced. The associated key characteristics include[10–17]: (1) the crossings are due to symmetry alone, that is, they occur on all bands independent of details such as chemical composition; (2) pairs of band crossings with opposite chirality are separated in $k$-space by about half a reciprocal lattice vector; (3) the band crossings may be enforced on entire planes[11,12], forming so-called nodal planes (NPs) with non-zero topological charge; and (4) their existence may be controlled by means of symmetry breaking. Thus, if in a material the Fermi surfaces (FSs) cross such topological NPs, they enforce pairwise FS degeneracies with large Berry curvatures. The topology of these FS degeneracies, which we refer to as topological protectorates (TPs), will be independent of material-specific details

and, moreover, may be controlled by symmetry breaking. The putative existence of topological NPs has been studied in phononic metamaterials[18–20], and mentioned in a study of non-magnetic chiral systems focusing on Kramers–Weyl fermions[21].

To demonstrate the formation of symmetry-enforced TPs at the intersection of NPs with the FS, we decided to study the ferromagnetic state of manganese silicide (MnSi), which has attracted great interest for its itinerant-electron magnetism[22], helimagnetism, skyrmion lattice[23] and quantum phase transition[24]. Crystallizing in space group (SG) 198, MnSi is a magnetic sibling of non-magnetic RhSi (ref. [25]), CoSi (ref. [26]) and PdGa (ref. [27]), in which sizeable Fermi arcs and multifold fermions were recently inferred from angle-resolved photoemission spectroscopy. MnSi is ideally suited for our study, as magnetic fields exceeding around 0.7 T stabilize ferromagnetism with magnetic screw-rotation symmetries enforcing NPs.

## Initial assessment

A first theoretical assessment establishes that a ferromagnetic spin polarization along a high-symmetry direction, for example, [010], reduces the symmetries from SG 198 ($P2_13$) of paramagnetic MnSi to the magnetic SG 19.27 ($P2_12_1'2_1'$) (Supplementary Note 1, Extended Data Fig. 1). This SG contains two magnetic screw rotations $\theta\bar{C}_2^x$ and $\theta\bar{C}_2^z$ (Fig. 1a), that is, 180° screw rotations around the $x$ and $z$ axes combined

[1]Physik Department, Technische Universität München, Garching, Germany. [2]Centre for QuantumEngineering (ZQE), Technische Universität München, Garching, Germany. [3]Max-Planck-Institute for Solid State Research, Stuttgart, Germany. [4]MCQST, Technische Universität München, Garching, Germany. ✉e-mail: marc.wilde@ph.tum.de; a.schnyder@fkf.mpg.de; christian.pfleiderer@tum.de

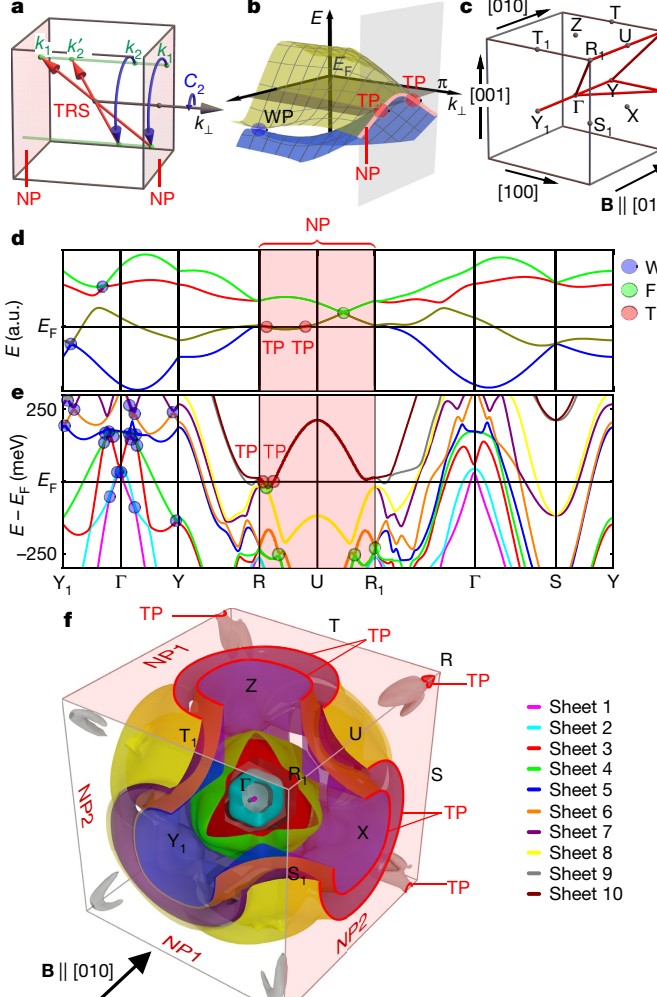

**Fig. 1 | Symmetries, band topology, Fermi surface protectorates and band structure of ferromagnetic MnSi. a**, Action of the magnetic screw rotations and time-reversal symmetry (TRS) on the $k$-points in the BZ. **b**, Pairs of energy bands $E(k)$ close to the Fermi energy $E_F$ forming a topological NP (red line) on the BZ boundary that is perpendicular to the screw-rotation axis. This NP is the topological partner of a single Weyl point (WP) in the bulk (blue dot) of opposite topological charge. **c**, High-symmetry paths in the cubic primitive BZ. Special $k$-points are denoted by the orthorhombic primitive notation with subscripts for easier identification. **d**, Generic tight-binding band structure illustrating the generic band degeneracies of ferromagnetic MnSi with its magnetic space group, SG 19.27, namely Weyl points, four-fold degenerate points (FPs), NPs and TPs. **e**, Band structure of ferromagnetic MnSi for magnetization along [010] as calculated using DFT. Ten bands cross the Fermi level, as distinguished by different colours corresponding to the FS sheets numbered in **f**. **f**, Calculated FS sheets adapted to match the experimental data under magnetic field along [010], as discussed in Methods. Note the presence of NPs on the BZ boundaries, $k_x = \pm\pi$ and $k_z = \pm\pi$, as well as TPs marked in red. a.u., arbitrary units.

with time-reversal symmetry $\theta$. These rotations act like mirror symmetries, as they relate Bloch wave functions at $(k_x, k_y, k_z)$ to those at $(-k_x, k_y, k_z)$ and $(k_x, k_y, -k_z)$, respectively, leaving the planes $k_x = 0$ and $k_z = 0$ and the BZ boundaries $k_x = \pm\pi$ and $k_z = \pm\pi$ invariant. Squaring $\theta\bar{C}_2^x$ and $\theta\bar{C}_2^z$ and letting them operate on the Bloch state $|\psi(\mathbf{k})\rangle$, one finds that $(\theta\bar{C}_2^x)^2|\psi(\mathbf{k})\rangle = e^{ik_x}|\psi(\mathbf{k})\rangle$ and $(\theta\bar{C}_2^z)^2|\psi(\mathbf{k})\rangle = e^{ik_z}|\psi(\mathbf{k})\rangle$. Hence, by Kramers theorem[28], all Bloch states on planes with $k_x = \pm\pi$ or $k_z = \pm\pi$ are two-fold degenerate. Moving away from these BZ boundaries, the symmetries are lowered such that the Bloch states become non-degenerate. Therefore, all bands in ferromagnetic MnSi are forced to cross at $k_x = \pm\pi$ and $k_z = \pm\pi$, representing a duo of NPs.

The topological charge $\nu$ of this duo of NPs (Fig. 1b) may be determined with the fermion doubling theorem[29], which states that $\nu$ summed over all band crossings must be zero. We note that besides the NPs, there is an odd number of symmetry-enforced band crossings on the $Y_1$–$\Gamma$–Y and $R_1$–U–R lines forming Weyl points ($\nu = \pm1$) and four-fold points ($\nu = \pm2$), respectively (Fig. 1c, d, Extended Data Fig. 2, Supplementary Note 1). Moreover, due to the effective mirror symmetries, accidental Weyl points away from these high-symmetry lines must form pairs or quadruplets with the same $\nu$. As the sum over $\nu$ of all of these Weyl and four-fold points is odd, the duo of NPs must carry a non-zero topological charge to satisfy the fermion doubling theorem. Hence, the duo of NPs at the BZ boundary is the topological partner of a single Weyl point on the $Y_1$–$\Gamma$–Y line (Fig. 1b). This is a counter-example to Weyl semimetals, in which Weyl points occur always in pairs.

Shown in Fig. 1d is the band structure of a generic tight-binding model satisfying SG 19.27 (Supplementary Note 2), where pairs of bands form NPs on the BZ boundaries $k_x = \pm\pi$ and $k_z = \pm\pi$, whereas on the $Y_1$–$\Gamma$–Y and $R_1$–U–R lines there are Weyl and four-fold points, respectively. Explicit calculation of the Chern numbers shows that all of these band crossings, including those at the NPs, exhibit non-zero topological charges as predicted above. In turn, all of the FSs carry substantial Berry curvatures. The numerical analysis shows that these Berry curvatures become extremal at the NPs and close to the four-fold and Weyl points (Extended Data Fig. 3). By the bulk–boundary correspondence[3,4], the non-trivial topology of these band crossings generates large Fermi arcs on the surface, which extend over half of the BZ of the surface (Extended Data Fig. 4). These arguments may be extended to 254 of the 1,651 magnetic SGs, of which 33 have NPs whose topological charges are enforced to be non-zero by symmetry alone (Supplementary Note 3).

## Calculated electronic structure

Figure 1e shows the density functional theory (DFT) band structure of MnSi, taking into account spin–orbit coupling, for the experimental moment of 0.41 Bohr magnetons ($\mu_B$) per Mn atom along the [010] direction (Methods, Extended Data Fig. 5). Ten bands are found to cross the Fermi level (Fig. 1e). In agreement with our symmetry analysis and the tight-binding model (Fig. 1d), we find the same generic band crossings, namely: (1) NPs on the BZ boundaries $k_x = \pm\pi$ and $k_z = \pm\pi$; (2) an odd number of Weyl points along $Y_1$–$\Gamma$–Y; and (3) an odd number of four-fold points along $R_1$–U–R.

The calculated FSs as matched to experiment are shown in Fig. 1f, highlighting the NPs at the BZ boundaries at $k_x = \pm\pi$ and $k_z = \pm\pi$ (see Extended Data Table 1 for key parameters and Extended Data Fig. 5). Eight FS sheets centred at $\Gamma$ comprise two small isolated hole pockets (sheets 1 and 2), two intersecting hole pockets with avoided crossings and magnetic breakdown due to spin–orbit coupling (sheets 3 and 4) and two pairs of jungle-gym-type sheets (sheets 5 and 6, and sheets 7 and 8). Sheets 9 and 10 are centred at R, comprising eight three-fingered electron pockets around the [111] axes and a tiny electron pocket, respectively. The sheet pairs (5, 6), (7, 8) and (9, 10) extend beyond the BZ boundaries with pairwise sticking at the NPs. They represent TPs (marked in red) with extremal Berry curvatures protected by the magnetic screw rotations $\theta\bar{C}_2^x$ and $\theta\bar{C}_2^z$. In contrast, sheets 5 to 10 do not form TPs at the BZ boundary $k_y = \pm\pi$, because the moment pointing along [010] breaks $\theta\bar{C}_2^y$.

Rotating the direction of the magnetization away from [010] distorts the FS sheets, where TPs exist only on those BZ boundaries parallel to the magnetization (Supplementary Videos 1 and 2). For instance, rotating the moments within the $x$–$y$ plane away from [010] breaks the magnetic screw rotation $\theta\bar{C}_2^x$, but keeps $\theta\bar{C}_2^z$ intact. In turn, the TPs gap out on the $k_y = \pm\pi$ and $k_x = \pm\pi$ planes, whereas they remain degenerate at the $k_z = \pm\pi$ planes (Extended Data Fig. 1, Supplementary Note 1).

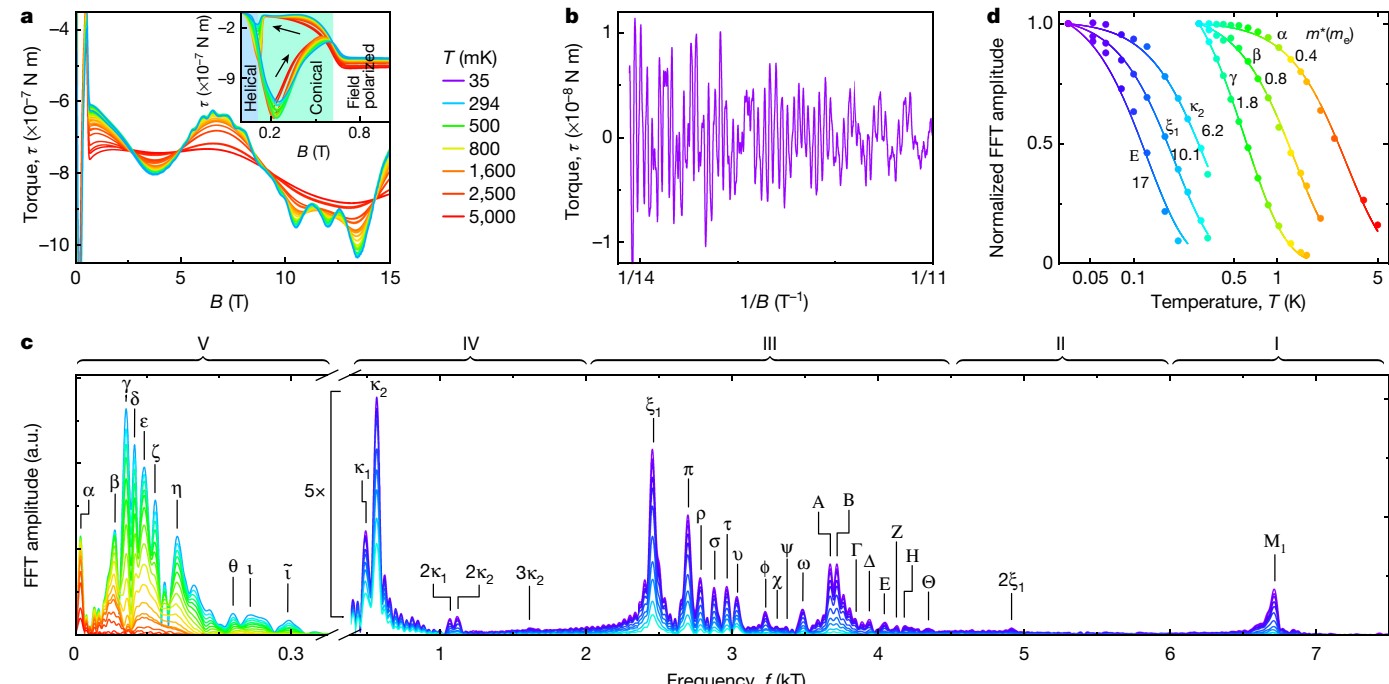

**Fig. 2 | Typical dHvA data of ferromagnetic MnSi. a**, dHvA oscillations detected in the magnetic torque $\tau$ as a function of magnetic field for a fixed field direction $\varphi = 82.5°$. Different colours represent different temperatures within the range 0.035 K to 5 K. The inset shows the hysteretic behaviour in the regime of the helical and conical phases at low fields. **b**, High-field part of the magnetic torque $\tau(1/B)$ at $T = 35$ mK with low-frequency components removed. **c**, FFT spectra of $\tau(1/B)$ for the same field angle and temperature range as in **a**. The spectra naturally group into five regimes (labelled by I–V), each of which exhibits a number of pronounced dHvA frequencies (Greek letters). **d**, Normalized FFT amplitudes of six selected dHvA frequencies as a function of temperature. The lines represent fits to the Lifshitz–Kosevich formula, from which we obtain the effective masses $m^*$ for the corresponding extremal FS orbits.

## Experimental results

To experimentally prove the mechanism causing generic TPs at the intersection of the FS with symmetry-enforced NPs and their dependence on the direction of the magnetization, we mapped out the FS by means of the de Haas–van Alphen (dHvA) effect using capacitive cantilever magnetometry (Methods, Extended Data Fig. 5, Supplementary Note 4). In the following, we focus on magnetic field rotations in the (001) plane, where $\varphi$ denotes the angle of the field with respect to [100]. This plane proves to be sufficient to infer the main FS features. Complementary data for the (001) and $(\bar{1}\bar{1}0)$ planes are presented in Extended Data Fig. 5. Typical torque data at different temperatures for $\varphi = 82.5°$ (Fig. 2a, b) show pronounced dHvA oscillations for magnetic fields exceeding $B \approx 0.7$ T. The hysteretic behaviour below about 0.7 T (Fig. 2a, inset) originates from the well understood helimagnetic and conical phases[30]. Figure 2b shows the oscillatory high-field part of the torque $\tau(1/B)$ at temperature $T = 35$ mK with the low-frequency components removed for clarity. To extract the dHvA frequencies, a fast Fourier transform (FFT) analysis of $\tau(1/B)$ was carried out, where the effects of demagnetizing fields and the unsaturated magnetization were taken into account (Methods). The FFT frequencies correspond to extremal FS cross-sections in low effective fields of about 0.7–1.9 T (Methods).

Typical dHvA frequencies and FFT amplitudes, shown for $\varphi = 82.5°$ in Fig. 2c, show five different regimes of dHvA frequencies labelled I to V. They comprise over 40 dHvA frequencies corresponding to different extremal FS orbits, as denoted by Greek letters (Fig. 2c, Extended Data Table 2). In our data analysis, we delineated artefacts due to the finite FFT window, such as the side lobes between $\kappa_2$ and $2\kappa_1$, or $3\kappa_2$ and $\xi_1$ (Methods). Fitting the temperature dependence of the FFT amplitudes within Lifshitz–Kosevich theory[31], the effective masses for each of the orbits were deduced ranging from $m^* = 0.4m_e$ to $m^* = 17m_e$, where $m_e$ is the bare electron mass (Fig. 2d).

To relate the dHvA frequencies to the calculated FS orbits, the torque amplitude was inferred from the DFT band structure by means of the Lifshitz–Kosevich formalism, using small rigid band shifts of the order of 10 meV to improve the matching following convention (Methods, Extended Data Table 1). The assignment to experiment was based on the consistency between dHvA frequency, angular dispersion, strength of torque signal, field dependence of the dHvA frequencies, effective masses and presence of magnetic breakdown, as explained in Methods, Extended Data Table 2, Extended Data Figs. 6, 7, Supplementary Note 5.

Figure 3a shows an intensity map of the experimental data of the (001) plane as a function of $\varphi$, where the theoretical dHvA branches are depicted by coloured lines (colours correspond to the FS sheets in Fig. 1). For comparison, Figure 3b shows an intensity map of the calculated dHvA spectra, where the experimental frequencies are marked by grey crosses.

For regimes I to IV, featuring contributions of the large FS sheets (5, 6) and (7, 8), all frequencies may be assigned unambiguously (Extended Data Figs. 6, 7, Supplementary Note 5). Namely, regime I contains the loop orbits around U associated with pair (5, 6) (blue and orange) and the neck orbit of sheet 8 (yellow). Regime II exhibits the dHvA branches originating from neck orbits around $\Gamma$–Y–$\Gamma$ on sheet 7 (purple). The neck orbits of sheet 8 (yellow), which evades detection because of the large slope of the dispersion, its high mass and the suppression of the magnetic torque near the [010] high-symmetry direction, is consistent with an anomalous frequency splitting at the expected crossing with the loop orbits of pair (5, 6) (blue and orange) around 6.5 kT (Supplementary Note 5). Regime III arises from both pairs (5, 6) and (7, 8), that is, neck orbits around $\Gamma$–Y–$\Gamma$ of (5, 6) and loop orbits around U of (7, 8). The remaining cascade of frequencies in regime III reflects breakdown orbits (translucent yellow) arising from avoided crossings between sheets 3 and 4 (red and green). Regime IV is, finally, dominated by sheet 2 of the isolated hole pocket and the first harmonic of sheet 2.

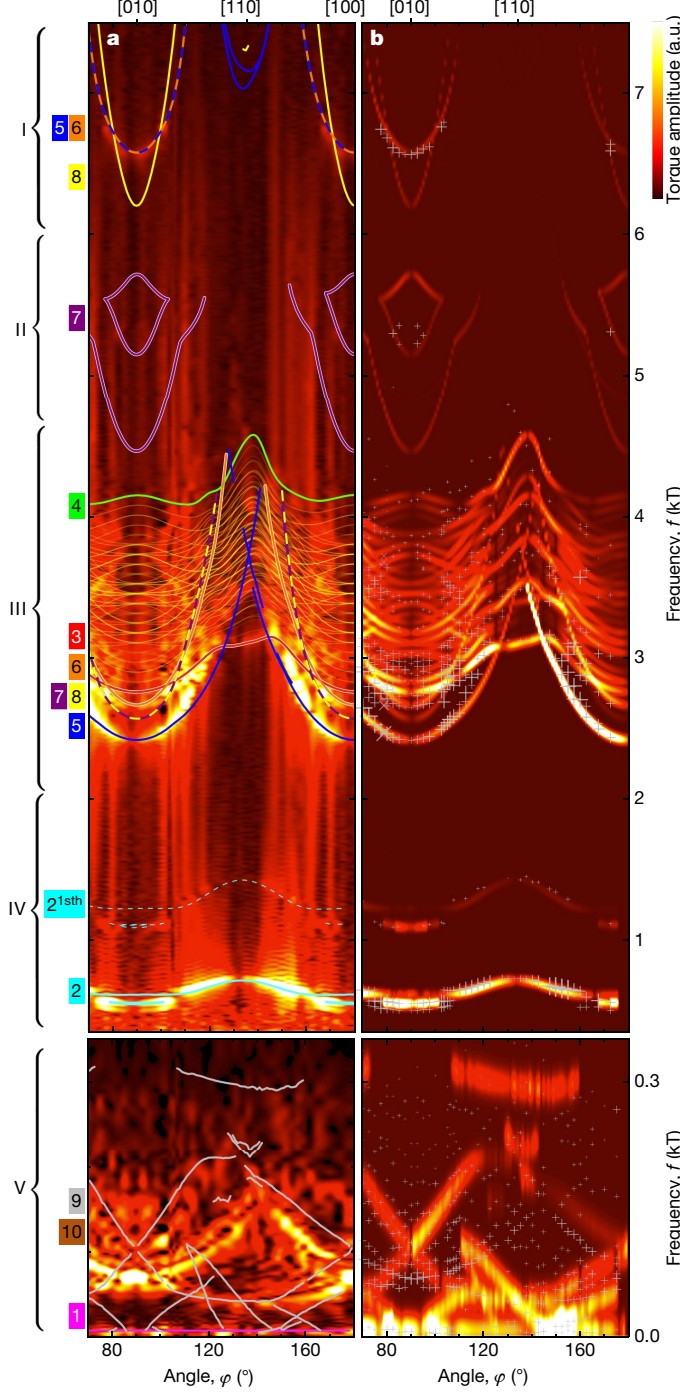

**Fig. 3 | Experimental and theoretical dHvA spectra in the (001) plane as a function of field angle φ. a**, FFT amplitudes of the experimentally observed dHvA spectra at $T = 280$ mK as a function of frequency $f$ and field angle $\varphi$. The thin coloured lines represent the theoretical dHvA branches, calculated from the ab initio band structure, where the colour and number indicates the FS sheet (Fig. 1f) from which the dHvA branch originates. The first harmonic (1sth) of the branches originating from sheet 2 is also labelled for clarity. A line cut of this colour map for fixed field angle $\varphi = 82.5°$ is shown in Fig. 2c. More than 40 dHvA branches were observed as listed in Extended Data Table 2. **b**, Torque amplitudes of the dHvA spectra inferred from the ab initio band structure (Methods), as a function of $f$ and $\varphi$, with the experimental frequencies of the dHvA branches indicated by crosses. To obtain a quantitative matching between theoretical and experimental dHvA branches, small rigid energy shifts to the ab initio bands were applied, as summarized in Extended Data Table 1, Supplementary Note 4. The detailed procedure how the experimental and theoretical dHvA branches were matched is described in the main text and in Supplementary Note 5.

As the magnetic torque generically vanishes at high-symmetry directions, which corresponds to the ⟨100⟩ axes in regimes I to IV, the associated FS sheets are centred at the Γ point. Likewise, the lowest frequency in regime V corresponds to a Γ-centred FS sheet, which can be assigned to the small hole pocket of sheet 1. In stark contrast, for regime V above about 0.05 kT, the high-symmetry directions correspond to the ⟨111⟩ axes, whereas the torque for the ⟨100⟩ axes is finite (see also Fig. 3b, Extended Data Fig. 5g). Hence, regime V is related to FS pockets in the vicinity of the R point that may be assigned to FS sheets (9, 10). This allows for a basic estimate of the size and the effective mass of FS sheets (9, 10) without the need for a detailed account of their shape, completing the assignment. The calculations demonstrate the presence of symmetry-enforced crossings of sheets (9, 10) if they intersect the NPs (Fig. 1).

To confirm that we observed the entire FS, we calculated the Sommerfeld coefficient of the specific heat from the density of states at the Fermi level as rescaled by the measured mass enhancements (Extended Data Table 1). Excellent agreement is observed within a few percent of experiment[32], $\gamma \approx 28$ mJ mol$^{-1}$ K$^{-2}$ at $B = 12$ T. This analysis reveals, that sheets (5, 6), (7, 8) and (9, 10), which form TPs, contribute 86% to the total density of states at the Fermi level.

## Topological NPs

Spectroscopic evidence of the symmetry-enforced topological band degeneracies at the BZ boundaries may be inferred from FS sheets (5, 6). Identical characteristics are observed for FS sheets (7, 8) (Extended Data Fig. 7, Supplementary Note 5). We note that the dHvA cyclotron orbits are perpendicular to the NPs for fundamental reasons, piercing through them at specific points of the TPs. As shown in Fig. 4a, a magnetic field parallel to [010] leads to extremal cross-sections for FS sheets (5, 6), supporting cyclotron orbits in the vicinity of the U and the $Y_1$ points on planes depicted by blue and green shading, respectively. Centred with respect to the U point are possible cyclotron orbits comprising different segments of FS sheets 5 and 6, which interact at TP1 to TP4 with the BZ boundaries at $k_x = \pm\pi$ and $k_z = \pm\pi$. In the absence of the non-symmorphic symmetries, these intersections would exhibit anticrossing and magnetic breakdown, leading to several orbits with different cross-sections and hence several dHvA frequencies. Instead, the behaviour is distinctly different to magnetic breakdown or Klein tunnelling[33,34].

As the BZ boundaries at $k_x = \pm\pi$ and $k_z = \pm\pi$ represent symmetry-enforced NPs, the crossing points of sheets 5 and 6 at TP1 to TP4 are, hence, protected band degeneracies at which the wavefunctions are orthogonal, that is, TP1 to TP4 are part of the TPs that suppress transitions between orbits (we call orbits containing at least one TP 'topological orbits'). In turn, two independent topological orbits (topological orbits 1 and 2) with identical areas and hence the same dHvA frequencies are expected (Fig. 4b, top). This is in excellent agreement with experiment, which shows a single dHvA frequency for field parallel [010] ($\varphi = 90°$ in Fig. 4c). Rotating the direction of the magnetic field within the $x$–$y$ plane away from [010], the NP at $k_x = \pm\pi$ gaps out, whereas the NP at $k_z = \pm\pi$ remains protected. Thus, the associated loop orbits around U (Fig. 4b, bottom) continue to include two points on the FS at $k_z = \pm\pi$ (TP3 and TP4), leading to two additional topological orbits (topological orbits 3 and 4) of identical cross-section with the same dHvA frequency, in perfect agreement with the observed spectra (Fig. 4c).

Comparing the extremal cross-sections of the neck orbits around Γ–$Y_1$–Γ with those around Γ–X–Γ, the latter crosses an NP whereas the former does not. With respect to Γ–X–Γ, there would be two extremal cross-sections with identical areas, positioned symmetrically with respect to X (Fig. 4d, top left), whereas for the cross-sections with respect to Γ–$Y_1$–Γ there are two extremal orbits with different areas positioned asymmetrically with respect to $Y_1$ (Fig. 4d, bottom left). Thus, within our symmetry analysis and our DFT calculations, we expect

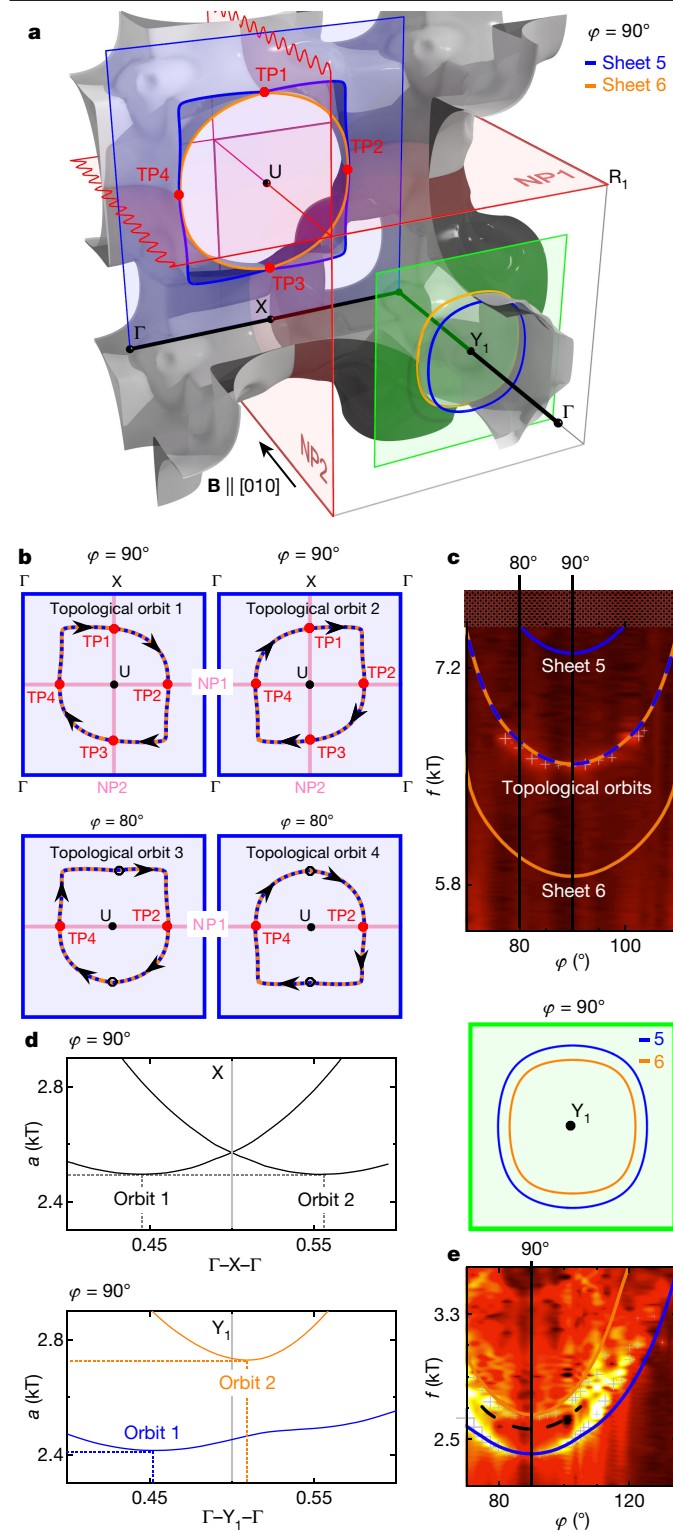

**Fig. 4 | Extremal orbits and spectroscopic signatures of NPs and TPs.** Identical features presented here for sheet pair (5, 6) are also observed for FS sheet pair (7, 8) (Extended Data Fig. 7, Supplementary Note 5). **a**, FS sheet pair (5, 6) for a field ($B$) along the [010] direction (for an alternative colour shading see Extended Data Fig. 6d1). Planes illustrating loop- and neck-type orbits around the U point and the $\Gamma$–$Y_1$–$\Gamma$ line are indicated by blue and green shading, respectively. Loop orbits with respect to the U point intersect at TP1 to TP4 with the NPs on the $k_x = \pi$ and $k_z = \pi$ BZ boundaries. The NPs enforce degeneracies at TP1 to TP4, where the wave functions are orthogonal. **b**, Instead of anticrossing and magnetic breakdown, topological orbits stabilize. Top: cross-sectional areas under field along [010] at $\varphi = 90°$. Bottom: schematic cross-sectional areas under rotated field for $\varphi = 80°$. **c**, Intensity map of dHvA spectra in the regime of loop- and neck-type orbits around the U point (Fig. 3). The spectra are in excellent agreement with the topological orbits. No evidence for independent orbits of FS sheets 5 and 6 are observed. **d**, Top left: symmetrical positions of extremal orbits 1 and 2 in a plot of the FS cross-sectional area $a$ along $\Gamma$–X–$\Gamma$ with respect to the NP at the X point. The orbits give rise to identical dHvA frequencies. Note that these orbits are not accessible experimentally. Bottom left: asymmetrical position of extremal FS cross-sections along $\Gamma$–$Y_1$–$\Gamma$ with respect to the BZ boundary at $k_y = \pm\pi$. Top right: the associated orbits give rise to different dHvA frequencies. **e**, Intensity map of dHvA spectra in the regime of neck-type orbits around the $Y_1$ point (Fig. 3). Spectra are in excellent agreement with two orbits as shown in **d** (bottom left and top right), that is, no NP at the BZ boundary at $k_y = \pm\pi$ containing $Y_1$.

various properties, such as anomalous Hall currents[35] or the nonlinear optical responses[36]. Indeed, large anomalous contributions to the Hall response are in excellent quantitative agreement with ab initio calculations, where the calculated FS and Berry curvatures were essentially identical to the FS we report here[37]. Our calculations imply also sizeable Fermi arcs at the surface of MnSi and related magnetic compounds such as FeGe and $Fe_{1-x}Co_xSi$, connecting the topological charge of the NPs directly with a Weyl point (Extended Data Fig. 4). These Fermi arcs reflect the presence of duos of NPs. Analogous Fermi arcs will not exist in non-magnetic materials with SG 198[25–27], which support trios of NPs (Supplementary Note 1).

In systems with symmetry-enforced NPs and TPs, tiny changes of the direction of the magnetization will control the topological band crossing in the bulk and the Fermi arcs, causing massive changes of Berry curvature that may be exploited technologically. The formation of TPs irrespective of the complexity of the FS raises the question of whether they affect the transport properties[38] and enable exotic states of matter[39]. Extending the analysis presented here to all 1,651 magnetic SGs, we find that there is a large number of candidate materials, such as $CoNb_3S_6$ (ref. [40]) or $Nd_5Si_3$ (ref. [41]) with similar TPs (Extended Data Table 3, Supplementary Note 3), which await to be explored from a fundamental point of view and harnessed for future technologies.

## Online content

1. von Neumann, J. & Wigner, E. Über das Verhalten von Eigenwerten bei adiabatischen Prozessen. *Z. Phys.* **30**, 467–470 (1929).
2. Herring, C. Accidental degeneracy in the energy bands of crystals. *Phys. Rev.* **52**, 365–373 (1937).
3. Chiu, C.-K., Teo, J. C. Y., Schnyder, A. P. & Ryu, S. Classification of topological quantum matter with symmetries. *Rev. Mod. Phys.* **88**, 035005 (2016).
4. Armitage, N. P., Mele, E. J. & Vishwanath, A. Weyl and Dirac semimetals in three-dimensional solids. *Rev. Mod. Phys.* **90**, 015001 (2018).
5. Burkov, A. Weyl metals. *Annu. Rev. Condens. Matter Phys.* **9**, 359–378 (2018).

a single dHvA branch for neck orbits parallel to a NP compared with two dHvA branches for neck orbits that are not parallel to a NP (Fig. 4d, top right). Keeping in mind that only neck orbits around $Y_1$ are accessible experimentally, we clearly observe two branches, giving strong evidence that there are no NPs on the $k_y = \pm\pi$ BZ boundary (Fig. 4e).

## Concluding remarks

The symmetry-enforced NPs and TPs that are generically located at the Fermi level, which support large Berry curvatures, may account for

6. Wang, Q. et al. Large intrinsic anomalous Hall effect in half-metallic ferromagnet $Co_3Sn_2S_2$ with magnetic Weyl fermions. *Nat. Commun*. **9**, 3681 (2018); correction **9**, 4212 (2018).

7. Huang, X. et al. Observation of the chiral-anomaly-induced negative magnetoresistance in 3D Weyl semimetal TaAs. *Phys. Rev. X* **5**, 031023 (2015).

8. Liang, S. et al. Experimental tests of the chiral anomaly magnetoresistance in the Dirac–Weyl semimetals $Na_3Bi$ and GdPtBi. *Phys. Rev. X* **8**, 031002 (2018).

9. Huang, S.-M. et al. A Weyl fermion semimetal with surface Fermi arcs in the transition metal monopnictide TaAs class. *Nat. Commun*. **6**, 7373 (2015).

10. Michel, L. & Zak, J. Elementary energy bands in crystals are connected. *Phys. Rep*. **341**, 377–395 (2001).

11. Young, S. M. et al. Dirac semimetal in three dimensions. *Phys. Rev. Lett*. **108**, 140405 (2012).

12. Furusaki, A. Weyl points and Dirac lines protected by multiple screw rotations. *Sci. Bull*. **62**, 788–794 (2017).

13. Zhao, Y. X. & Schnyder, A. P. Nonsymmorphic symmetry-required band crossings in topological semimetals. *Phys. Rev. B* **94**, 195109 (2016).

14. Zhang, J. et al. Topological band crossings in hexagonal materials. *Phys. Rev. Mater*. **2**, 074201 (2018).

15. Yu, Z.-M., Wu, W., Zhao, Y. X. & Yang, S. A. Circumventing the no-go theorem: a single Weyl point without surface Fermi arcs. *Phys. Rev. B* **100**, 041118 (2019).

16. Wu, W. et al. Nodal surface semimetals: theory and material realization. *Phys. Rev. B* **97**, 115125 (2018).

17. Türker, O. & Moroz, S. Weyl nodal surfaces. *Phys. Rev. B* **97**, 075120 (2018).

18. Xiao, M. & Fan, S. Topologically charged nodal surface. Preprint at https://arxiv.org/abs/1709.02363 (2017).

19. Yang, Y. et al. Observation of a topological nodal surface and its surface-state arcs in an artificial acoustic crystal. *Nat. Commun*. **10**, 5185 (2019).

20. Xiao, M. et al. Experimental demonstration of acoustic semimetal with topologically charged nodal surface. *Sci. Adv*. **6**, eaav2360 (2020).

21. Chang, G. et al. Topological quantum properties of chiral crystals. *Nat. Mater*. **17**, 978–985 (2018).

22. Lonzarich, G. G. Magnetic oscillations and the quasiparticle bands of heavy electron systems. *J. Magn. Magn. Mater*. **76–77**, 1–10 (1988).

23. Mühlbauer, S. et al. Skyrmion lattice in a chiral magnet. *Science* **323**, 915–919 (2009).

24. Pfleiderer, C., McMullan, G. J., Julian, S. R. & Lonzarich, G. G. Magnetic quantum phase transition in MnSi under hydrostatic pressure. *Phys. Rev. B* **55**, 8330–8338 (1997).

25. Sanchez, D. S. et al. Topological chiral crystals with helicoid-arc quantum states. *Nature* **567**, 500–505 (2019).

26. Rao, Z. et al. Observation of unconventional chiral fermions with long Fermi arcs in CoSi. *Nature* **567**, 496–499 (2019).

27. Schröter, N. B. M. et al. Observation and control of maximal Chern numbers in a chiral topological semimetal. *Science* **369**, 179–183 (2020).

28. Kramers, H. A. Théorie générale de la rotation paramagnétique dans les cristaux. *Proc. Amsterdam Acad*. **33**, 959–972 (1930).

29. Nielsen, H. & Ninomiya, M. A no-go theorem for regularizing chiral fermions. *Phys. Lett. B* **105**, 219–223 (1981).

30. Bauer, A. et al. Symmetry breaking, slow relaxation dynamics, and topological defects at the field-induced helix reorientation in MnSi. *Phys. Rev. B* **95**, 024429 (2017).

31. Shoenberg, D. *Magnetic Oscillations in Metals* (Cambridge Univ. Press, 1984).

32. Bauer, A. et al. Quantum phase transitions in single-crystal $Mn_{1-x}Fe_xSi$ and $Mn_{1-x}Co_xSi$: crystal growth, magnetization, ac susceptibility, and specific heat. *Phys. Rev. B* **82**, 064404 (2010).

33. Alexandradinata, A. & Glazman, L. Geometric phase and orbital moment in quantization rules for magnetic breakdown. *Phys. Rev. Lett*. **119**, 256601 (2017).

34. van Delft, M. R. et al. Electron–hole tunneling revealed by quantum oscillations in the nodal-line semimetal HfSiS. *Phys. Rev. Lett*. **121**, 256602 (2018).

35. Xiao, D., Chang, M.-C. & Niu, Q. Berry phase effects on electronic properties. *Rev. Mod. Phys*. **82**, 1959–2007 (2010).

36. Morimoto, T. & Nagaosa, N. Topological nature of nonlinear optical effects in solids. *Sci. Adv*. **2**, e1501524 (2016).

37. Franz, C. et al. Real-space and reciprocal-space Berry phases in the Hall effect of $Mn_{1-x}Fe_xSi$. *Phys. Rev. Lett*. **112**, 186601 (2014).

38. Smith, M. F. Small-angle interband scattering as the origin of the $T^{3/2}$ resistivity in MnSi. *Phys. Rev. B* **74**, 172403 (2006).

39. Grover, T. & Fisher, M. P. A. Quantum disentangled liquids. *J. Stat. Mech*. **1014**, P10010 (2014).

40. Tenasini, G. et al. Giant anomalous Hall effect in quasi-two-dimensional layered antiferromagnet $Co_{1/3}NbS_2$. *Phys. Rev. Res*. **2**, 023051 (2020).

41. Boulet, P., Weizer, F., Hiebl, K. & Noël, H. Structural chemistry, magnetism and electrical properties of binary Nd silicides. *J. Alloys Compd*. **315**, 75–81 (2001).

## Methods

### Sample preparation

For our study, two MnSi samples were prepared from a high-quality single crystalline ingot obtained by optical float-zoning[42]. The samples were oriented by X-ray Laue diffraction and cut into $1 \times 1 \times 1\,\mathrm{mm}^3$ cubes with faces perpendicular to [100], [110], and [110] and [110], and [111] and [112] cubic equivalent directions, respectively. Both samples exhibited a residual resistivity ratio close to 300.

### Experimental methods

Quantum oscillations of the magnetization, that is, the dHvA effect, was measured by means of cantilever magnetometry measuring the magnetic torque $\boldsymbol{\tau} = \mathbf{m} \times \mathbf{B}$. The double-beam type cantilevers sketched in Extended Data Fig. 5e were obtained from CuBe foil by standard optical lithography and wet-chemical etching. The cantilever position was read out in terms of the capacitance between the cantilever and a fixed counter electrode using an Andeen-Hagerling AH2700A capacitance bridge, similar to the design described in refs. [43,44].

Angular rotation studies were performed in a $^3$He insert with a manual rotation stage at a base temperature $T = 280\,\mathrm{mK}$ under magnetic fields up to 15 T. In addition, the effective charge carrier mass was determined using a dilution refrigerator insert with fixed sample stage under magnetic fields up to 14 T (16 T using a Lambda stage) at temperatures down to 35 mK.

We discuss partial rotations in the (001) and ($\bar{1}\bar{1}0$) crystallographic planes. The angle $\varphi$ is measured from [100] in the (001) plane and the angle $\theta$ is measured from [001] in the ($\bar{1}\bar{1}0$) plane. Corresponding data are shown in Fig. 3a and Extended Data Fig. 5g. Owing to the topology of the FS and the simple cubic BZ, the (001) plane rotation shows most of the extremal orbits and is already sufficient for an assignment to the FS sheets. For this reason, the discussion of the dHvA data in the main text focuses on the rotation in the (001) plane.

The response of the cantilever was calibrated by means of the electrostatic displacement, taking into account the cantilever bending line obtained from an Euler–Bernoulli approach[45]. Applying a d.c. voltage, $U$, to the capacitance $C_0 = \varepsilon_0 A / d_0$, defined by the area $A$, the plate distance $d_0$ and the vacuum permittivity $\varepsilon_0$, leads to an electrostatic force $F = C_0 U^2 / 2 d_0$. This force is equivalent to a torque $\tau = \beta F L$, where $L$ is the effective beam length and $\beta = 0.78$ is a geometry-dependent prefactor accounting for the different mechanical response of a bending beam to a torque and force, respectively. From this, the calibration constant $K(C) = \tau / \Delta C$ quantifying the capacitance change $\Delta C$ in response to the torque was obtained for different values of $C$. Changes in $K(C)$ up to 10% were recorded during magnetic field sweeps. The torque was calculated using

$$\tau(C) = \int_{C_0}^{C} K(C')\,\mathrm{d}C'. \tag{1}$$

### Evaluation of the dHvA signal

The dependence of the capacitance, $C(B_{\mathrm{ext}})$, was converted into torque and corrected as described below, where $B_{\mathrm{ext}}$ is the applied magnetic field. An exemplary torque curve obtained at $T = 280\,\mathrm{mK}$ and $\varphi = 82.5°$ is shown in Fig. 2a. In the regime below $B \approx 0.7\,\mathrm{T}$ the transitions from helical to conical and field-polarized state generated a strongly hysteretic behaviour. At higher fields, magnetic quantum oscillations on different amplitude and frequency scales could be readily resolved. The first low-frequency components appeared at magnetic fields as low as $B \approx 4\,\mathrm{T}$, whereas several high-frequency components, corresponding to larger extremal cross-sections, could only be resolved in high fields (Fig. 2b). Consequently, the data acquisition and evaluation was optimized by treating low- and high-frequency components separately.

To eliminate the non-oscillatory component of the signal, low-order polynomial fits or curves obtained by adjacent averaging over suitable field intervals were subtracted from the data, producing consistent results. FFTs of $\tau(1/B)$ were used to determine the frequency components contained in the signal. Field sweeps were performed from 0 T to 15 T at $0.03$–$0.04\,\mathrm{T\,min^{-1}}$ and from 15 T to 10 T at $0.008\,\mathrm{T\,min^{-1}}$. FFTs over the range 4 T to 15 T (10 T to 15 T) were performed to evaluate frequency components below (above) $f = 350\,\mathrm{T}$ for measurements in the $^3$He insert and from 10 T to 14 T (11 T to 16 T with Lambda stage) in the dilution refrigerator. The values correspond to the applied field before taking into account demagnetization. Rectangular FFT windows were chosen to maximize the ability to resolve closely spaced frequency peaks. See Supplementary Note 4 for details.

### Internal magnetic field and dHvA frequency $f(B)$ in a weak itinerant magnet

MnSi is a weak ferromagnet with an unsaturated magnetization up to the largest magnetic fields studied. This results in two different peculiarities concerning the observed dHvA frequencies. (1) The field governing the quantum oscillations is the internal field[31] $\mathbf{B}_{\mathrm{int}} = \mu_0 \mathbf{H}_{\mathrm{ext}} + \mu_0 (1 - N_{\mathrm{d}}) \mathbf{M}$, where $\mu_0$ is the vacuum permeability, $\mathbf{H}_{\mathrm{ext}}$ is the applied magnetic field and $\mathbf{M}$ is the magnetization. Taking into account the demagnetization factor[46] $N_{\mathrm{d}} = 1/3$ for a cubic sample to first order yields a field correction $\Delta B = B_{\mathrm{int}} - B_{\mathrm{ext}} = \frac{2}{3}\mu_0 M_{\mathrm{exp}} \approx 0.131\,\mathrm{T}$, where $M_{\mathrm{exp}}$ is the low-field value of the magnetization in the field-polarized phase determined experimentally. The applied field was corrected by this value. The field dependence of the magnetic moment yields only a minor correction of the internal field that may be neglected. (2) The effect of the unsaturated magnetization on the Fermi surface is more prominent and may be described in a good approximation as a rigid Stoner exchange splitting that scales with the magnitude of the magnetization. Consequently, FS cross-sectional areas are enlarged with increasing $B$ for the majority electron orbits and minority hole orbits. Cross-sectional areas shift downwards for majority hole and minority electron orbits.

This change in cross-sectional area is not directly proportional to the change in the observed dHvA frequencies $f$, that is, the dHvA frequencies deviate from the field-dependent frequency $f_{\mathrm{B}}(B) = \frac{\hbar}{2\pi e} A_k(B)$ obeying the Onsager relation (here $A_k$ is the extremal cross-sectional area in $k$-space, $\hbar$ is the reduced Planck constant and $e$ is the electron charge). The frequency $f$ observed may be inferred[47] from the derivative of the dHvA phase factor $2\pi\left(\frac{f_{\mathrm{B}}(B)}{B} - \gamma\right) \pm \frac{1}{4}$ with respect to $1/B$:

$$f(B) = \frac{\mathrm{d}}{\mathrm{d}B^{-1}}\left(\frac{f_{\mathrm{B}}(B)}{B}\right) = f_{\mathrm{B}}(B) - B\frac{\mathrm{d}f_{\mathrm{B}}(B)}{\mathrm{d}B}. \tag{2}$$

Thus, a linear relation $f_{\mathrm{B}}(B)$ results in a constant $f(B)$. This may be understood intuitively, because a linear term in $f_{\mathrm{B}}(B)$ leads only to a phase shift since the oscillations are periodic in $1/B$. Equation (2) shows that $f(B)$ is the zero-field intercept of the tangent to $f_{\mathrm{B}}(B)$.

In the Stoner picture of rigidly split bands, $f_{\mathrm{B}}(B)$ may be related to the magnetization[47,48] using

$$f_{\mathrm{B}}(B) - f_0 = \pm\frac{m_{\mathrm{b}}}{m_{\mathrm{e}}}\frac{Is}{4\mu_{\mathrm{B}}^2}M(B), \tag{3}$$

where $I$ is the Stoner exchange parameter, $m_{\mathrm{b}}$ is the band mass, the $\pm$ is for electron and hole orbits, respectively, $s = \pm 1$ is the spin index and $f_0$ is the hypothetical frequency without exchange splitting. Note, that this model is only meaningful in the field-polarized regime $B \gtrsim 0.7\,\mathrm{T}$. Using the experimental $M(B)$ curve of MnSi[32], we estimate that the frequencies $f(B)$ in the windows used for $f > 350\,\mathrm{T}$ defined above with centre fields $B_{\mathrm{average}} = 2B_{\mathrm{high}}B_{\mathrm{low}}/(B_{\mathrm{low}} + B_{\mathrm{high}})$ ranging from 11.8 T to 13.2 T correspond to the extremal cross-sections at $B \approx 1.7$–$1.9\,\mathrm{T}$ (Extended Data Fig. 5f). For the window used for frequencies $f < 350\,\mathrm{T}$, it is $B_{\mathrm{average}} = 6.5\,\mathrm{T}$ and

$f(B)$ corresponds to the extremal cross-sections at $B \approx 0.7$ T. Thus, even under large magnetic fields, the experimental frequency values correspond to a field-polarized state in a low field.

## Quantum oscillatory torque and Lifshitz–Kosevich equation

Evaluation and interpretation of the quantum oscillatory torque magnetization was performed using the Lifshitz–Kosevich formalism[31]. The components of **M** parallel ($\parallel$) and perpendicular ($\perp$) to the field are given by:

$$M_{\mathrm{osc},\parallel} = -\left(\frac{e}{\hbar}\right)^{3/2} \frac{e\hbar f B^{1/2} V}{m^* 2^{1/2} \pi^{5/2} \sqrt{A''}} \sum_{p=1}^{\infty} \frac{R_{\mathrm{T}} R_{\mathrm{D}}}{p^{3/2}} \sin\left(2\pi p\left(\frac{f}{B}-\gamma\right) \pm \frac{\pi}{4}\right), \quad (4)$$

and

$$M_{\mathrm{osc},\perp} = -\frac{1}{f}\frac{\partial f}{\partial \varphi} M_{\mathrm{osc},\parallel}. \quad (5)$$

where $V$ is the sample volume, $p$ is the harmonic index, $A''$ is the curvature of the cross-sectional area parallel to **B**, and $f$ is the dHvA frequency observed (see comments above). The phase $\gamma = 1/2$ corresponds to a parabolic band. In general, the phase includes also contributions due to Berry phases when the orbit encloses topologically non-trivial structures in $k$-space. The $\pm$ holds for maximal and minimal cross-sections, respectively. The torque amplitude is given by $\tau_{\mathrm{osc}} = M_{\mathrm{osc},\perp} B$. The torque thus vanishes in high-symmetry directions where $f(\varphi)$ is stationary. This feature of $\tau$ may be used to infer additional information about the symmetry properties of a dHvA branch. $R_{\mathrm{T}}$ describes the temperature dependence of the oscillations

$$R_{\mathrm{T}} = \frac{X}{\sinh(X)} \text{ with } X = \frac{2\pi^2 p m^* k_{\mathrm{B}} T}{e\hbar B}, \quad (6)$$

from which the effective mass $m^*$ including renormalization effects can be extracted, where, $k_{\mathrm{B}}$ is the Boltzmann constant. Equation (6) was fitted to the temperature dependence of the FFT peaks using the average fields $B_{\mathrm{average}}$ defined above. No systematic changes in the mass values were observed within the standard deviation of the fits when different window sizes were chosen. See Supplementary Note 4 for details. The Dingle factor

$$R_{\mathrm{D}} = \exp\left(-\frac{\pi p m^*}{eB\tau}\right) = \exp\left(-\frac{\pi p}{\omega_{\mathrm{c}}\tau}\right) \quad (7)$$

describes the influence of a finite scattering time $\tau$. Here, $\omega_{\mathrm{c}} = eB/m^*$ is the cyclotron frequency.

## DFT calculations

The band structure and FS sheets of MnSi in the field-polarized phase were calculated using DFT. The calculations included the effect of spin–orbit coupling. In all calculations, the magnetic part of the exchange-correlation terms was scaled[49] to match the experimental magnetic moment of $0.41\mu_{\mathrm{B}}$ per Mn atom at low fields. As input for the DFT calculations, the experimental crystal structure of MnSi was used, that is, space group $P2_13$ (198) with an experimental lattice constant $a = 4.558$ Å. Both Mn and Si occupy Wyckoff positions $4a$ with coordinates $(u, u, u)$, $(-u + 1/2, -u, u + 1/2)$, $(-u, u + 1/2, -u + 1/2)$, $(u + 1/2, -u + 1/2, -u)$ where $u_{\mathrm{Mn}} = 0.137$ and $u_{\mathrm{Si}} = 0.845$ (Extended Data Fig. 5a).

Calculations were carried out using WIEN2k[50], ELK[51] and VASP[52,53] using different versions of the local spin density approximation. The results are consistent within the expected reproducibility of current DFT codes[54]. The remaining uncertainties motivate a comprehensive experimental FS determination as reported in this study. In the main text, we focus on the results obtained with WIEN2k, using the local spin density approximation parametrization of Perdew and Wang[55] and a sampling of the full BZ with a $23 \times 23 \times 23$ Γ-centred grid. The results of Extended Data Figs. 1, 2, 4 were obtained using VASP with the PBE functional[56] and a BZ sampling with a $15 \times 15 \times 15$ $k$-mesh centred around Γ.

Bands used for the determination of the Fermi surface were calculated with WIEN2k on a $50 \times 50 \times 50$ $k$-mesh. Owing to the presence of spin–orbit coupling, but the absence of both inversion and time-reversal symmetry, band structure data had to be calculated for different directions of the spin quantization axis. For a given experimental plane of rotation, calculations were performed in angular steps of 10°. The bands were then interpolated $k$-point-wise using third-order splines to obtain band structure information in 1° steps.

For the prediction of the dHvA branches from the DFT results, the Supercell $k$-space Extremal Area Finder (SKEAF)[57] was used on interpolated data corresponding to $150 \times 150 \times 150$ $k$-points in the full BZ. The theoretical torque amplitudes shown in Fig. 3b were calculated directly from the prefactors in equations (4) and (5) convoluted with a suitable distribution function.

To compute the surface states of MnSi in the field-polarized phase (Extended Data Fig. 4), we first constructed a DFT-derived tight-binding model using the maximally localized Wannier function method as implemented in Wannier90[58]. Using this tight-binding model, we computed the momentum-resolved surface density of states by means of an iterative Green's function method, using WannierTools[59]. The symmetry eigenvalues of the DFT bands were computed from expectation values using VASP pseudo wavefunctions, as described in ref. [60].

## Magnetic breakdown

The probabilities for magnetic breakdown at a junction $i$ is given by $p_i = e^{-\frac{B_0}{B}}$. The probability for no breakdown to occur is thus $q_i = 1 - p_i$. The breakdown fields $B_0$ were calculated from Chamber's formula

$$B_0 = \frac{\pi\hbar}{2e}\sqrt{\frac{k_{\mathrm{g}}^3}{a+b}}, \quad (8)$$

where $k_{\mathrm{g}}$ is the gap in $k$-space and $a$ and $b$ are the curvatures of the trajectories at the breakdown junction[31]. In our study of MnSi, we observed magnetic breakdown in particular between sheets 3 and 4, which exhibit up to eight junctions depending on the magnetic field direction and between FS sheet pairs touching the BZ surfaces on which the NP degeneracy is lifted. Only breakdown orbits that are closed after one cycle are considered in the analysis. Further details can be found in the Supplementary Note 5.

## Assignment of dHvA orbits and rigid band shifts

The assignment of the experimental dHvA branches to the corresponding extremal FS cross-sections was based on the following criteria: (1) dHvA frequency—determining sheet size in terms of the cross-sectional area; (2) angular dispersion—relating to sheet shape, topology and symmetry; (3) torque signal strength—relating to sheet shape and symmetry; (4) direction of $f(B)$ shift—relating to spin orientation and charge carrier type; (5) effective mass—relating to the temperature dependence; (6) magnetic breakdown behaviour—relating to proximity of neighbouring sheets.

The majority of the observed dHvA branches could be related directly to the FS as calculated. In addition, we used the well-established procedure of small rigid band shifts to optimize the matching. While this procedure is, in general, neither charge nor spin conserving, it results in a very clear picture of the experimental FS. One has to bear in mind, however, that the deviations between the true FS and the calculated FS are not due to a rigid band shift (this might be justified, for example, in case of unintentional doping, which we rule out here). Rather, it may be attributed to differences in the band dispersions that originate in limitations of our DFT calculations (for example, neglecting electronic correlations and the coupling to the spin fluctuation spectrum).

The dHvA orbits, the assignments to a specific extremal cross-section, the observed and predicted frequencies, the observed and predicted masses and mass enhancements are listed in Extended Data Table 2. Extended Data Table 1 summarizes the resulting characteristic properties of the FS sheets including their contribution to the density of states at the Fermi level.

## Symmetry analysis

The symmetry-enforced band crossings and the band topology follow from the non-trivial winding of the symmetry eigenvalues through the BZ. This winding of the eigenvalues is derived in Supplementary Note 1, both for the paramagnetic and ferromagnetic phases of MnSi. Supplementary Note 1 also contains the derivation of the topological charges of the NPs, Weyl points and four-fold points, which are obtained from generalizations of the Nielsen–Ninomiya theorem[29]. To illustrate the band topology for ferromagnets in SG 19.27 and SG 4.9, two tight-binding models are derived in Supplementary Note 2, which includes also a discussion of the Berry curvature and the surface states. The classification of NPs in magnetic materials is given in Supplementary Note 3. It is found that among the 1,651 magnetic SGs, 254 exhibit symmetry-enforced NPs. We find that (at least) 33 of these have NPs whose topological charge is guaranteed to be non-zero due to symmetry alone.

## Data availability

Materials and additional data related to this paper are available from the corresponding authors upon reasonable request.

42. Neubauer, A. et al. Ultra-high vacuum compatible image furnace. *Rev. Sci. Instrum.* **82**, 013902 (2011).
43. Wilde, M. A. et al. Magnetometry on quantum Hall systems: thermodynamic energy gaps and the density of states distribution. *Phys. Status Solidi B* **245**, 344–355 (2008).
44. Wilde, M., Heitmann, D. & Grundler, D. *Magnetization of Interacting Electrons in Low-Dimensional Systems* Ch. 10, 245 (Springer Nanoscience and Technology, 2010).
45. Wilde, M. *Magnetization Measurements on Low-Dimensional Electron Systems in High-Mobility GaAs and SiGe Heterostructures*. PhD thesis, Universität Hamburg (2004).
46. Aharoni, A. Demagnetizing factors for rectangular ferromagnetic prisms. *J. Appl. Phys.* **83**, 3432–3434 (1998).
47. van Ruitenbeek, J. M. et al. A de Haas–van Alphen study of the field dependence of the Fermi surface in ZrZn₂. *J. Phys. F* **12**, 2919–2928 (1982).
48. Kimura, N. et al. de Haas–van Alphen effect in ZrZn₂ under pressure: crossover between two magnetic states. *Phys. Rev. Lett.* **92**, 197002 (2004).
49. Hoshino, T., Zeller, R., Dederichs, P. H. & Weinert, M. Magnetic energy anomalies of 3d systems. *Europhys. Lett.* **24**, 495–500 (1993).
50. Blaha, P. et al. Wien2k: an apw+lo program for calculating the properties of solids. *J. Chem. Phys.* **152**, 074101 (2020).
51. The Elk Code (GNU General Public License, 2021); https://elk.sourceforge.io/
52. Kresse, G. & Furthmüller, J. Efficiency of ab-initio total energy calculations for metals and semiconductors using a plane-wave basis set. *Comput. Mater. Sci.* **6**, 15–50 (1996).
53. Kresse, G. & Furthmüller, J. Efficient iterative schemes for ab initio total-energy calculations using a plane-wave basis set. *Phys. Rev. B* **54**, 11169–11186 (1996).
54. Lejaeghere, K. A. Reproducibility in density functional theory calculations of solids. *Science* **351**, aad3000 (2016).
55. Perdew, J. P. & Wang, Y. Accurate and simple analytic representation of the electron-gas correlation energy. *Phys. Rev. B* **45**, 13244–13249 (1992).
56. Perdew, J. P., Burke, K. & Ernzerhof, M. Generalized gradient approximation made simple. *Phys. Rev. Lett.* **77**, 3865–3868 (1996).
57. Rourke, P. & Julian, S. Numerical extraction of de Haas–van Alphen frequencies from calculated band energies. *Comput. Phys. Commun.* **183**, 324–332 (2012).
58. Mostofi, A. A. et al. Wannier90: a tool for obtaining maximally-localised Wannier functions. *Comput. Phys. Commun.* **178**, 685–699 (2008).
59. Wu, Q., Zhang, S., Song, H.-F., Troyer, M. & Soluyanov, A. A. Wanniertools: an open-source software package for novel topological materials. *Comput. Phys. Commun.* **224**, 405–416 (2018).
60. Gao, J., Wu, Q., Persson, C. & Wang, Z. Irvsp: to obtain irreducible representations of electronic states in the VASP. *Comput. Phys. Commun.* **261**, 107760 (2021).

**Acknowledgements** We thank D. Grundler, F. Rucker, A. Leonhardt, A. Rosch, T. Rapp, S. G. Albert and S. M. Sauther for support and discussions. Preliminary band structure calculations for a limited number of field orientations using FLEUR and JuDFT KKR-GGA were carried out in collaboration with F. Freimuth, B. Zimmermann and Y. Mokrousov in the very early stages of this study. M.A.W., A.B. and C.P. were supported through DFG TRR80 (project-id 107745057, project E1 and E3), DFG SPP 2137 (Skyrmionics) under grant number PF393/19 (project-id 403191981), DFG GACR Projekt WI 3320/3-1, ERC Advanced Grants number 291079 (TOPFIT) and number 788031 (ExQuiSid), and Germany's excellence strategy EXC-2111 390814868.

**Author contributions** M.A.W. and C.P. conceived the experiment and devised its interpretation together with A.P.S. A.B. prepared and characterized the samples. M.D. and M.A.W. conducted the measurements and analysed the data. M.A.W., A.N. and K.A. performed comprehensive band structure calculations. M.A.W. connected the experimental data with the calculated band structure. M.M.H., K.A. and A.P.S. performed the symmetry analysis and identified the topological properties of the band structure. M.M.H. and K.A. calculated the surface states and the Berry curvatures. M.A.W., A.P.S. and C.P. wrote the manuscript with contributions from M.M.H. and K.A. All authors discussed the data and commented on the manuscript.

**Funding** Open access funding provided by Max Planck Society.

**Competing interests** The authors declare no competing interests.

**Additional information**
**Correspondence and requests for materials** should be addressed to M.A.W., A.P.S. or C.P.

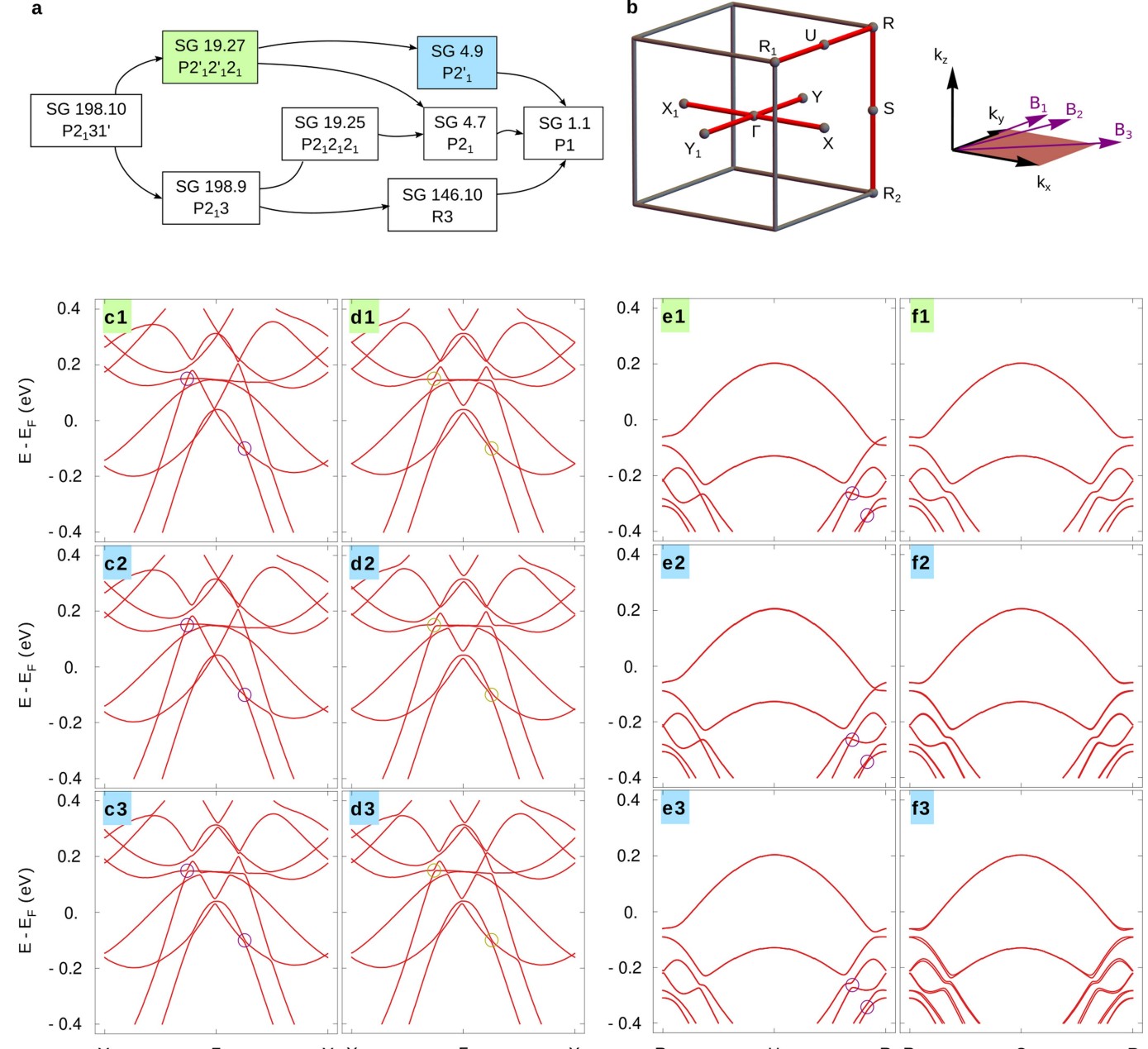

**Extended Data Fig. 1 | Magnetic space groups and electronic band structures for different directions of the magnetization. a**, Magnetic subgroups of space group 198 ($P2_13$) and their group–subgroup relations. The magnetic space groups describing the symmetries for magnetizations along [010] and within the $x$–$y$ plane are highlighted in green and blue, respectively. **b**, Orthorhombic BZ for ferromagnetic MnSi (left) and magnetization directions used for the ab initio calculations in **c**–**f** (right). **c**–**f**, Ab initio electronic band structure of ferromagnetic MnSi along the four high-symmetry paths indicated in **b**: $Y_1$–$\Gamma$–$Y$ (**c**), $X_1$–$\Gamma$–$X$ (**d**), $R_1$–$U$–$R$ (**e**) and $R_2$–$S$–$R$ (**f**). In the first, second and third rows, the magnetization is oriented along [010], 10° rotated into the $x$–$y$ plane and along [110], respectively. Some of the Weyl points and four-fold degenerate points at (or near) the high-symmetry lines are highlighted by violet and brown circles.

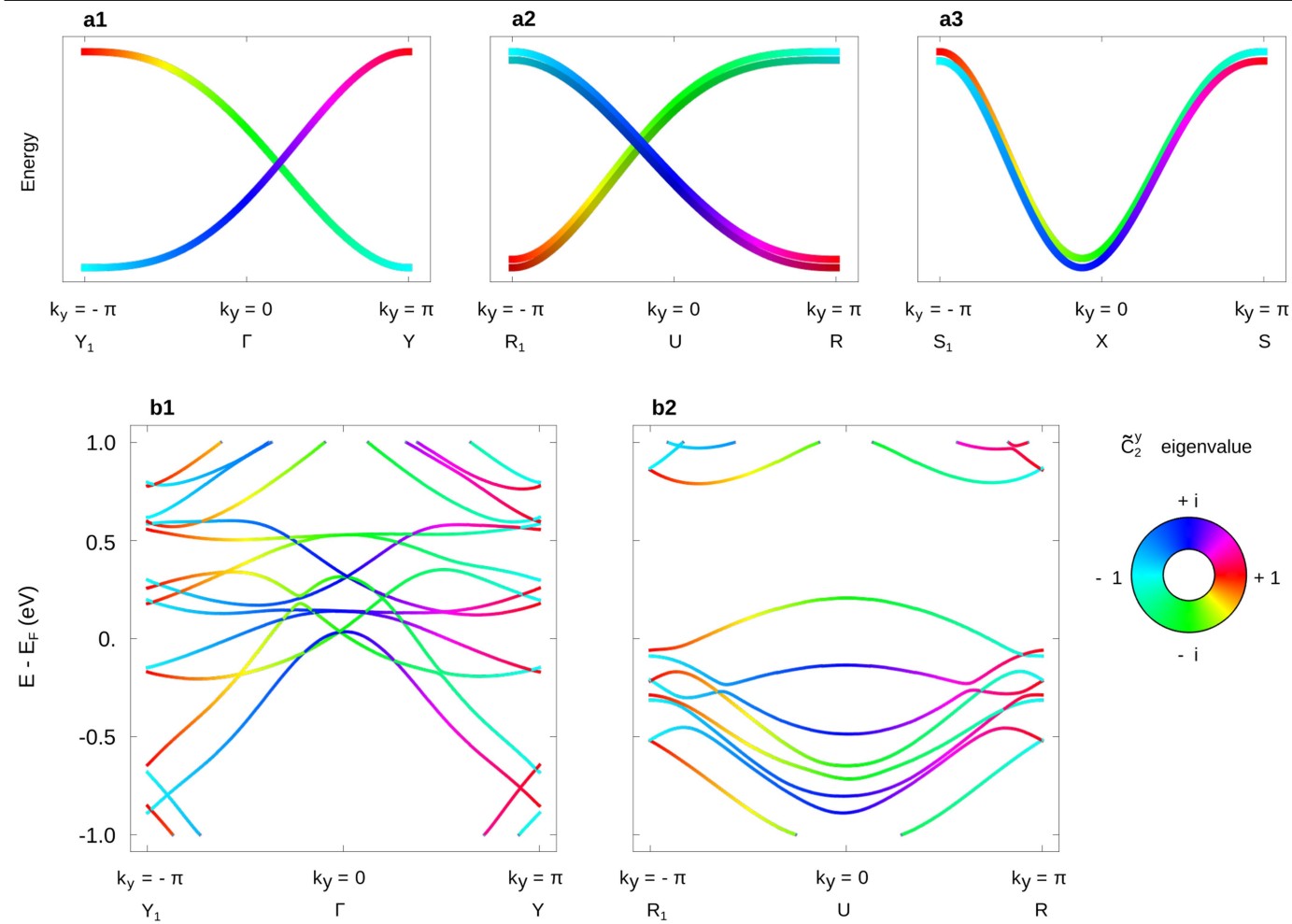

**Extended Data Fig. 2 | Momentum dependence of the screw-rotation eigenvalues. a**, Schematic band connectivity diagrams for a minimal set of bands along the $Y_1$–Γ–Y line (**a1**), the $R_1$–U–R line (**a2**) and the $S_1$–X–S (**a3**) line, respectively. The eigenvalues of the screw-rotation symmetry $\widetilde{C}_2^y$ are indicated by colour. **b**, Ab initio electronic band structure of MnSi in the [010] FM phase with the $\widetilde{C}_2^y$ eigenvalues indicated by colour. The crossings of bands with different colour on the $Y_1$–Γ–Y line (**b1**) and on the $R_1$–U–R line (**b2**) are Weyl points and four-fold degenerate points, respectively.

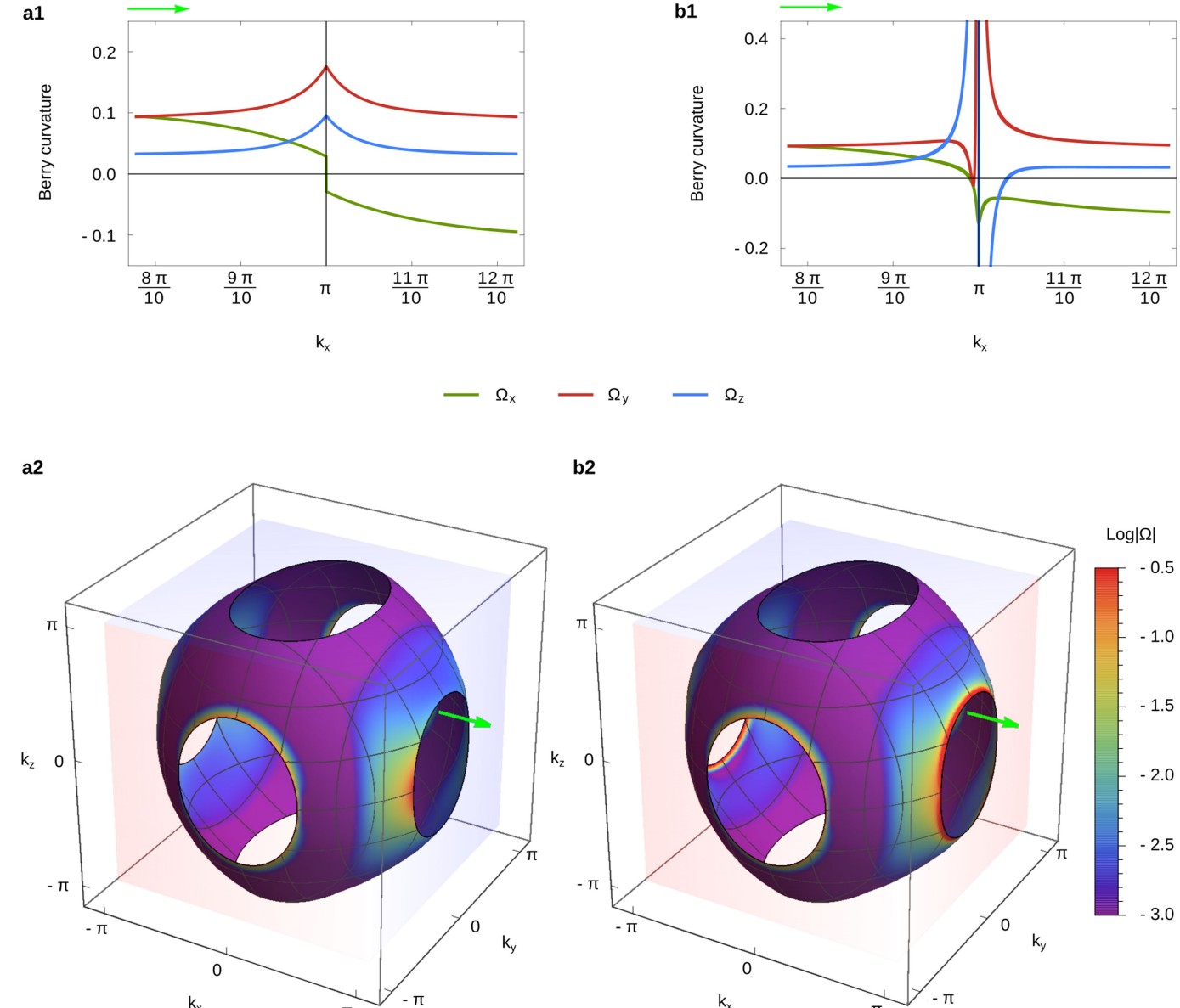

**Extended Data Fig. 3 | Berry curvature on the Fermi surface. a,** Berry curvature $\Omega_\mu(\mathbf{k})$ on one of the Fermi surfaces of a tight-binding model in SG 19.27, corresponding to ferromagnetic MnSi with the magnetization pointing along [010]. **a1** shows the three components of $\Omega_\mu$ as a function of $k_x$, along the direction indicated by the green arrow in **a2**. The absolute value of the Berry curvature $|\mathbf{\Omega}(\mathbf{k})|$ is indicated in **a2** by a logarithmic colour code. **b,** Same as **a** but for a tight-binding model in SG 4.9, corresponding to ferromagnetic MnSi with the magnetization rotated into the $x$–$y$ plane.

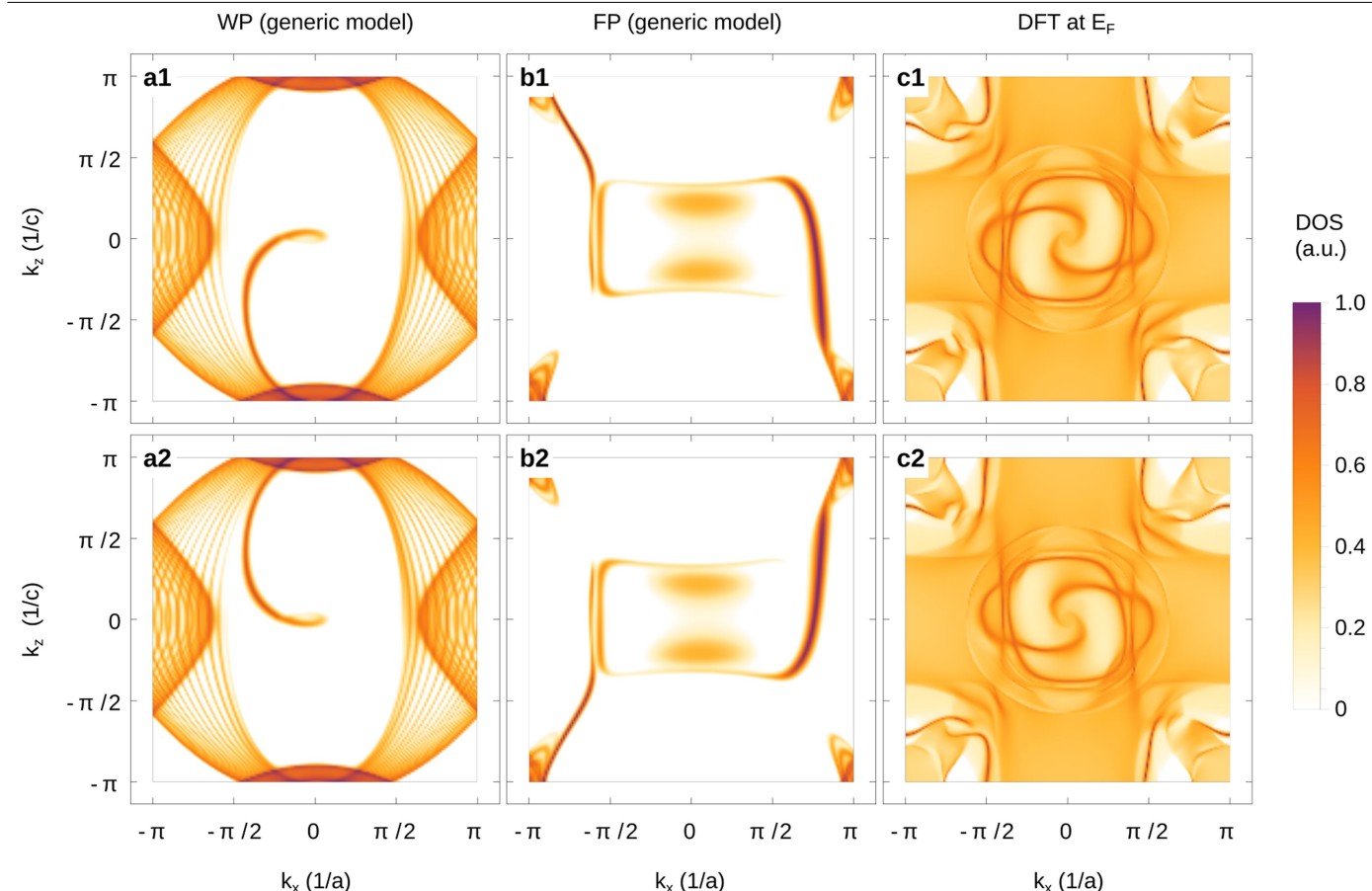

**Extended Data Fig. 4 | Topological surface states. a–c**, Density of states (DOS) at the (010) surface of the tight-binding model with SG 19.27 (**a**, **b**) and ferromagnetic MnSi with the magnetization aligned along [010] (**c**). The first and second rows display the DOS at the top and bottom surfaces, respectively. In **a**, the surface DOS is shown at an energy $E = -1.2$ of the single Weyl point (WP) on the $Y_1$–Γ–Y line. A single Fermi arc emanates from the projected Weyl point and connects to the $k_z = \pi$ NP. In **b**, the surface DOS is shown at the energy $E = +1.4$ of the four-fold point (FP) on the $R_1$–U–R line, whose chirality $\nu = -2$ is compensated by two accidental Weyl points in the bulk. Two Fermi arcs emanate from the projected four-fold point and connect to the accidental Weyl points in the bulk. In **c**, the DFT-derived surface DOS of ferromagnetic MnSi is shown at the Fermi level $E = E_F$. Fermi arcs emanate from the projected Weyl points on the $Y_1$–Γ–Y line and connect with the bulk bands forming NPs on the BZ boundaries.

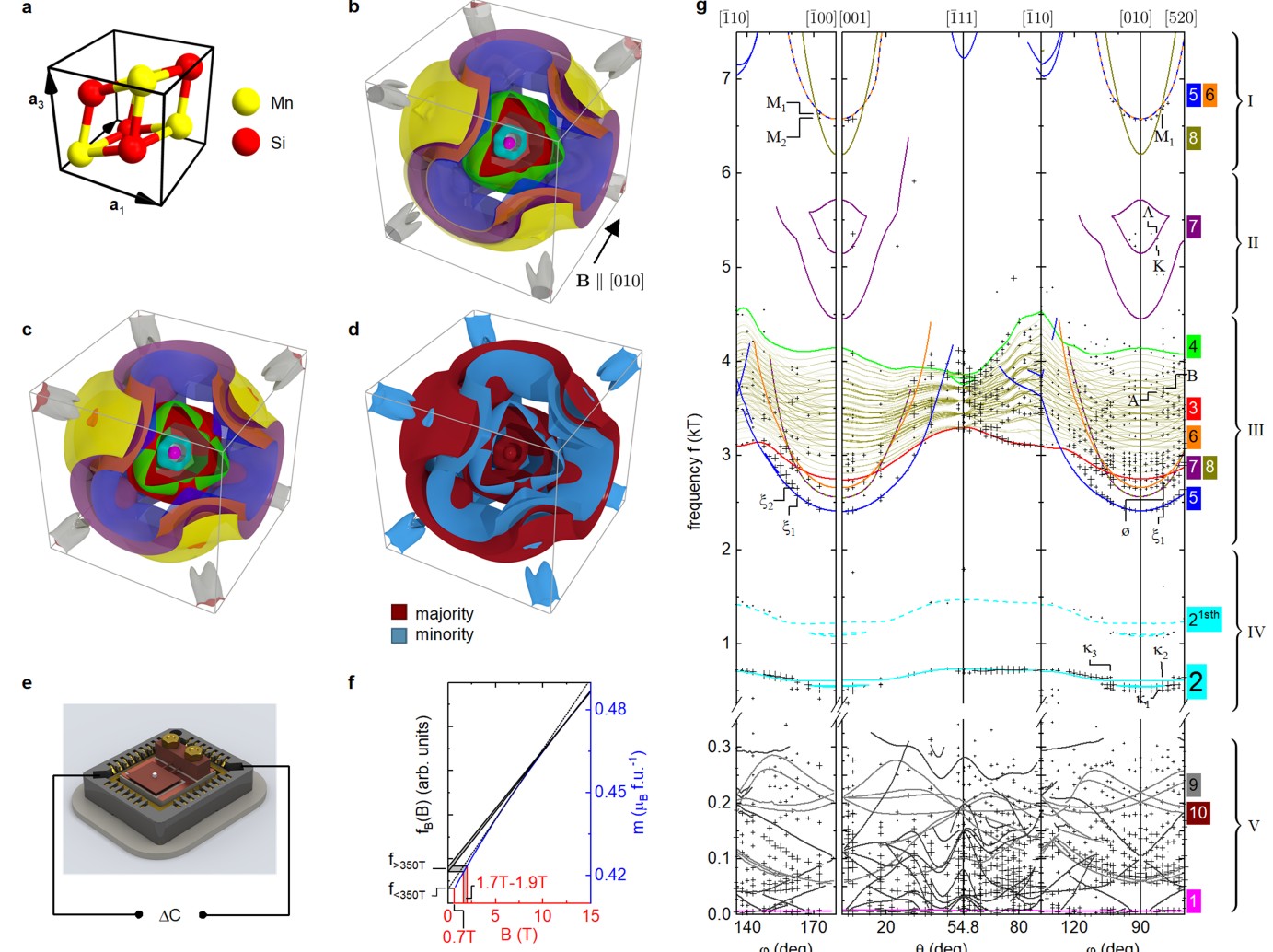

**Extended Data Fig. 5 | Crystal structure, calculated Fermi surfaces, experimental methods, and dHvA spectra in the (001) and ($\bar{1}$10) planes.**
**a**, Crystal structure of MnSi. **b**, Fermi surface as calculated within local spin density approximation without rigid band shifts. **c**, Calculated Fermi surface neglecting spin–orbit coupling. **d**, Calculated Fermi surface neglecting spin–orbit coupling and highlighting majority and minority spin. **e**, Sketch of the cantilever magnetometer chip with capacitive readout. **f**, Magnetic field

dependence of the frequency $f_B(B)$ tracking the magnetic-field dependence of the unsaturated magnetization in the field-polarized phase. The frequency $f(B)$ observed corresponds to the zero-field intercept of the tangent to $f_B(B)$.
**g**, Experimental dHvA frequency branches (crosses) for rotation in the (001) and ($\bar{1}$10) planes together with the theory (lines) matched to the experiment. See Supplementary Note 5 for details.

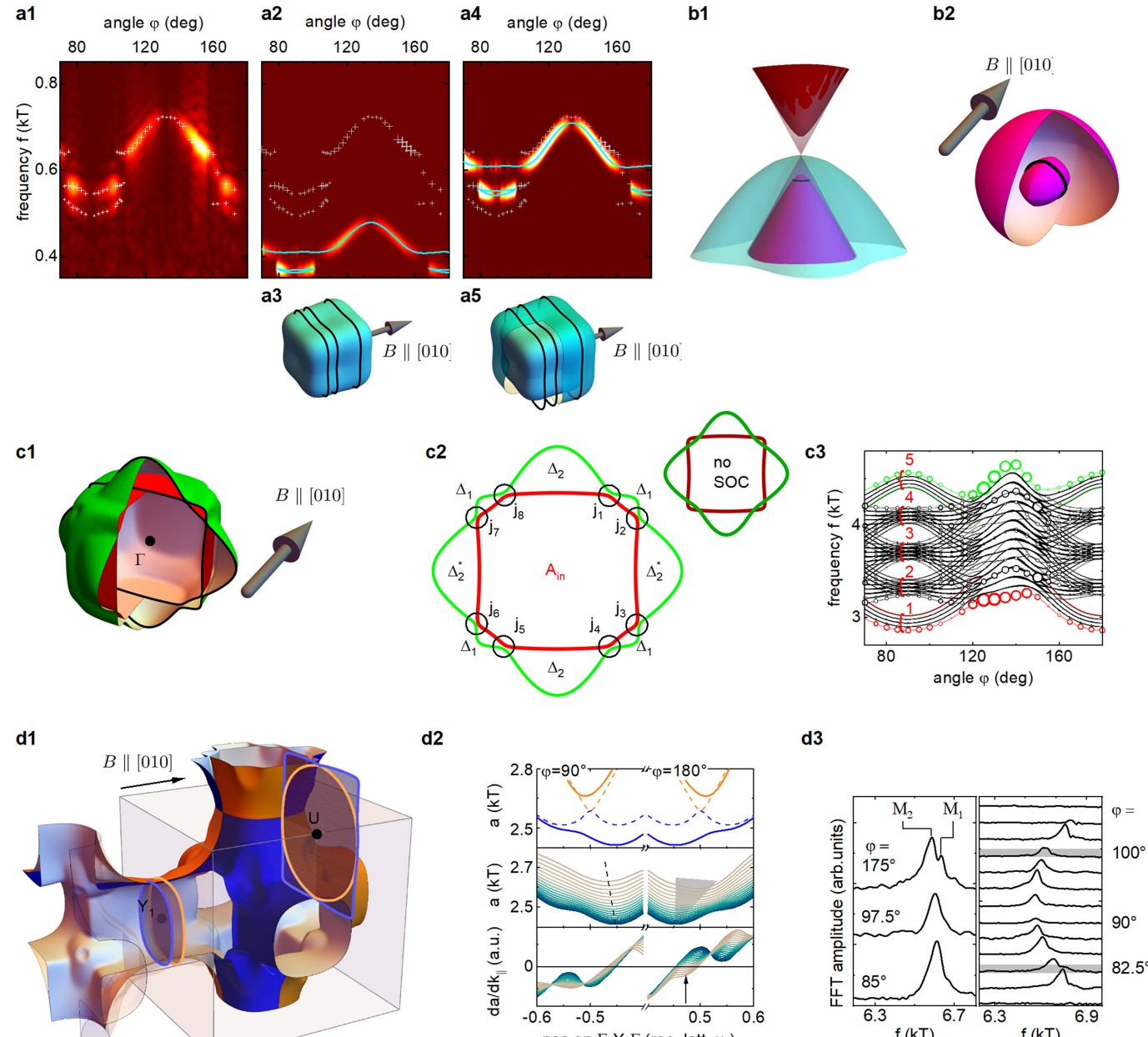

**Extended Data Fig. 6 | Details of the assignment of experimental dHvA orbits to FS sheets 1 to 6. a1**, Experimental signature of sheet 2. Colour scale corresponds to experimental FFT amplitude and crosses show positions of maxima. **a2**, Torque signal predicted from DFT as calculated. Lines show theoretical branches, crosses show experimental positions. **a3**, FS sheet 2 as calculated. Three extremal orbits 2Γ and 2ΓY(1,2) are present for **B** close to [010], which are assigned to κ$_{1,2,3}$. **a4**, Calculated dHvA branch including a small upward band shift of 20 meV, yielding a good match with experiment (crosses). **a5**, Comparison of as-calculated (inner) and matched (outer) FS sheet 2. **b1**, Dispersions of bands 1, 2 and 3 in the $k_x$–$k_z$ plane without (transparent) and with spin–orbit coupling (solid), showing a spin-1 excitation-like three-fold degeneracy that is lifted by spin–orbit coupling. Since band 2 (cyan) crosses the Fermi level, the α branch must originate from band 1 and not band 3. The Fermi level matching the experimental frequency is shown in black. **b2**, FS sheet 1 as calculated (outer) and matched to experiment (inner). The α branch is assigned to orbit 1Γ. **c1**, FS sheets 3 and 4 exhibit extremal orbits with 8 breakdown junctions j$_1$ to j$_8$ for **B** close to [010] as shown in **c2**. The inset shows the two extremal orbits that arise when spin–orbit coupling is neglected. **c3**, 256 breakdown orbit branches originating from sheets 3 and 4. Symbol size reflects

orbit probability. The torque amplitude is not considered in this graph. The breakdown orbits group into five sets labelled in red. The branches ρ and H are assigned to the inner and outer orbits 3Γ and 4Γ, respectively. **d1**, FS sheets 5 and 6 as in Fig. 4a with two neck orbits 5ΓY and 6ΓY and the loop orbits 5U6U assigned to (ξ$_1$, ξ$_2$), π and M$_1$, respectively. **d2**, Top: neck cross-sectional areas of sheet 5 (blue) and 6 (orange) versus $k_∥$ neglecting (dashed) and including (solid) spin–orbit coupling for $φ = 90°$ and $φ = 180°$. Middle: cross-sectional area $a$ of sheet 5 versus $k_∥$ for field directions 70°–90° and 160°–180°. Dashed line: position of single extremal area around $φ = 90°$. Shaded grey area: neck being on the verge of developing a second minimum close to 180° but not around 90° that could give rise to ξ$_2$. Bottom: derivative d$a$/d$k_∥$, where zero-crossings correspond to extremal orbits. **d3**, FFT amplitude of the loop orbits around $φ = 90°$ and $φ = 180°$. Left: a distinct splitting of the M$_1$ branch into M$_1$ and M$_2$ is observed close to $φ = 180°$ but not around $φ = 90°$. Right: the FFT amplitude of the M$_1$ branch shows unexpected secondary minima (shaded areas) close to $φ = 90°$ on both sides. Both effects may be connected to either the quasi-degeneracy of the U5U6 orbits shown in Fig. 4b3, b4 or to a crossing with the 8ΓY branch.

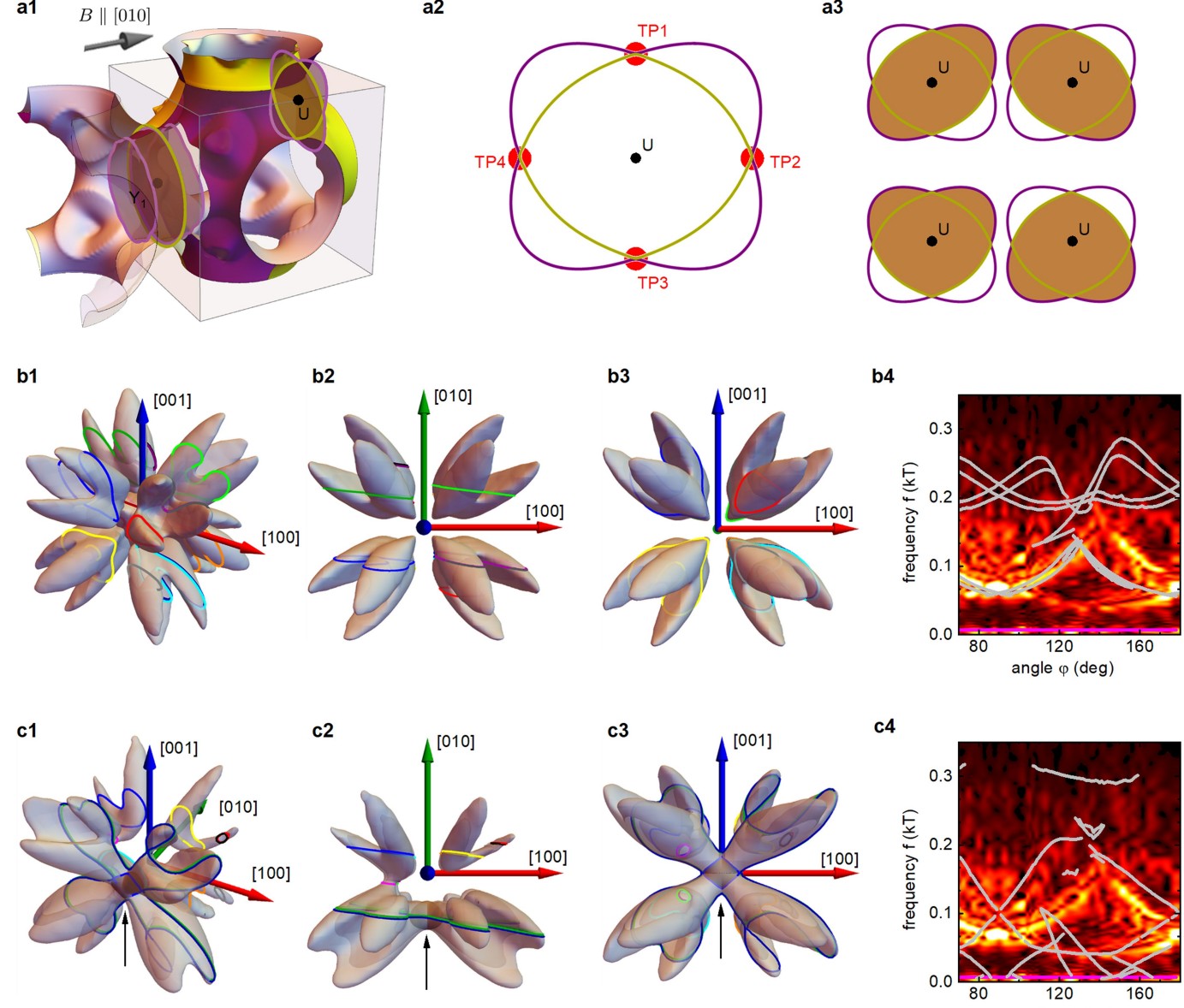

**Extended Data Fig. 7 | Assignment of experimental dHvA orbits to FS sheets 7 to 10. a1**, Along Γ–Y₁–Γ three neck orbits on sheet 7 (purple), one neck orbit on sheet 8 (yellow), and two loop orbits around U are predicted for **B**∥[010]. **a2**, Loop orbits are shared between the sheets at TP1 to TP4 in analogy to the sheet pair (5, 6). **a3**, For **B**∥[010], the upper two lens-shaped orbits exist, while for **B** in the (001) plane away from [010] the lower two heart-shaped orbits are allowed in addition. **b**, Sheet 9 neglecting spin–orbit coupling in perspective (**b1**), top (**b2**) and back (**b3**) views for $\varphi$ = 83°, that is, **B** slightly off the [010] direction. Bands 9 and 10 are shifted upward by 10 meV for an optimal match to the low frequencies as shown in **b4**, where grey lines correspond to the

calculations. In total, 15 orbits are predicted for this specific field direction alone. Without spin–orbit coupling, band 10 does not cross the Fermi level for this shift. **c1–c3**, Sheets 9 and 10 and predicted dHvA orbits including spin–orbit coupling for $\varphi$ = 83° and a shift of 11 meV yielding a good match as shown in **c4**. Sheet 10 resides inside sheet 9 as highlighted by black arrows. It occurs only for field directions where two or more 'banana bunches' cross the BZ surface and connect. In the situation depicted here, (001) is an NP, thereby connecting parts of sheets 9 and 10 in such a way that extremal orbits cross from one sheet to the other.

## Extended Data Table 1 | Key properties of the FS sheets

| sheet no. | topology | location | carrier type e/h | spin character | $f(B)$-shift |
|---|---|---|---|---|---|
| 1 | pocket | $\Gamma$-centered | h | majority | ↘ |
| 2 | pocket | $\Gamma$-centered | h | majority | ↘ |
| 3 | pocket | $\Gamma$-centered | h | mixed | - |
| 4 | pocket | $\Gamma$-centered | h | mixed | - |
| 5 | jungle-gym | - | necks: h | minority | necks: ↗ |
|   |           |   | loops: e | minority | loops: ↘ |
| 6 | jungle-gym | - | necks: h | minority | necks: ↗ |
|   |           |   | loops: e | minority | loops: ↘ |
| 7 | jungle-gym | - | necks: h | majority | necks: ↘ |
|   |           |   | loops: e | majority | loops: ↗ |
| 8 | jungle-gym | - | necks: h | majority | necks: ↘ |
|   |           |   | loops: e | majority | loops: ↗ |
| 9 | pockets | $\Gamma$-$R$ | e | minority | ↗ |
| 10 | pocket/none | $R$ | h | minority | ↘ |

| sheet no. | $D(E_F)$ (states/(eV u.c.)) | $E_F$-shift (mJ/(mol K$^2$)) | $D(E_F + E_F\text{-shift})$ | $\gamma$ (mJ/(mol K$^2$)) | $m^*/m_b$ | $\gamma^*$ |
|---|---|---|---|---|---|---|
| 1 | 0.011 | 27 | 0.001 | $10^{-4}$ | ~ 5 | 0.004 |
| 2 | 0.084 | −20 | 0.11 | 0.07 | 5.9 | 0.41 |
| 3 | 0.36 | 8.5 | 0.36 | 0.21 | 7.3 | 1.53 |
| 4 | 0.62 | 9.5 | 0.62 | 0.36 | 5.0 | 1.80 |
| 5 | 1.66 | 4 | 1.66 | 0.98 | 5.5 | 5.39 |
| 6 | 1.74 | 4 | 1.74 | 1.03 | 5.6 | 5.77 |
| 7 | 2.44 | −4 | 2.47 | 1.46 | 5 | 7.3 |
| 8 | 1.74 | −4 | 1.77 | 1.04 | ~ 5 | 5.2 |
| 9 | 1.09 | −11 | 0.84 | 0.5 | ~ 1.5 | 0.75 |
| 10 | 0.36 | −11 | 0 − 0.08 | 0 − 0.05 | - | - |
| **sum** | **10.1** | | **9.65** | **5.65** | **5.1** | **28.15** |
| | | | | specific heat experiment[32]: | | **28** |

Information as calculated and matched to experiment. Top: sheet number, topology and location, carrier type and spin refer to the dominant properties of carriers on the corresponding dHvA orbits. There are sheets with both strongly mixed electron/hole and mixed spin character. The column labelled $f(B)$-shift states the direction of the expected frequency shift with increasing magnetic field. Bottom: $D(E_F)$ states the density of states as calculated. $E_F$-shift states the shift of the Fermi level used to achieve an optimal match to experiment. $D(E_F + E_F\text{-shift})$ states the density of states following the shift of $E_F$. $\gamma$ corresponds to the contribution of the shifted band to the Sommerfeld coefficient. $m^*/m_b$ is the mass enhancement factor, where $m^*$ is determined from the Lifshitz–Kosevich behaviour of the dHvA amplitude and $m_b$ is the bare band mass obtained in DFT. $\gamma^*$ is the Sommerfeld coefficient scaled with the mass enhancement. The Sommerfeld coefficient of 28.15 mJ (mol K$^2$)$^{-1}$ inferred from the dHvA data matches the value of the experimentally determined specific heat at $B = 12$ T within a few per cent, confirming that all thermodynamically important parts of the FS were observed.

**Extended Data Table 2 | Assignment of observed dHvA branches to orbits on the calculated FS**

| Branch | Orbit | $f_{\text{exp.}}$ [kT] | $f_{\text{pred.}}$ [kT] | $m^*$ [$m_e$] | $m_b$ [$m_e$] | $\frac{m^*}{m_b}$ | $\varphi$ [deg] |
|---|---|---|---|---|---|---|---|
| $\alpha$ | 1$\Gamma$ | 0.007 $\searrow$ | 0.068 $\searrow$ | 0.4±0.1 | 0.1 | 4 | 82.5 |
| $\beta$ | 9$\Gamma$R(1) | 0.054 | $-\searrow$ | 0.8±0.1 | 0.6 | 1.4 | 82.5 |
| $\gamma$ | 9$\Gamma$R(2) | 0.070 | $-\searrow$ | 2.7±0.3 | 1.9 | 1.4 | 82.5 |
| $\delta$ | 9$\Gamma$R(3) | 0.082 | $-\searrow$ | 2.3±0.6 | 1.9 | 1.2 | 82.5 |
| $\epsilon$ | 9$\Gamma$R(4) | 0.095 | $-\searrow$ | 2.0±0.5 | – | – | 82.5 |
| $\zeta$ | 9$\Gamma$R(5) | 0.110 | $-\searrow$ | 2.4±0.4 | – | – | 82.5 |
| $\eta$ | 9$\Gamma$R(6) | 0.141 | $-\searrow$ | 2.5±0.5 | – | – | 82.5 |
| $\mu$ | 9$\Gamma$R(7) | 0.130 | $-\searrow$ | – | 1.9 | – | 152.5 |
| $\theta$ | 9$\Gamma$R10R(1) | 0.225 | $-\searrow$ | 3.5±0.6 | – | – | 82.5 |
| $\iota$ | 9$\Gamma$R10R(2) | 0.248 | $-\searrow$ | 5.4±0.6 | – | – | 82.5 |
| $\tilde{\iota}$ | 9$\Gamma$R10R(3)) | 0.290 | $-\searrow$ | 5.4±0.6 | ~ 2.0 | ~ 2.7 | 82.5 |
| $\kappa_1$ | 2$\Gamma$ | 0.488 $\searrow$ (0.523) | 0.369 $\searrow$ | 6.3±0.6 | 1.1 | 6.0 | 82.5 (106) |
| $\kappa_2$ | 2$\Gamma$Y(1) | 0.566 $\searrow$ (0.564) | 0.371 $\searrow$ | 6.2±0.1 | 1.1 | 5.9 | 82.5 (106) |
| $\kappa_3$ | 2$\Gamma$Y(2) | 0.641 $\searrow$ | 0.411 $\searrow$ | 6.5±0.5 | 1.2 | 5.6 | 106 |
| 2$\kappa_1$ | 2$\kappa_1$ | 1.065 | 2$f_{\kappa_1}$ | 14.2±0.8 | 2$m_{\kappa_1}$ | – | 82.5 |
| 2$\kappa_2$ | 2$\kappa_2$ | 1.120 | 2$f_{\kappa_2}$ | 14.0±0.6 | 2$m_{\kappa_2}$ | – | 82.5 |
| 3$\kappa_2$ | 3$\kappa_2$ | 1.610 | 3$f_{\kappa_2}$ | 16±6 | 3$m_{\kappa_2}$ | – | 82.5 |
| $\xi_1$ | 5$\Gamma$Y(1) | 2.459 $\nearrow$ (2.576) | 2.532 $\nearrow$ | 10.3±0.1 | 2.0 | 5.4 | 82.5 (165) |
| $\xi_2$ | 5$\Gamma$Y(2) | 2.653 | – | – | – | – | 165 |
| $ø'$ | 7U8U | 2.658 | 2.765 $\nearrow$ | 10.0±0.3 | 2.0 | 5.0 | 82.5 |
| $\pi$ | 6$\Gamma$Y | 2.701 $\nearrow$ | 2.822 $\nearrow$ | 11.1±0.3 | 2.0 | 5.6 | 82.5 |
| $\rho$ | 3$\Gamma$ | 2.786 $\searrow$ | 2.891 $\rightarrow$ | 10.9±0.4 | 1.5 | 7.1 | 82.5 |
| $\rho'$ | 3$\Gamma$4$\Gamma$(1) | 2.833 | 2.934 $\rightarrow$ | – | 1.5 | – | 82.5 |
| $\sigma$ | 3$\Gamma$4$\Gamma$(2) | 2.879 | 2.976 $\rightarrow$ | 11.2±0.4 | 1.5 | 7.5 | 82.5 |
| $\sigma'$ | 3$\Gamma$4$\Gamma$(3) | 2.918 | 3.021 $\rightarrow$ | – | – | – | 82.5 |
| $\tau$ | 3$\Gamma$4$\Gamma$(4) | 2.966 | 3.019 $\rightarrow$ | 10.2±0.4 | 1.5 | 6.8 | 82.5 |
| $\upsilon$ | 3$\Gamma$4$\Gamma$(5) | 3.034 $\searrow$ | 3.061 $\searrow$ | 8.7±0.3 | 1.5 | 5.9 | 82.5 |
| $\upsilon'$ | 3$\Gamma$4$\Gamma$(6) | 3.105 | 3.231 $\rightarrow$ | – | – | – | 82.5 |
| $\varphi$ | 3$\Gamma$4$\Gamma$(7) | 3.229 | 3.453 $\rightarrow$ | 13.2±0.4 | – | – | 82.5 |
| $\chi'$ | 3$\Gamma$4$\Gamma$(8) | 3.350 | 3.583 $\rightarrow$ | – | – | – | 82.5 |
| $\psi$ | 3$\Gamma$4$\Gamma$(9) | 3.450 | 3.715 $\rightarrow$ | 11.6±0.6 | – | – | 82.5 |
| $\omega$ | 3$\Gamma$4$\Gamma$(10) | 3.485 $\nearrow$ | 3.626 $\rightarrow$ | 11.6±0.6 | – | – | 82.5 |
| $A$ | 3$\Gamma$4$\Gamma$(11) | 3.671 | 3.931 $\rightarrow$ | 13.6±0.1 | – | – | 82.5 |
| $B$ | 3$\Gamma$4$\Gamma$(12) | 3.717 $\searrow$ | 4.017 $\rightarrow$ | 13.7±0.3 | – | – | 82.5 |
| $\Gamma'$ | 3$\Gamma$4$\Gamma$(13) | 3.840 | 4.323 $\nearrow$ | – | 3.2 | – | 82.5 |
| $\Delta$ | 3$\Gamma$4$\Gamma$(14) | 3.940 | 4.366 $\rightarrow$ | 14±1 | 3.2 | 4.4 | 82.5 |
| $E$ | 3$\Gamma$4$\Gamma$(15) | 4.040 | 4.409 $\rightarrow$ | 17±4 | 3.2 | 5.5 | 82.5 |
| $Z$ | 3$\Gamma$4$\Gamma$(16) | 4.120 | 4.451 $\rightarrow$ | 15±3 | 3.2 | 4.7 | 82.5 |
| $H$ | 4$\Gamma$ | 4.180 | 4.493 $\rightarrow$ | 16±5 | 3.1 | 5.1 | 82.5 |
| $\Theta$ | 7$\Gamma$Y(1) | 4.350 | 4.569 | 15±5 | 4.0 | 3.8 | 82.5 |
| 2$\xi_1$ | 2$\xi_1$ | 4.920 | 2$f_{\xi_1}$ $\searrow$ | 24±5 | 2$m_{\xi_1}$ | – | 82.5 |
| $K$ | 7$\Gamma$Y(2) | 5.304 | 5.179 $\searrow$ | – | 4.2 | – | 85 |
| $\Lambda$ | 7Y | 5.304 | 5.481 $\searrow$ | – | 3.4 | – | 85 |
| $M_1$ | 5U6U | 6.715 $\searrow$ (6.634) | 6.627 $\searrow$ | 15.1±0.2 | ~2.8 | 5.4 | 82.5 (175) |
| $M_2$ | 5U6U | 6.587 | – | – | – | – | 175 |
| – | 8$\Gamma$Y | – | 6.610 $\searrow$ | – | 4.0 | – | – |

Arrows denote the direction of the frequency shift with increasing magnitude of $B$. Error bars of effective masses reflect the standard deviation of the Lifshitz–Kosevich fits. Peaks at frequencies marked with a prime were only observed in magnetic field sweeps up to 16 T. Frequency values in brackets are given at a second angle $\varphi$ in the (001) plane measured from the [100] direction.

## Extended Data Table 3 | Catalogue of space groups with symmetry-enforced NPs

| | | | | | | | | |
|---|---|---|---|---|---|---|---|---|
| 4.8 [t] | 17.8 [t] | 18.17 [t] | 19.26 [T] | 20.32 [t] | 26.67 | 29.100 | 31.124 | 33.145 |
| 36.173 | 76.8 [t] | 78.20 [t] | 90.96 [t] | 91.104 [t] | 92.112 [T] | 94.128 [T] | 95.136 [t] | 96.144 [T] |
| 113.268 | 114.276 | 169.114 [t] | 170.118 [t] | 173.130 [t] | 178.156 [t] | 179.162 [t] | 182.180 [t] | 185.198 |
| 186.204 | 198.10 [T] | 212.60 [T] | 213.64 [T] | | | | | |

| | | | | | | | | |
|---|---|---|---|---|---|---|---|---|
| 4.9 [t] | 11.54 | 14.79 | 17.10 [t] | 18.18 [t] | 18.19 [t] | 19.27 [T] | 20.34 [t] | 26.68 |
| 26.69 | 29.101 | 29.102 | 31.125 | 31.126 | 33.146 | 33.147 | 36.174 | 36.175 |
| 51.294 | 51.296 | 52.310 | 52.311 | 53.327 | 53.328 | 54.342 | 54.344 | 55.357 |
| 55.358 | 56.369 | 56.370 | 57.382 | 57.383 | 57.384 | 58.397 | 58.398 | 59.409 |
| 59.410 | 60.422 | 60.423 | 60.424 | 61.436 | 62.446 | 62.447 | 62.448 | 63.463 |
| 63.464 | 64.475 | 64.476 | 90.98 [t] | 90.99 [t] | 92.114 [T] | 92.115 [T] | 94.130 [T] | 94.131 [t] |
| 96.146 [T] | 96.147 [T] | 113.269 | 113.271 [t] | 114.277 | 114.279 [t] | 127.390 | 127.393 | 128.402 |
| 128.405 | 129.414 | 129.417 | 130.426 | 130.429 | 135.486 | 135.489 | 136.498 | 136.501 |
| 137.510 | 137.513 | 138.522 | 138.525 | 169.115 [t] | 170.119 [t] | 173.131 [t] | 176.147 | 178.157 [t] |
| 178.158 [t] | 179.163 [t] | 179.164 [t] | 182.181 [t] | 182.182 [t] | 185.199 | 185.200 | 186.205 | 186.206 |
| 193.258 | 193.259 | 194.268 | 194.269 | | | | | |

| | | | | | | | | |
|---|---|---|---|---|---|---|---|---|
| 3.5 [t] | 3.6 [t] | 4.10 [t] | 16.4 [t] | 16.5 [t] | 16.6 [T] | 17.11 [t] | 17.13 [t] | 17.14 [T] |
| 17.15 [T] | 18.20 [t] | 18.21 [T] | 18.22 [T] | 18.24 [t] | 19.28 [T] | 19.29 [t] | 20.36 [t] | 21.42 [t] |
| 21.44 [t] | 25.61 | 25.64 | 25.65 | 26.71 | 26.72 | 26.76 | 27.82 | 27.85 |
| 27.86 | 28.94 | 28.95 | 28.96 | 28.98 | 29.104 | 29.105 | 29.109 | 30.118 |
| 30.119 | 30.120 | 30.122 | 31.128 | 31.129 | 31.133 | 32.139 | 32.142 | 32.143 |
| 33.149 | 33.150 | 33.154 | 34.161 | 34.162 | 34.164 | 35.169 | 35.171 | 36.178 |
| 37.184 | 37.186 | 75.4 [t] | 75.6 [t] | 76.11 [t] | 77.16 [t] | 77.18 [t] | 78.23 [t] | 81.36 [t] |
| 81.38 [t] | 89.92 [t] | 89.93 [t] | 89.94 [T] | 90.100 [T] | 90.102 [t] | 91.109 [T] | 91.110 [T] | 92.116 [T] |
| 92.117 [t] | 93.124 [t] | 93.125 [T] | 93.126 [T] | 94.132 [T] | 94.134 [t] | 95.141 [T] | 95.142 [T] | 96.148 [T] |
| 96.149 [t] | 99.168 | 99.170 | 100.176 | 100.178 | 101.184 | 101.186 | 102.192 | 102.194 |
| 103.200 | 103.202 | 104.208 | 104.210 | 105.216 | 105.218 | 106.224 | 106.226 | 111.256 |
| 111.257 | 111.258 | 112.264 | 112.265 | 112.266 | 113.272 | 113.274 | 114.280 | 114.282 |
| 115.288 | 115.290 | 116.296 | 116.298 | 117.304 | 117.306 | 118.312 | 118.314 | 168.112 [t] |
| 171.124 [t] | 172.128 [t] | 177.154 [t] | 180.172 [t] | 181.178 [t] | 183.190 | 184.196 | 195.3 [T] | 207.43 [T] |
| 208.47 [T] | 215.73 | 218.84 | | | | | | |

Table listing all magnetic SGs with symmetry-enforced NPs. The list is grouped into three blocks: 32 SGs with time-reversal symmetry (describing non-magnetic materials), 94 SGs without time-reversal symmetry (describing ferro- or ferrimagnets), and 129 SGs with a symmetry that combines time-reversal symmetry with a translation (describing antiferromagnets). For the NPs to have non-zero topological charge, the SG must be chiral (labelled by '[t]' or '[T]'). The 33 SGs labelled by '[T]' have NPs whose topological charge is enforced to be non-zero by symmetry, as discussed in Supplementary Note 3.