## [Peer Review File · Nature]

Manuscript Title: Symmetry-enforced topological nodal planes at the Fermi surface of a 2 magnet

Editorial Notes:

Redactions – Mention of other journals

This document only contains reviewer comments, rebuttal and decision letters for versions considered at *Nature*. Mentions of the other journal have been redacted.

Reviewer Comments & Author Rebuttals

Reviewer Reports on the Initial Version:

Ref #1

In this work, the authors study the symmetry enforced nodal planes of the field-polarized ferromagnetic MnSi by comparative mapping of the extremal cross-sections of Fermi Surface between theoretical and experimental dHvA spectroscopy, and generalize the discussion of topological nodal planes to all 1651 magnetic space groups. The theoretical analysis is very detailed, and makes a catalog of magnetic space group with symmetry-enforced nodal planes. In this way, the present work provided a comprehensive knowledge for theoreticians and experimentalists to explore the topological nodal planes in magnetic materials. However, for the experimental aspect, there are a mass of works need to do, so that make the conclusion more solid, reveal the topological properties of the nodal planes more visible, and demonstrate the control ability of the magnetic field on the topological nodal planes. Below are my comments and suggestions that might be clarified before its publication.

Comments:

1. Topological nodal plane is not a new concept, which has been widely studied and experimental observed in previous works [Xiao M, et al. arXiv:1709.02363 (2017), Chang, G., et al. *Nature Mater* 17, 978 (2018), Yang, Y., et al. *Nat Comm.* 10, 5185 (2019), Xiao M, et al. *Science advances*, 6, 8 (2020)]. The main innovation of this work is to generalize this concept into the magnetic materials MnSi, and try to control its physical properties by the external magnetic field. However, MnSi seems not a suitable material to study the underlying physics of topological nodal planes for the following reason.

First, one important criterion for topological nodal planes is that the materials with appropriate electron filling. As the authors elaborated in the supplementary S1.B, the magnetic space group SG 19.27 (P212'12'1) has a minimal insulating filling of $\nu \in 4Z$. The topological nodal planes are only formed between bands $n = 2m - 1$ and $n = 2m$, where m is an integer. However, this is not case in ferromagnetic MnSi, where both Mn and Si atoms occupy 4a Wyckoff positions, indicating that ferromagnetic MnSi possess $4Z$ number of electrons filling per primitive cell. Thus, MnSi is not a typical compound with "filling-enforced" topological nodal planes. Though the Fermi Surface cross nodal planes as shown in Fig1.e and Fig1.f, detailed discussion of the difference between the typical topological nodal planes and the nodal planes in MnSi should be supported both from theoretical and experimental results.

Second, the Fermi surfaces in MnSi are too complicated, which make the existence of the fermi arcs originated from nodal planes is unambiguous even in the authors' calculations. As theoretical calculated surface states shown in Fig E4, the Fermi arcs will merge into the bulk states totally, even they really exist.

Thirdly, the complex FSs in MnSi may obliterate the properties of nodal planes. To figure out this problem, I strongly suggest the authors to compare the transport measurements when magnetic field applied along (010) and (110) and the other direction. If the magnetic-controlled

electromagnetic response, the anomalous Hall effect can be observed, it will make this work more valuable.

2. The experimental identification of the nodal planes on the FS is indirect. The ferromagnetic phase of MnSi possess complex FS, in which sheets 5-10 are formed by nodal planes from DFT calculations. To identification of the complex FS sheet from experimental dHvA spectra, the authors systematically compared with theoretical results as shown in Fig3. The quantitatively mapping is not enough to confirm the FS sheets of experimental results.

First, there is not experimental spectra in sheet 7 and sheet 8, which form TPs with larger cross-sections than sheet 5 and sheet 6, the authors simply explained as too weak to observe it, then what's the exactly value of it?

Second, there is another obvious orbital except sheet 2 in region IV in experimental data, while there is not emerging in theoretical results, can authors provide more detail information of this frequency?

Thirdly, the dHvA spectra in region V is fuzzy, which is hardly resolve the FS sheet 9 and 10. As a completely experimental comparison, can the authors add more theoretical dHvA spectra to Fig E5.g? For these results, it's tough to convince readers that an unambiguous identification of the FS sheets was achieved. It is therefore extremely important to clarify experimental frequencies before a decisive conclusion can be drawn. It would be nice to improve the resolution for lower temperature, as the dHvA oscillations of torque were resolved in temperature $T = 35$ mK.

Additional questions:

1. For the magnetic moments oriented along $[010]$, the two nodal planes at the $k_x = \pi$ and $k_z = \pi$ with total topological charge $\nu_{\text{npd}} = 1$ or -1 , which compensated the topological charge of Weyl point along Γ_1 - Γ_5 - Γ_1 , indicating each nodal plane with fractional topological charge? Can the authors give a more detailed theoretical analysis of it? Especially for surface states in the $[010]$, why the chiral Fermi arcs connected to the Γ_5 - k_z direction but not k_x -direction?
2. In supplementary S3.A, the 3 conditions that need to be satisfied for the existence of symmetry-enforced nodal planes are necessary and sufficient condition? As the irreducible co-representations of the magnetic space groups has been posted in Bilbao Crystallographic Server, the analyzed results from authors are all self-consistent with this website?
3. The FS in Fig.1f is enlarged? If it is, please illustrate it. Because the frequency is Fig2.b and Table EII are just few KT, it is puzzling to me that even these frequencies can map to large FS?
4. For the discussion of FFT magnetic torque's spectra in Fig2.b, there have many contiguous frequencies, is there comes from Zeeman splitting? The other question puzzled to me is that some frequencies are marked, while some frequencies are not, can the authors give a detail explanation?
5. The authors deduced the effective mass for each orbital ranging from $m^* = 0.4 m_e$ to $17 m_e$. However, the maximal effective mass is $15.1 m_e$ as shown in Fig2.c.
6. It's hard to follow some interesting results of the current manuscript without references. For example, $\gamma = 30 \text{ mJmol}^{-1}\text{K}^{-2}$ at $B = 9 \text{ T}$ in line 165, the value of experimentally determined specific heat in Table EI, and especially for the methods of Internal magnetic field and dHvA frequency $f(B)$ in a weak itinerant magnet, can authors add relative references in these places?
7. The reference 37 in upper right of $N_d = 1/3$ is misleading.
8. What's the m_b in eq(3) in methods? It does not appear elsewhere in the manuscript.
9. The $B_{\text{av}} = 12 \text{ T}$ in methods should specify the FFT range in $10 \sim 15 \text{ T}$, in order to distinct the $4 \sim 15 \text{ T}$ as demonstrated before.

In summary, the present conclusion in this work is not completely self-consistent. More comprehensive analysis between the theoretical calculations and experimental measurements should be addressed. The innovations did not meet the requirement of Nature. More experimental properties related to the topological nodal planes, especially the electromagnetic response, need to be explored.

Ref #2

The main argument made by the authors in this manuscript is that the magnetic screw rotation symmetry, combined with time reversal symmetry, not only force all bands to cross at the Brillouin zone (BZ) boundaries, forming nodal planes and topological protectorates, but also induce Weyl nodes and 4-fold points within the BZ. The authors demonstrated this picture through combined theoretical and experimental studies on a helimagnetic material MnSi which is well known for showing Skyrmion lattice and a pressure-tuned quantum phase transition. From crystalline symmetry analyses and first principle calculations for this material, the authors first predicted symmetry-enforced nodal planes with topological charges at the BZ boundaries parallel to the magnetization axis. Further, their theoretical analyses also showed the non-trivial topology of these band crossings generates surface Fermi arcs covering large areas of surface BZ. Then the authors attempted demonstrating the predicted topological nodal structure through angular dependent dHvA quantum oscillation measurements. They observed dHvA signatures consistent with the predicted topological orbits. Overall, the work reported in this manuscript is novel and interesting. It can possibly facilitate discoveries of new magnetic topological materials, since the argument of this manuscript can also be extended to 254 magnetic space groups. The authors have indeed found 33 of them have nodal planes with non-zero topological charges. These predictions would inspire further experimental studies on materials with those space groups.

However, this manuscript is relatively weak on its experimental side. The main argument of symmetry-enforced nodal planes at the BZ boundaries is only based on the comparison of the theoretical and experimental cyclotron orbits. Although this is a reasonable approach, the band structure of MnSi is extremely complicated. It encompasses 10 different FSs and its dHvA spectrum consists of 40 frequencies. While the authors showed the correspondence between the experimentally probed bands and the theoretically calculated ones, there are striking discrepancies between experimental and theoretical dHvA spectra (Fig. 3a and 3b), particularly in the frequency regimes of IV and V.

The main experimental proof for the topological protectorates on the nodal planes given by the authors is the consistency between the theoretical and experimental dHvA spectra of the topological orbits which is comprised of different segments of FS sheets 5 and 6 (Fig. 4b1 and 4c). Although this result seems to support the claim of TP and NP, it is not a direct evidence. The complication of the FS, as well as the inconsistency between theoretical and experimental dHvA spectra shown in Fig. 3, puts additional concern on this experimental proof. The most direct way to prove such complicated nodal band crossings is band dispersion measurements by ARPES. I understand ARPES measurements under magnetic fields are impossible at present. Given the nodal planes at the BZ boundaries carry large Berry curvature, it may lead to exotic transport properties such as large intrinsic anomalous Hall effect. Even though this is also not a direct probe for the nodal structure, the consequence of the topological nodal planes can be manifested by such measurements.

In general, quasi-particles excited near the Dirac/Weyl points can be described as emergent relativistic fermions, which are characterized by small/zero effective mass, high mobility and Berry phase in cyclotron motions. Given the author showed MnSi features Weyl nodes, four-fold points and nodal planes, relativistic fermion behavior would be expected in this material. Nevertheless, I noted the effective masses m^* extracted from the temperature dependences of dHvA oscillation amplitudes {Fig. 2c} do not seem consistent with this expectation. m^* is $\sim 15m_e$ for FS sheets (5,6) and (7,8), far larger than that of free electron mass. Other FS sheets also do not show massless behavior though their masses are not as large as $15m_e$.

In summary, the results reported in this manuscript is interesting and it may guide the search for new magnetic topological materials. The weakness is that the authors chose a material with an extremely complicated band structure to demonstrate the concept of symmetry-enforced nodal planes through dHvA measurements, such that the interpretation of the data is not quite

convincing. I think this manuscript can be considered further if the concerns I summarized above can be addressed.

Ref #3

Marc Wilde and co-workers report on the observation of Magnetic-field-controlled topological protectorates of the Fermi surface in the ferromagnetic material MnSi. They employ a combination of theoretical methods, density functional theory and band topology calculations, together with de Haas-van Alphen (dHvA) quantum oscillation measurements to determine the complex Fermi surface of MnSi. It is a well-written manuscript, I congratulate the authors for this thorough piece of work.

The authors state that three key challenges in terms of material research concerning spintronic devices and quantum information technology are still unresolved:

- (a) Identify topological band degeneracies that are generically located at the Fermi level
- (b) the ability to easily control topological degeneracies
- (c) demonstrate the relevance of topological degeneracies in large, multi-sheeted Fermi surfaces

The theory section is very elaborate and highlights the importance of this work also for an experimental condensed matter physicist. The authors, however, need to be more specific what type of novel physics emerges from this paper and the specific study of MnSi that the manuscript can be considered in Nature. Point (a) and (b) and be identified and controlled in materials, where adjacent electron and hole pockets give rise to, for instance, magnetic breakdown phenomena. In other words, the field-tunability is inherent to the specific Fermi surface of the material under study.

I have a few questions concerning the experiments. The authors perform dHvA experiments to determine the Fermi surface and the cyclotron masses at the Fermi energy. The spectrum in Figure 2b looks quite impressive and one does need some imagination to believe that all the observed frequencies revealed in Figure 2a originate from the raw data. I strongly encourage the authors to consider the following comments and questions.

- 1) Technical questions: what is/are the window function(s) used for the FFT analysis? I do see quite some side lobes in the FFT spectra. Do the FFT spectra remain robust using different window functions?
- 2) I advise the authors to show the 35 mK curve and graph where the high-frequency oscillations are better seen (by subtracting the background)
- 3) The authors state that 'FFTs over the range 4 to 15 T (10 T to 15 T) were performed to evaluate frequency components below (above) $f = 350$ T.' Can the authors comment on the error bars of the carrier masses? Did the authors perform a more thorough range analysis to ensure that the carrier masses are not over and/or underestimated?

I agree and think that the authors did a great job in the assignment, see Table EII, of the orbits (comparison theory and experiment). Concerning magnetic breakdown and the different breakdown orbits, I wonder whether the authors could try to provide more experimental evidence concerning their assignment by the comparison of their cyclotron masses with the sum of the individual associated orbits for each breakdown orbit.

- 4) Can the authors provide an insight into the masses of the magnetic breakdown orbits by looking at the sum of the masses of the individual orbits?
- 5) There is a type on page 58: Lidshitz-Kosevich

To conclude, I do not see enough new physics in this manuscript to recommend immediate publication in Nature. This manuscript, taking into account my comments and suggestions, may be suitable for another journal within the nature family ([REDACTED] or [REDACTED]).

Author Rebuttals to Initial Comments:

We wish to thank the referees for their efforts made to review our manuscript. We were very impressed by the detailed reports and the large number of considerate comments, all of which we found very helpful to improve the presentation of our results. Taken together, the referees made us aware that we did not explain sufficiently well some of the main results of our study including certain conventions used to present our quantum oscillatory data.

We have revised the manuscript to better communicate our study. For ease of communication we also prepared a version of the revised manuscript in which the changes are highlighted in color shading. However, due to technical reasons this marked-up version of the manuscript does not highlight the changes in the figure captions and tables, which were minimal, as well the new references we added. In addition, we have prepared a revised version of the animation illustrating the changes of the Fermi surface topology and topological protectorates under changes of field orientation. The referees may find it helpful to take a look at this animation as well.

Before turning to the point-by-point replies given below, it appears most efficient to begin with a summary of the most important aspects.

We feel that the opinions expressed by the referees reflect the present state of the art how to search for and identify topological crossings in the electronic structure of materials, being born out of a long and successful tradition. We wish to emphasize that we propose important changes of perspective where to search for and how to identify topological crossings. In doing so, it is helpful to distinguish three main areas, addressed by the referees:

(A) Conceptual aspects

From a conceptual point of view the identification of topological crossings follows, roughly speaking, three guiding questions. This has shaped the view how to identify topological crossings, but accounts also for the main challenges faced by the field.

(1) Are there topological band crossings anywhere in the electronic structure?

The identification of topological band crossings ultimately requires material-specific, high-precision calculations of the electronic structure. In terms of a classification accidental- and symmetry-enforced topological crossings may be distinguished, where a zoo of zero- and one-dimensional crossings have been identified. In contrast, two-dimensional topological crossings have been mentioned for a few specific settings only. Moreover, to the best of our knowledge, no generic symmetry-enforced mechanism causing topological nodal planes in magnetic systems has been reported in the literature.

In our manuscript we report the identification of a generic mechanism causing two-dimensional topological crossings, notably the simultaneous presence of non-symmorphic plus time-reversal symmetry. Presenting a systematic theoretical assessment of this mechanism for all magnetic space groups, we identify numerous candidate systems (space groups and specific materials including MnSi). Moreover, we emphasize that the symmetry breaking may be used experimentally to switch the topological nodal planes on and off by means of the direction of the magnetization.

(2) Are the topological band crossings located sufficiently close to the Fermi level?

Referred to as filling-enforcement, it is common practice to infer the distance of topological band crossings to the Fermi level from the band filling based on the number of valence

electrons. For zero- and one-dimensional topological crossings the notion of filling-enforcement may be useful in selected cases and rules-of-thumb may be given. However, in practice, filling-enforcement is essentially only realized in special cases and does not represent the generic situation, due to, e.g., correlations, magnetism, electron-phonon interactions, impurity- or self-doping, and further subtleties.

Regarding two-dimensional topological crossings (nodal planes), to the best of our knowledge, there are no rules that guarantee filling-enforcement. In fact, filling-enforced nodal planes require a prohibitive combination of different criteria, as explained in detail in the point-by-point reply. This touches a main message of our manuscript, implicating a major change of perspective. Namely, nodal planes no longer require considerations of filling enforcement. Put differently, we demonstrate theoretically that Fermi surface sheets display a symmetry-enforced pairwise topological crossing where they intersect topological nodal planes. Therefore, these topological crossings are inherently insensitive to any details of the system affecting the precise position of the Fermi level.

(3) What is the relevance of the topological band crossings for the physical properties?

The notion of filling-enforced zero- and one-dimensional topological band crossings has been widely addressed by efforts to identify systems with nominally simple Fermi surfaces. Indeed, an abundance of studies report zero- and one-dimensional band crossings, based on theoretical calculations and measurements of the electronic structure. In contrast, only few studies address the implications of zero- and one-dimensional band crossing for the transport properties as measured in the real materials.

Considering the literature, the coexistence of topologically trivial and non-trivial parts of the Fermi surface represents a major field of research for the future. It is not clear if topologically trivial parts of the Fermi surface simply 'short-circuit' topological crossings on other parts of the Fermi surface or vice versa. There may even be a completely new class of phenomena born out of the interplay of quasiparticle excitations near topologically trivial and non-trivial parts of the Fermi surface. Since all of the Fermi surface sheets crossing topological nodal planes exhibit topological crossings there is, a priori, no need to focus on materials with particularly simple Fermi surfaces.

As for MnSi, a large anomalous Hall effect in excellent quantitative agreement with *ab initio* calculations has been reported long ago, providing direct evidence of large Berry curvatures. This is described in detail in the point-by-point reply and the revised manuscript.

(B) Experimental method

Experimentally, the objective of our study concerns demonstration of the mechanism that those Fermi surface sheets crossing topological nodal planes exhibit symmetry-enforced crossings. It concerns also the demonstration that the existence of the nodal planes depends on the direction of the magnetization, thus proving the symmetry-enforced mechanism as well as the capability to control the existence of the topological crossing experimentally.

While ARPES has become the method of choice in a very wide range of studies of the electronic structure, it is subject to several constraints. Namely, (i) ARPES cannot be used under applied magnetic fields. (ii) ARPES does not offer the energy and momentum resolution required to resolve the crossings. (iii) ARPES in most cases provides indirect

information on the the topological properties in the bulk by tracking surface arcs emanating from the bulk crossings.

We used the de Haas – van Alphen effect, because it allows to resolve the individual Fermi surface sheets in the bulk as a function of angle. The momentum und energy resolution is unprecedented and permits to establish the existence of crossings unambiguously. However, being less frequently used, certain conventions of presenting the data are not broadly known. In particular, harmonics and numerical artefacts due to the choice of window function used for calculating Fast Fourier Transforms are normally not labelled explicitly. This may erroneously leave the impression, that the FFT spectra include features which are not accounted for by the calculated dHvA spectra. Unfortunately, this was not explained properly in the first version of the manuscript leaving the impression that major parts of the dHvA spectra are not accounted for theoretically.

(C) Choice of material

The main interest of our study concerns the experimental demonstration that topological crossings are enforced by symmetry at the intersection between Fermi surface sheets and topological nodal planes. For this reason, a magnetic material is needed in which the effects of topological nodal planes intersecting with Fermi surface sheets can be tracked unambiguously.

It is important to emphasize, that materials supporting a spontaneous exchange splitting and associated uniform magnetization under an applied magnetic field involve d- or even f-electrons. In turn, even nominally simple materials feature large numbers of Fermi surface sheets.

Amongst magnetic materials the Fermi surface sheets and associated dHvA spectra of MnSi are particularly amenable to establish the effect of topological nodal planes, we identify theoretically, for the following reasons (further arguments are presented as part of the point-by-point replies)

- The unit cell of MnSi is simple cubic, yielding the simplest possible shape of the Brillouin zone. This results in the most simple overall shape of the Fermi surface sheets. Moreover, the nodal planes coincide with the Brillouin zone surface (this is contrasted, e.g., by face-centered cubic, or body-centered cubic or hexagonal unit cells of even the simple elemental magnets such as in Fe, Ni and Co, which, however, do not feature the non-symmorphic symmetries enforcing nodal planes).
- de Haas van Alphen frequencies associated with different Fermi surface sheets are spread over a wide range, such that five well-separated regimes may be distinguished. This makes possible the unambiguous assignment and interpretation of the entire spectra.
- Magnetic breakdown involves at most two orbits, thus being tractable; this compares to very complex breakdown situations encountered frequently involving three and more orbits, for instance in systems such as Fe, Ni or Co.
- The Fermi surface sheets intersecting with the nodal planes (pairs (5,6) and (7,8)) are large. In turn, the topological protectorates are also large. At the same time the same Fermi surface sheets (pairs (5,6) and (7,8)) are well separated from the other Fermi surface sheets, such that their spectra are not subject to magnetic breakdown with any of the other Fermi surface sheets.

- The dHvA orbits associated with FS sheets (5,6) and (7,8) permit simultaneous observation of the crossing with and without the nodal plane, thus providing direct access to the symmetry-enforcement and the associated capability of switching the topological crossings on and off.
- The symmetry of the bare torque signal permits unambiguous identification of the remaining pair of FS sheets that intersects with the nodal planes (FS sheets (9,10)), despite the enormous sensitivity of the theoretical predictions for these specific sheets, thus enabling a reasonable estimate of their size.

Given the generic preconditions associated with an experimental demonstration of the effects of nodal planes in magnetic materials, MnSi represents a remarkably tractable and controlled case.

Referee #1 (Remarks to the Author):

In this work, the authors study the symmetry enforced nodal planes of the field-polarized ferromagnetic MnSi by comparative mapping of the extremal cross-sections of Fermi Surface between theoretical and experimental dHvA spectroscopy, and generalize the discussion of topological nodal planes to all 1651 magnetic space groups. The theoretical analysis is very detailed, and makes a catalog of magnetic space group with symmetry-enforced nodal planes. In this way, the present work provided a comprehensive knowledge for theoreticians and experimentalists to explore the topological nodal planes in magnetic materials. However, for the experimental aspect, there are a mass of works need to do, so that make the conclusion more solid, reveal the topological properties of the nodal planes more visible, and demonstrate the control ability of the magnetic field on the topological nodal planes. Below are my comments and suggestions that might be clarified before its publication.

We reply: We wish to thank the referee for her/his great efforts made to review our manuscript and for preparing a very detailed and considerate report helping us greatly to improve the presentation of our results. We were delighted to read that the first referee welcomes our study as *“a comprehensive knowledge for theoreticians and experimentalists to explore the topological nodal planes in magnetic materials.”* We were especially pleased to read that the referee indicates that our paper may be published once the points she/he raised have been addressed appropriately.

Detailed inspection of the large number of comments and recommendations of the first referee revealed that all questions may be readily answered. Concerning seemingly unresolved issues raised by the first referee, it transpires that all of the information and data were already available. We trust that they were overlooked by the referee because of insufficiencies of our presentation. We apologize for these short-comings and hope the referee finds the revised manuscript much improved.

Referee #1:

Comments:

1. Topological nodal plane is not a new concept, which has been widely studied and experimental observed in previous works [Xiao M, et al. arXiv:1709.02363 (2017), Chang, G., et al. Nature Mater 17, 978 (2018), Yang, Y., et al. Nat Comm. 10, 5185 (2019), Xiao M, et al. Science advances, 6, 8 (2020)].

We reply: We agree entirely with the referee that topological nodal planes are not a new concept as such, as already stated in the first version of our manuscript. However, as documented by the references provided by the referee, and consistent with the theoretical studies we cited already in the first version of our manuscript, topological nodal planes had neither been investigated properly from a theoretical point of view as concerns a full symmetry analysis, nor studied experimentally in the electronic structure of real materials, notably magnetic materials.

An important line of previous work was reported for phononic metamaterials. Starting with the seminal theoretical proposal by Meng Xiao and Shanhui Fan [arXiv:1709.02363 (2017)], who constructed a highly specialized scenario, where they started with a Hamiltonian describing topological nodal planes that allowed them to identify a specific symmetry (G_{22}).

This symmetry was subsequently analyzed by means of a tight binding model. The authors proposed further to test this scenario in phononic metamaterials. Such topological nodal planes were then indeed observed experimentally in two studies of phononic metamaterials [Yang, Y., et al. Nat Comm. **10**, 5185 (2019), Xiao M, et al. Science Advances **6**, 8 (2020)]. Thus, these studies started from a bespoke band structure and demonstrated topological nodal planes for this specific band structure in a purely classical setting.

The work we report in our manuscript differs from these seminal studies in the following ways:

- Whereas phononic metamaterials represent a classical setting to test specific models, we address the relevance of topological nodal planes for the electronic structure of real materials, i.e., systems in which quantum effects, the Pauli exclusion principle, and electronic correlations are important, and where the topological nodal planes are a property of the ground state Fermi sea.
- In our theoretical analysis we demonstrate that a very simple symmetry (screw rotation plus time-reversal) is sufficient for the formation of topological nodal planes. Our analysis establishes further that all bands are then affected generically in materials belonging to certain magnetic space groups!
- Analyzing all 1650 magnetic space groups we identify magnetic space groups that support symmetry-enforced topological nodal planes.
- We demonstrate further, both theoretically and experimentally, that due to the symmetry-enforced mechanism a magnetic field permits to switch the topological nodal planes on and off.

We are also grateful that the referee mentions the work by Chang, et al. [Nature Mater **17**, 978 (2018)]. Due to a mishap, the reference to this work unfortunately was lost during the final revision of our manuscript when shortening the text to match the stringent length requirements of Nature. We sincerely apologize for this mistake.

Writing this, it is important to note that Chang et al. report a theoretical study focusing entirely on the topological properties and the putative existence of Kramers-Weyl fermions in *non-magnetic* chiral materials. The presence of nodal planes is mentioned only briefly, where the topological properties are not analyzed in any detail. Further, Chang *et al.* report a tight-binding model of paramagnetic SG 19, Te in paramagnetic SG 4, and α -Ag₂Se_{0.3}Te_{0.7} in paramagnetic SG 19. The last two materials exhibit nodal planes about 0.4 eV *below* the Fermi level, and thus far away from the Fermi surface, i.e., the nodal planes bear no importance on the topological crossing points in these materials! On this note we wish to emphasize, that it is of key importance to identify real materials, where the nodal planes are located at the Fermi energy to be relevant for the physical properties and their relevance in applications.

In particular, Chang et al do not address (i) the symmetry-enforced origin of topological nodal planes we report, (ii) the interplay of nodal planes with the Fermi surface, where we show the formation of topological protectorates, (iii) a full assessment of the distribution and origin of topological charges across the entire band structure, and, (iv) the possibility to exploit the symmetry-enforcement for a tuning of the topological protectorates as a function of field orientation.

In the revised version of our manuscript we account in more detail for these previous studies, to bring out better the novel aspects of our work.

Referee #1: The main innovation of this work is to generalize this concept into the magnetic materials MnSi, and try to control its physical properties by the external magnetic field. However, MnSi seems not a suitable material to study the underlying physics of topological nodal planes for the following reason.

First, one important criterion for topological nodal planes is that the materials with appropriate electron filling. As the authors elaborated in the supplementary S1.B, the magnetic space group SG 19.27 (P212'12'1) has a minimal insulating filling of $\nu \in 4Z$.

The topological nodal planes are only formed between bands $n=2m-1$ and $n=2m$, where m is an integer. However, this is not case in ferromagnetic MnSi, where both Mn and Si atoms occupy 4a Wyckoff positions, indicating that ferromagnetic MnSi possess $4Z$ number of electrons filling per primitive cell. Thus, MnSi is not a typical compound with “filling-enforced” topological nodal planes.

Though the Fermi Surface cross nodal planes as shown in Fig1.e and Fig1.f, detailed discussion of the difference between the typical topological nodal planes and the nodal planes in MnSi should be supported both from theoretical and experimental results.

We reply: We greatly appreciate the remark of the referee about filling-enforced nodal planes and the possibility to clarify this very important issue at the heart of our main results. As explained in the summary given above, the notion of filling-enforcement represents a standard approach how to identify topological crossings close to the Fermi level. However, to the best of our knowledge, there are no generic arguments for filling-enforced topological nodal planes that dominate the entire Fermi surface. The notion of topological nodal planes as introduced by us and their intersection with FS sheets provides a completely different perspective, making specific filling requirements obsolete as elaborated in the following.

Indeed, consideration of filling enforcement requires considerable caution in magnetic systems. For a given spin species (spin up or spin down) the minimum number of connected bands is four, as shown in Fig. 1(d). Note that the tight-binding model of Supplementary Note S2A is only for one species of spin. Therefore, as the referee points out, if there is only this spin species (and if there are no strong correlations) the material must be (semi-)metallic if the filling of this spin species, i.e., the number of electrons of this spin species in the primitive unit cell, is different from $4Z$. Vice versa, if the filling is exactly $4Z$ then the material must be insulating.

However, in magnetic materials one generally needs to consider both spin species at the same time, as their bands overlap and, moreover, correlation effects might need to be considered as well. In addition, the size of the spontaneous spin polarization, which reflects the underlying magnetic interactions, may vary substantially, and is not related to the filling in any simple manner. Therefore, the argument expressed by the first referee cannot be applied in a straightforward manner.

Filling enforced topological phases were discussed in the context of topological insulators, i.e., there are certain topological insulators, whose insulating nature is enforced by the electron filling [see, e.g., H. C. Po et al., *Science Advances* **2**, e1501782 (2016)]. The discussion was extended to topological semi-metals supporting point nodes, i.e., Weyl or Dirac systems, where the chemical potential is located at the point node as enforced by the particular electron filling [see, e.g., S. M. Young et al., *Phys. Rev. Lett* **118**, 186401 (2017) and R. Chen et al., *Nature Physics* **14**, 55 (2018)].

The situation is more sophisticated for nodal-plane materials with band crossings on an entire plane. First of all, the nodal planes are in general dispersive, i.e., bands *within* the plane are dispersive, such that only parts of the nodal plane are at the Fermi level regardless of electron filling. We refer to these intersections as topological Fermi surface protectorates. Considering the proposals and calculations in the literature, we find weakly and strongly dispersive planes, which vary in proximity to the Fermi energy.

It is, nonetheless, tempting to consider the case where the nodal planes are non-dispersive, which may be possible, at least in principle, in the presence of additional lattice or chiral symmetries [for a realization of this in a metamaterial see M. Xiao et al., *Science Advances* **6**, eaav2360 (2020)]. This would lead to flat bands with a large density of states, for which, however, the absence of strong correlations can no longer be assumed and the usual arguments about filling enforcement become invalid. Hence, arguments about filling enforcement cannot be applied directly to magnetic nodal-plane materials, and would have to be extended to account for correlations and/or multiple spin species. This is an interesting and very rich topic for future research.

Finally, we would like to emphasize, that for a material to have symmetry-enforced nodal planes controlling the low-energy physics, only two conditions must be satisfied: (i) the material must crystallize in one of the space groups listed in Table EIII and (ii) the material must be metallic with Fermi surfaces that extend across the BZ boundary. That is, stringent filling conditions are not even necessary for the mechanism we propose to realize topological crossings.

Since there is a large number of magnetic space groups that enforce nodal planes, these conditions are generic and not restrictive at all. No fine tuning of the Fermi level is needed to be located at the topological crossings in stark contrast to, e.g., Weyl or Dirac semimetals. Moreover, even a considerable detuning of the chemical potential or change in the band structure (by, e.g., impurity doping or other deformations) are generally not able to shift the nodal plane away from the Fermi level or remove it altogether. In turn, the topological character of the Fermi surface (i.e., the topological protectorate) remains robust under perturbations, as long as the magnetic screw rotation symmetry is not broken.

We have revised the manuscript to communicate the state of the art and our findings better. In particular, we have added a brief discussion about these points in the Supplementary Note S2.

Referee #1: Second, the Fermi surfaces in MnSi are too complicated, which make the existence of the fermi arcs originated from nodal planes is unambiguous even in the authors' calculations. As theoretical calculated surface states shown in Fig E4, the Fermi arcs will merge into the bulk states totally, even they really exist.

We reply: We comment at first on the claim of the first referee, that the Fermi surface of MnSi is too complicated, before turning to her/his comments regarding the Fermi arcs.

The existence of symmetry-enforced topological nodal planes we predict theoretically requires experimental investigation of suitable magnetic materials featuring a spontaneous spin polarization. This implies that a high density of states is needed at the Fermi level available only in d- or f-electron systems. Considering typical elemental ferromagnets such as Fe, Ni, Co or weak ferromagnets such as ZrZn₂, Ni₃Al or UGe₂ (all of which are symmorphic, but otherwise similar to MnSi) illustrates that the Fermi surface forcibly comprises s-, p-, d- and even f-electron bands with a large number of sheets. In other words, to clarify our theoretical prediction, one is always confronted with Fermi surfaces that may be perceived as being too complicated.

However, within this class of materials, the Fermi surface of MnSi and the associated dHvA spectra offer several important advantages concerning the question under investigation as summarized above. Namely, a simple cubic unit cell, well-separated regimes of the dHvA spectra, comparatively limited and tractable magnetic breakdown, large separated FS sheets that intersect with the nodal planes where they feature topological protectorates (FS sheets 5/6 and 7/8, being in addition very similar), as well as the unique symmetries of FS sheets 9/10, being also well-separated. All of this conspires in a way that it is possible to demonstrate beyond doubt the symmetry-enforced mechanism we identify theoretically.

Emphasizing Fermi arcs, the first referee appears to allude to recent methodological procedures employed in various high-profile reports, which infer the existence of unusual topological band crossings from the observation of surface states by means of ARPES. It is important to emphasize, that the interest in recording Fermi arcs in these studies seems to be born out of the lack of alternative methods to determine the states in the bulk directly. In fact, ARPES may not be used under magnetic field to start with. More importantly, the de Haas van Alphen effect offers high-resolution angle-resolved information for each FS sheet in the bulk. Keeping in mind its material-specific advantages, MnSi, despite its seeming complexity at first sight, is hence very well suited.

It is nonetheless also instructive to consider the Fermi arcs of MnSi in their own right. First of all, we note that MnSi has the simplest type of nodal planes, namely nodal plane duos (two nodal planes) instead of nodal plane trios (three nodal planes). The latter occur in many paramagnetic and antiferromagnetic space groups. Fermi arcs exist only for the nodal plane duos, but not for the nodal plane trios. That is, there exist single Fermi arcs between unpaired Weyl points in the bulk BZ and the nodal plane duos at the BZ boundary, providing a clear signature of the nontrivial topology. For MnSi we find that near the Fermi level there are two unpaired Weyl points on $Y-\Gamma-Y_1$ from two different band pairs, leading to two Fermi arcs (see Fig. E4). The referee is correct that these Fermi arcs merge with bulk state before they reach the nodal plane duos. Nevertheless, the existence of these Fermi arcs proves the topology of the nodal plane duos, because all other Weyl points away from $Y-\Gamma-Y_1$ have multiplicity two, and hence the single Fermi arcs emanating from $Y-\Gamma-Y_1$ must connect to the nodal plane duos.

In addition, our theoretical analysis shows that there are many other (ferro-)magnetic space groups with topological nodal plane duos and associated Fermi arcs, where we identify already a sizable number of materials that crystallize in these magnetic space groups. The

study of these defines a new and rich field of research concerned with the identification and control of topological protectorates in different materials.

We have revised the manuscript to communicate better the advantages of choosing MnSi as a material for our study and the de Haas-van Alphen effect as the experimental method.

Referee #1: Thirdly, the complex FSs in MnSi may obliterate the properties of nodal planes. To figure out this problem, I strongly suggest the authors to compare the transport measurements when magnetic field applied along (010) and (110) and the other direction. If the magnetic-controlled electromagnetic response, the anomalous Hall effect can be observed, it will make this work more valuable.

We reply: We wish to thank the first referee for expressing this concern. It allows to emphasize that the topological nodal planes are symmetry-enforced as opposed to accidental and the complex FS certainly does not obliterate the effects of the nodal planes. The complexity of the Fermi surface proves to be an advantage, in principle, as all of the bands crossing the nodal planes are affected, regardless of the complexity of the system! This state of affairs is confirmed by the similarity of the observations for sheet (5,6) and (7,8) as explained above (see also the comment on FS sheets (7,8), which were observed in contrast to the impression of the first referee).

We also wish to thank the first referee for bringing up putative information on topological crossings accessible in transport measurements. Several papers have reported magnetotransport properties of MnSi, revealing in particular a large anomalous Hall effect in the spin-polarized state [e.g. M. Lee et al., Phys. Rev. B **75**, 172403 (2007)].

It transpires that some of us (A.B. and C.P.) were also involved in a study connecting the Hall effect of MnSi and $\text{Mn}_{1-x}\text{Fe}_x\text{Si}$ with ab initio calculations [C. Franz et al., Phys. Rev. Lett. **112**, 186601 (2014)]. We apologize sincerely for not having cited this paper. As a main result this study established quantitative consistency of the electronic structure with normal, anomalous and topological Hall contributions in MnSi and the Fe-doped compounds. The Fermi surface calculated as part of this study is in excellent agreement with the Fermi surface determined experimentally in our dHvA measurements. The account of the anomalous Hall effect establishes strong Berry phase contributions. While there is no doubt on the importance of large Berry curvatures, we agree with the referee, however, that this provides putative evidence of topological nodal planes, but does not prove their existence as such.

To go beyond the work reported in Franz et al. in Phys. Rev. Lett. **112**, 186601 (2014) requires a numerical assessment of the transverse and longitudinal magnetoresistance, with an account of the effects of intra- and interband transitions, as well as defect-related scattering [see also the discussion in section V(B) in R. Ritz et al. PRB **87**, 134424 (2013)]. This represents an independent research program well-beyond the work reported in our manuscript.

The first referee suggests that there may be differences of the transport properties for different crystallographic directions. We are not aware of studies having addressed such anisotropies. A careful quantitative study along these lines is already rather demanding on

the experimental level regarding a proper account of systematic errors, let alone comparison with ab initio predictions well beyond our study.

We have added a comment in the revised manuscript pointing out the putative evidence for strong Berry curvatures associated with topological nodal planes, as observed in the large anomalous Hall effect of MnSi.

Referee #1:

2. The experimental identification of the nodal planes on the FS is indirect. The ferromagnetic phase of MnSi possess complex FS, in which sheets 5-10 are formed by nodal planes from DFT calculations. To identification of the complex FS sheet from experimental dHvA spectra, the authors systematically compared with theoretical results as shown in Fig3. The quantitatively mapping is not enough to confirm the FS sheets of experimental results.

First, there is not experimental spectra in sheet 7 and sheet 8, which form TPs with larger cross-sections than sheet 5 and sheet 6, the authors simply explained as too weak to observe it, then what's the exactly value of it?

We reply: We wish to thank the first referee for this comment, which suggests that we did not communicate the complete set of data we recorded appropriately. It is important to emphasize that the assignment of the cyclotron orbits to the observed dHvA frequencies involves a combination of many different internally consistent arguments as summarized in the Methods section and Supplementary Notes S4 and S5. The assignment is not just based on a mere side-by-side comparison with the calculated spectra!

We disagree with the statement of the first referee that the evidence for topological nodal planes is indirect. The resolution and weight of the spectra are perfectly clear and unambiguous for both sheets (5,6) and (7,8).

Concerning sheet 7 and 8, the first referee is incorrect. We have observed several experimental branches that are clearly arising from sheet 7 and sheet 8. In particular:

- 1) Branch \varnothing in region III can be clearly assigned to the loop orbit shared between sheet 7 and sheet 8. This is the orbit that is analogous to the loop orbit shared between sheet 5 and sheet 6 discussed in detail in Fig. 4.
- 2) Branch Θ and branch K forming the small „eye“ in region II both originate from the neck of sheet 7.

There is thus unambiguous experimental evidence for the existence and the size of sheet 7 and sheet 8. While this was clearly stated in the text (lines 153 through 157 of the old text), there was also a remark, that we do not observe one specific heavy orbit of sheet 8. However, regarding this remark it is important to note that (i) an anomalous splitting of sheet 5/6 we observe at the calculated intersection with sheet 8 around 6.5 kT provides strong empirical evidence for the missing orbit, and (ii) the calculated heavy mass and the steep dispersion of sheet 8 provides very strong plausibility arguments, why this orbit is not

observed over an extended range with torque magnetometry, being insensitive around the $\langle 010 \rangle$ axis (see also the discussion on the symmetry of the signal of sheets 9/10).

Our data provide also clear evidence of the interplay of topological nodal planes with sheets 7 and 8. The key signatures are exactly analogous to the loop orbit shared between sheet 5 and sheet 6. Without the topological nodal planes, the topological orbit σ would not exist and we would have two trivial branches well above and well below σ . The lower of these trivial branches would display a minimum at about 2kT for $B \parallel [010]$ and would be well visible in the spectrum, owing to its band mass $m_b \sim 2m_e$ which is comparable to the very prominent branch of sheet 5 at $\sim 2.4kT$ which has $m_b \sim 1.9m_e$ (blue lines in Fig. 3a1). Thus, the existence of σ and the absence of the other two branches proves the existence of TPs also for this sheet pair.

In our manuscript we decided to present our key findings demonstrating topological nodal planes for the case of sheet 5 and sheet 6, because the corresponding branch is isolated in regime I, while that of sheet 7 and 8 is located in regime III and more difficult to appreciate at first sight. The key signatures of sheet 7 and 8, which are identical to sheets 5 and 6, are presented as part of the extended display items and the supplementary information for lack of space in the main text. Given the mere volume of information of our study, probably the first referee just overlooked this information accidentally and we apologize for not having presented the evidence more clearly.

In the revised manuscript we added a comment in the main text and extended the discussion of sheet 7 and sheet 8 in the Supplementary notes S5 to communicate better that the consequences of TPs on sheet 7 and sheet 8 are evident from the experimental spectra.

Referee #1: Second, there is another obvious orbital except sheet 2 in region IV in experimental data, while there is not emerging in theoretical results, can authors provide more detail information of this frequency?

We reply: We wish to thank the referee for raising this question. It transpires that higher harmonics of fundamental dHvA frequencies conventionally are not labelled. Indeed, the frequency branch in regime IV mentioned by the referee represents the first harmonic of the κ -branches. This first harmonic may be accounted for by the sum in Eq. (4) with $p=2$. This and higher harmonics can typically be observed in the FFT spectra for dHvA frequencies with a large signal amplitude. They are, however, much weaker due to their apparent mass that scales with the harmonic index p [see Eq. (6)]. In the revised manuscript we have labelled the branch under contention as the first harmonic for better clarity.

Regarding the theoretical calculations, it is likewise customary to show only the fundamental dHvA branches and not their harmonics, as the latter do not correspond to any additional extremal Fermi surface cross sections. For the sake of clarity and consistency we also show the first harmonic calculated theoretically in the revised manuscript (Fig. 3b).

Referee #1: Thirdly, the dHvA spectra in region V is fuzzy, which is hardly resolve the FS sheet 9 and 10. As a completely experimental comparison, can the authors add more theoretical dHvA spectra to Fig E5.g? For these results, it's tough to convince readers that an unambiguous identification of the FS sheets was achieved. It is therefore extremely

important to clarify experimental frequencies before a decisive conclusion can be drawn. It would be nice to improve the resolution for lower temperature, as the dHvA oscillations of torque were resolved in temperature $T = 35$ mK.

We reply: Regarding the main objectives of our study, notably demonstration of the existence of the symmetry-enforced topological nodal planes and their dependence on the magnetization, this mechanism is clearly proven by the properties of sheets 5/6 and 7/8.

It is therefore sufficient to demonstrate that all sheets of the FS have been detected, which requires to unambiguously determine the origin of the frequencies in regime V. This is possible due to the unique symmetry of the main branches in regime V, namely β , γ , δ , ϵ and μ , which proves unambiguously that they are due to sheets 9 and 10 along with a reasonable estimate of the size of the orbits even though there is no detailed match with theory!

Namely, the strength of the torque signal varies with $(1/f)(df/d\phi)$. In turn, the signal vanishes at high-symmetry directions where $f(\phi)$ is stationary. Inspecting Fig.3, all signals in regimes I to IV vanish at the [010] direction. This implies that they must be due to FS sheets centered at the Γ -point, i.e., FS sheets 1 to 8. In contrast, the branches associated with sheets 9 and 10 in regime V are not stationary around [010] and consequently the signal does not vanish around [010] as can be seen in Fig.3b. Instead, the branches are stationary around [-111] as shown in Extended Display Item Figure E5. Thus the FS sheets associated with the frequencies observed in regime V must be centered at the R-point, unambiguously connecting them with FS sheets 9 and 10!

With the bands underlying FS sheets 9 and 10 being very flat and extremely sensitive to the precise direction of the magnetic field in the vicinity of the R-point, the discrepancy of the experimental and theoretical frequencies of sheets 9 and 10 is due to limitations of the numerical calculations, being in addition exceptionally sensitive to the precise adjustment of the Fermi level. In this sense the clear separation of the frequency spectra in five regimes represents a major advantage of MnSi as compared to dHvA spectra observed in other metallic magnets.

While the calculations of FS sheets 9/10 are extremely sensitive to details, and precise agreement with experiment cannot be reached, they clearly demonstrate the presence of symmetry-enforced crossing of sheets 9/10 at the intersection with the nodal planes. This underscores our assessment and provides further evidence for the mechanism we propose.

Regarding the suggestion of the first referee to improve the resolution of the data in regime V, we note that the spectra in regime V were measured at $T = 280$ mK. Lowering the temperature would therefore not improve the quality of these spectra, since the carrier masses of these branches are between $0.4m_e$ and $5.4m_e$, yielding a temperature factor $R_T > 0.5$ at high fields (cf. Fig. 2(c) in the old text; Fig. 2(d) in the revised text). In fact, the resolution is limited by the FFT window size in this regime, and not so much by the temperature. Unfortunately, the resolution cannot simply be increased by increasing the window size (e.g., using high-field Bitter or pulse magnets), because the frequencies *shift* as a function of field due to the non-saturating magnetization as discussed in the methods section. This field-dependent shift adds to the FFT peak width.

Taken together, we can unambiguously assign a variety of experimental branches in regime V to the sheet pair 9 and 10 by their symmetry properties alone, even without knowing their

precise location on the sheets. This also fixes the approximate size of the FS pockets. The remaining quantitative mismatch between experiment and theory does not affect the main conclusions of our study. In the revised manuscript we have clarified our presentation regarding this aspect of our study.

Referee #1: Additional questions:

1. For the magnetic moments oriented along [010], the two nodal planes at the $k_x = \pi$ and $k_z = \pi$ with total topological charge $\nu_{\text{npd}} = 1$ or -1 , which compensated the topological charge of Weyl point along Γ_1 - Γ_1 , indicating each nodal plane with fractional topological charge? Can the authors give a more detailed theoretical analysis of it? Especially for surface states in the [010], why the chiral Fermi arcs connected to the Γ_1 - k_z direction but not k_x -direction?

We reply: The two nodal planes on $k_x = \pi$ and $k_z = \pi$ should be viewed as a single entity, which we refer to as a “nodal plane duo”. These two nodal planes together (i.e., the nodal plane duo) carry a quantized topological charge. It is not possible to assign topological charges to each of the two nodal planes individually, since there does not exist a 2D closed integration contour (see Supplementary Note S2C) which encloses only a single nodal plane and on which there exists everywhere a band gap. Any such contour must cross the other nodal plane, where the band gap closes. Hence the topological charge can only be defined for the two nodal planes together.

The handedness of the chiral Fermi arcs is determined by the Chern number of the Weyl point [see, e.g., N. Schröter et al., Nat. Phys. **15**, 759 (2019)]. Topology, however, does not determine to which of the two nodal planes the Fermi arc connects. This, presumably, depends on the surface termination and other details.

We extended the discussion of the topological charge of the “nodal plane duo” in Supplementary Note S2 to clarify these points.

Referee #1:

2. In supplementary S3.A, the 3 conditions that need to be satisfied for the existence of symmetry-enforced nodal planes are necessary and sufficient condition?

We reply: We wish to thank the referee for this question, which touches on an equally important as subtle point. The three conditions for *symmetry-enforced* nodal planes listed in Supplementary Note S3A are indeed both sufficient and necessary:

The only non-symmorphic symmetries that can possibly lead to enforced band crossings on *entire planes* are two-fold screw rotations (say, around the k_j axis) combined with time-reversal symmetry. This combined symmetry leads, by Kramers theorem, to nodal planes on the $k_j = \pm \pi$ plane, provided that this plane is actually a BZ boundary of the space group under consideration. Finally, the space group must not contain any symmetries that lead to two-fold degeneracies in all neighborhoods around the nodal plane, which excludes the combination of inversion with time-reversal symmetry (with possibly some translation part).

As an aside, we note that there exist also *accidental* nodal planes which are not symmetry-enforced and only protected by a topological invariant, but not by a non-symmorphic symmetry. Such accidental nodal planes have been discussed see, e.g., O. Türker and S.

Moroz, PRB **97**, 075120 (2018) and T. Bzdusek and M. Sigrist, PRB **96**, 155105 (2017). The perhaps most natural examples are Bogoliubov Fermi surfaces in unconventional superconductors [see, e.g., P. Brydon et al., PRB **98**, 224509 (2018)]. Classifications of these accidental nodal planes have been discussed in, e.g., [Zhao *et al.*, PRL **116**, 156402 (2016) and Chiu *et al.*, RMP **88**, 035005 (2016)].

We have extended the presentation at the beginning of Supplementary Note S3A to clarify these points.

Referee #1: As the irreducible co-representations of the magnetic space groups has been posted in Bilbao Crystallographic Server, the analyzed results from authors are all self-consistent with this website?

We reply: We wish to thank the referee for raising this question, which concerns the main theoretical achievements of our study. Indeed, our catalogue of magnetic space groups with nodal planes (i.e, Table EIII) is consistent with the irreducible co-representations that have recently been posted on the Bilbao Crystallographic Server. That is, all space groups with symmetry-enforced nodal planes have two-dimensional co-representations of the little groups on one (or more) BZ boundary planes.

We note, however, that the *topology* of the nodal planes cannot be determined from the co-representations alone, but must be inferred from the *global* topology in the entire BZ, i.e., from the chirality and multiplicity of all band crossings in the entire BZ. Figuring out this global topology for all relevant (magnetic) space groups is one of our major theoretical achievements, as discussed in Supplementary Note S3.

Elucidating this issue in further detail, it is important to distinguish accidental nodal planes from the symmetry-enforced nodal planes we identified in our study as explained above. Examples for such accidental nodal planes supporting a topological invariant have been presented by O. TÜRker and S. Moroz, PRB **97**, 075120 (2018) and T. Bzdusek and M. Sigrist, PRB **96**, 155105 (2017) and others.

In comparison to and as emphasized above, for symmetry-enforced nodal planes the three conditions presented in Supplementary Note S3A are indeed necessary and sufficient. The only symmetry that leaves two-fold planes invariant, while possibly enforcing crossings, are two-fold screw rotations with time-reversal. For them to fulfill Kramers theorem the existence of an invariant plane perpendicular to the rotation axis and at $k = \pm \pi$ is required. Finally, all possibilities that remove the nodal characteristics of the plane have to be excluded.

The only symmetry that leaves any point in the vicinity of the nodal plane invariant is the combination of inversion, time-reversal and optionally a translation. By excluding this combination there is no possibility to remove the nodal characteristics of the symmetry-enforced nodal plane.

It is also interesting to note that nodal planes may intersect with further symmetries. For instance, this is the case in in SG 26.67 (gray group with a trivial plane) for a screw rotation 2_{001} as combined with M_{100} und M_{010} . Anticommuting the mirror operation with the rotation results in nodal lines connecting the nodal planes.

We have expanded the presentation of our study in Supplementary Note S3 to clarify these points.

Referee #1:

3. The FS in Fig.1f is enlarged? If it is, please illustrate it. Because the frequency is Fig2.b and Table EII are just few KT, it is puzzling to me that even these frequencies can map to large FS?

We reply: The FS shown in Fig. 1f is drawn to scale and not enlarged, with the box illustrating the simple cubic first Brillouin zone. The primitive lattice constant is $a=4.558$ Angstrom, which corresponds to a reciprocal lattice vector with a magnitude of $\sim 1.38 \times 10^{10}$ 1/m. Thus, using the Onsager relation $f=(\hbar/2\pi e) \cdot A_k$, the area of a complete Brillouin zone face corresponds to 19.9kT. The largest orbits observed have an area of about 1/3 of a Brillouin zone face, i.e, 6.7kT.

Referee #1:

4. For the discussion of FFT magnetic torque's spectra in Fig2.b, there have many contiguous frequencies, is there comes from Zeeman splitting? The other question puzzled to me is that some frequencies are marked, while some frequencies are not, can the authors give a detail explanation?

We reply: We wish to thank the first referee for raising this very important technical issue which alludes to the discrimination of artefacts in the computation of the FFTs. Marked and labelled in Fig. 2(b) [Fig. 2(c) in the revised manuscript] are all peaks that correspond indeed to dHvA frequencies and their harmonics. In contrast, some of the peaks visible in Fig. 2(b) [Fig. 2(c) in the revised manuscript] represent side lobes arising from the FFT analysis in a finite reciprocal field window.

In our data analysis we applied a wide range of complementary methods and tests to delineate quantum oscillatory components of the torque signal from spurious artefacts due to the specific implementation of the Fast Fourier transforms (FFTs) and background subtraction. Notably, in the determination of the FFTs this included the use of different window functions and the use of different background subtraction schemes, as well as the calculation of FFTs from *synthesized* quantum oscillations using the Lifshitz-Kosevich (LK) formalism. In addition, we considered the angular evolution of dHvA branches to confirm that the dHvA frequencies were identified correctly. An extensive discussion of these tests is presented below as part of the reply to the third referee, which expressed this question more technically.

It is, finally, also important to emphasize that the oscillations we observe experimentally do not reflect any Zeeman splitting, since the experiment was performed in the field-polarized phase of magnetic MnSi, where an exchange splitting on the scale of 360 meV separates spin-up and spin-down electrons, i.e., the exchange splitting exceeds the Zeeman energy by many orders of magnitude.

Referee #1:

5. The authors deduced the effective mass for each orbital ranging from $m^*=0.4$ me to 17 me. However, the maximal effective mass is 15.1me as shown in Fig2.c.

We reply: We thank the referee for pointing this out. For better consistency we adapted Fig. 2(c) [Fig. 2(d) in the revised manuscript] and replaced the temperature dependence of the M orbit, which exhibits an effective mass of $15.1m_e$, with the temperature dependence of the E orbit, which exhibits an effective mass of $17m_e$. In the first version of the manuscript we had shown the temperature dependence of the M orbit, as it corresponds to the highest dHvA frequency, i.e., Fig. 2(c) in the old manuscript displayed selected temperature dependences between the lowest and highest frequencies visible in Fig. 2(b) [Fig. 2(c) in the revised manuscript].

Referee #1:

6. It's hard to follow some interesting results of the current manuscript without references. For example, $\gamma = 30 \text{ mJmol}^{-1}\text{K}^{-2}$ at $B = 9 \text{ T}$ in line 165, the value of experimentally determined specific heat in Table EI, and especially for the methods of Internal magnetic field and dHvA frequency $f(B)$ in a weak itinerant magnet, can authors add relative references in these places?

We reply: We wish to thank the first referee for requesting further references to the literature. Several groups have reported thermodynamic and transport properties of MnSi in the field polarized state. Concerning our own work, the value of γ was actually determined as part of a series of studies reported in, e.g., A. Bauer et al., Phys. Rev. B **82**, 064404 (2010). In the revised manuscript this paper is now cited in both positions, i.e., line 165 and in Table EI.

Regarding the literature reporting the frequency correction for the influence of the non-saturating magnetization this goes back to Ruitenbeek et al., J. Phys F: Met. Phys **12**, 2919 (1982) in pulsed-field study of ZrZn_2 , and has been applied for instance by Kimura et al., Phys. Rev. Lett. **92**, 197002 (2004) in a high-pressure dHvA study of ZrZn_2 . The papers by Ruitenbeek and Kimura are now cited in the revised manuscript.

Referee #1:

7. The reference 37 in upper right of $N_d=1/3$ is misleading.

We reply: Thank the referee for pointing this out. We changed the position of the superscript reference.

Referee #1:

8. What's the m_b in eq(3) in methods? It does not appear elsewhere in the manuscript.

We reply: We apologize for this omission. m_b is the band mass (excluding mass enhancements). In the revised version of the manuscript m_b is now defined following eq. (3) in the methods section.

Referee #1:

9. The $B_{av} = 12 \text{ T}$ in methods should specify the FFT range in 10~15T, in order to distinct the 4~15 T as demonstrated before.

We reply: In the revised manuscript the average field for all FFT windows is now explicitly stated in the methods section.

Referee #1: In summary, the present conclusion in this work is not completely self-consistent. More comprehensive analysis between the theoretical calculations and experimental measurements should be addressed. The innovations did not meet the requirement of Nature. More experimental properties related to the topological nodal planes, especially the electromagnetic response, need to be explored.

We reply: We would like to thank the first referee again for her/his great efforts made to review our paper. We trust that we clarified all of the questions and comments raised by the referee satisfactorily, in particular as concerns the status of our experimental data. As emphasized in our reply, a brute force ab initio account of the electromagnetic response of MnSi, in excellent agreement with the observation of a large anomalous Hall effect, was already published in 2014, where the importance of the topological properties of the electronic structure was not addressed. Taken together, we hope that the first referee finds the revised manuscript much improved, recommending its publication in Nature.

Referee #2 (Remarks to the Author):

The main argument made by the authors in this manuscript is that the magnetic screw rotation symmetry, combined with time reversal symmetry, not only force all bands to cross at the Brillouin zone (BZ) boundaries, forming nodal planes and topological protectorates, but also induce Weyl nodes and 4-fold points within the BZ.

We reply: We wish to thank the second referee for her/his efforts made to review our manuscript. While the individual statements in this first part of the summary of the second referee are correct, the wording seems to be somewhat unclear regarding the main achievements we report. While we suspect that this represents just a mishap, we hope that it is acceptable that we repeat the key achievements of our paper in our own words to avoid any ambiguities further below. Namely, the combination of magnetic screw rotation with time-reversal symmetry enforces nodal planes, known since the seminal work of Herring and others in the 1940s. These nodal planes may carry a topological charge that depends on the specific magnetic space group. Moreover, they may be located at the BZ boundary depending on unit cell. In turn, we establish that Fermi surface sheets that intersect with topological nodal planes exhibit topological crossings that support a topological charge. The mechanism we identify is generic, remarkably simple, and guarantees that these topological crossings, which we term topological protectorates, are always located exactly at the Fermi level.

We show further that time-reversal symmetry breaking allows to control the formation of topological nodal planes, and thus the existence of the topological crossing points. Performing a full analysis of the topological properties of these systems, we establish the presence of Weyl nodes and 4-fold points where the charge of the topological nodal planes originates in a single Weyl point.

Thus, we identify the Weyl-nodes and 4-fold points in the BZ not just as a property in its own right. Rather, they are identified as the result of a full assessment of the topological charges in the BZ in order to systematically clarify the origin of the topological charge of the nodal planes. To the best of our knowledge, such a complete assessment of the topological charges has not been reported in recent papers on materials with the same SG [e.g., e.g. Zhicheng Rao et al., *Nature* **567**, 496 (2019), D. S. Sanchez, et al., *Nature* **567**, 500 (2019), Yuanfeng Xu, et al., *Nature* **586**, 702 (2020) and N. B. M. Schröter et al., *Science* **369**, 179 (2020)]. It represents an important contribution in its own right.

Referee #2: The authors demonstrated this picture through combined theoretical and experimental studies on a helimagnetic material MnSi which is well known for showing Skyrmion lattice and a pressure-tuned quantum phase transition. From crystalline symmetry analyses and first principle calculations for this material, the authors first predicted symmetry-enforced nodal planes with topological charges at the BZ boundaries parallel to the magnetization axis. Further, their theoretical analyses also showed the non-trivial topology of these band crossings generates surface Fermi arcs covering large areas of surface BZ. Then the authors attempted demonstrating the predicted topological nodal structure through angular dependent dHvA quantum oscillation measurements. They observed dHvA signatures consistent with the predicted topological orbits.

We reply: We were pleased to read that the second referee in this part of her/his report summarizes accurately that we evaluated our general findings on symmetry-enforced

topological nodal planes for the case of MnSi by means of an analysis of the crystal symmetries, a tight-binding model, density functional theory, and a quantum oscillatory study.

The second referee appreciates that we established the formation of symmetry-enforced topological nodal planes perpendicular to the magnetization axis, and that these planes provide topological charges to the FS at those BZ boundaries perpendicular to the magnetization axis. The referee also points out, that we predict the formation of Fermi arcs due to the topological nodal planes as a consequence of the full analysis. Finally, the referee acknowledges, that the dHvA signatures are consistent with the predicted topological orbits.

Referee #2: Overall, the work reported in this manuscript is novel and interesting. It can possibly facilitate discoveries of new magnetic topological materials, since the argument of this manuscript can also be extended to 254 magnetic space groups. The authors have indeed found 33 of them have nodal planes with non-zero topological charges. These predictions would inspire further experimental studies on materials with those space groups.

We reply: We were pleased to learn that the second referee welcomes our study as ‘novel and interesting’, appreciating that the arguments we present are generic and applicable to a wide range of materials, inspiring future experimental studies.

Referee #2: However, this manuscript is relatively weak on its experimental side. The main argument of symmetry-enforced nodal planes at the BZ boundaries is only based on the comparison of the theoretical and experimental cyclotron orbits. Although this is a reasonable approach, the band structure of MnSi is extremely complicated. It encompasses 10 different FSs and its dHvA spectrum consists of 40 frequencies. While the authors showed the correspondence between the experimentally probed bands and the theoretically calculated ones, there are striking discrepancies between experimental and theoretical dHvA spectra (Fig. 3a and 3b), particularly in the frequency regimes of IV and V.

We reply: We were surprised to learn that the second referee rates the experimental evidence we report as weak. The second referee expresses three points of concern. As these three points of critique represent also the essence of the more technical comments of the second referee given further below, we comment at first on a more general note and provide more detailed replies below. The three points of critique are as follows:

1. *The second referee criticizes that the evidence we present is “only based on the comparison of the theoretical and experimental cyclotron orbits”, which she/he finds a “reasonable approach”.*

The comment by the referee may be understood in two ways. First, regarding the criteria entering the interpretation of the dHvA spectra, and second, whether quantum oscillatory studies represent the best experimental method to demonstrate our theoretical findings.

Regarding the interpretation of the dHvA spectra we wish to emphasize, that the assignment of the orbits is not just based on a side-by-side comparison with the

calculated spectra. Instead we used a long list of internally consistent criteria as described in great detail in the methods section and the supplementary information. As concern the best choice of experimental method, it seems apt to comment on the suitability of ARPES, which has been highly successful in many studies of topological materials reported in the literature, and which the second referee endorses strongly further below.

There is no doubt, that ARPES represents the only possibility to obtain surveys of the overall band structure, especially for samples of mediocre quality. There are, however, several serious constraints with ARPES that are rarely mentioned. These are as follows: (i) ARPES cannot be used under applied magnetic fields. (ii) ARPES provides an overall intensity and cannot resolve specific contributions from different bands, let alone FS sheets. (iii) ARPES does not provide the required energy and momentum resolution to reveal directly the specific features under investigation. (iv) ARPES represents an inherently surface sensitive probe, where the annealing procedures used commonly may generate reconstructions and damage. (v) ARPES is not suitable for studies at ultra-low temperatures, i.e. the low mK range.

Especially points (ii) and (iii) are reflected in recent high-profile papers on other isostructural representatives in SG198 of MnSi, e.g. Zhicheng Rao et al., *Nature* **567**, 496 (2019), D. S. Sanchez, et al., *Nature* **567**, 500 (2019), Yuanfeng Xu, et al., *Nature* **586**, 702 (2020) and N. B. M. Schröter et al., *Science* **369**, 179 (2020). The evidence for Dirac-Kramers fermions reported in these papers is inferred from the observation of surface states without direct evidence of the effects of spin-orbit coupling or the actual topological crossing points.

In comparison, while quantum oscillations are less widely used because they are much more demanding in terms of the experimental conditions, they probe directly the Fermi surface and thus the specific information of interest. Further, quantum oscillations, at least in principle, permit to distinguish individual bands. In addition, quantum oscillations achieve an energy and momentum resolution orders of magnitude higher than ARPES suitable for resolving the actual crossing or absence thereof with meaningful accuracy. Finally, quantum oscillations represent the only method providing information on the bulk Fermi surface under applied magnetic field. Taken together, we therefore disagree strongly with the referee. In our case quantum oscillations represent an extremely powerful probe without any alternative.

2. *The second referee refers to the Fermi surface of MnSi as being “extremely complicated” comprising 10 FS sheets and 40 frequencies.*

As emphasized above, we were pleased to read that the referee welcomes the theoretical identification of symmetry-enforced topological crossing points of the Fermi surface and the broad analysis of the magnetic space groups as ‘novel and interesting’ inspiring future experimental studies. Representing a main new result of our study, the formation of topological crossing points may be controlled by virtue of the orientation of the magnetization. The importance of our results translates into the question which materials may be, in principle, suitable to provide experimental evidence of the mechanism we identified theoretically.

Recognizing the dependence of symmetry-enforced topological nodal planes on the orientation of the spontaneous magnetization, representing the method of choice to generate the symmetry-breaking of interest, implies the need to study Fermi surfaces in suitable magnetic compounds. In turn, this implies the need for a high density of states at the Fermi level available only in materials containing d- or f-electron elements. Considering typical elemental ferromagnets such as Fe, Ni, Co, or weak ferromagnets such as ZrZn_2 , Ni_3Al or UGe_2 the Fermi surfaces comprise s-, p-, d- and even f-electron bands, with a large number of sheets. Within this class of materials, MnSi offers several advantages as compared with typical d- and f-electron compounds. Namely, a simple cubic unit cell, well-separated regimes of the dHvA spectra, comparatively limited and tractable magnetic breakdown, large separated FS sheets that intersect with the nodal planes where they feature topological protectorates (FS sheets (5,6) and (7,8), being in addition very similar), as well as the unique symmetries of FS sheets (9,10), being also well-separated. All of this conspires in a way that it is possible to demonstrate beyond doubt the symmetry-enforced mechanism we identify theoretically.

We therefore strongly disagree with the second referee. Regarding the questions addressed in our study, investigation of the Fermi surface of MnSi, being tractable, is an excellent choice!

3. *The second referee criticizes that there are striking discrepancies between experiment and theory, especially in the frequency regimes IV and V.*

We greatly appreciate this comment by the second referee, which made us aware that we did not present our findings sufficiently well. Referring to frequency regimes IV and V strongly suggests that the second referee is, on the one hand, not familiar with certain conventions of presenting dHvA spectra, and, on the other hand, does not appreciate the connection of the objectives of our study with the information contained in the dHvA spectra. We apologize for not having explained these points better.

Namely in frequency regime IV, following convention, one set of frequency branches was not labelled because they were simply the first harmonics of the fundamental dHvA frequency branches labelled by κ_i . Keeping this in mind, all frequencies in regime IV are in outstanding agreement with theory. To avoid this misunderstanding in the revised manuscript, we labelled the first harmonics with $2\kappa_i$ and also display the first harmonics as calculated theoretically.

Concerning regime V, it is important to note that the abundance of frequencies seen here, based on symmetry arguments associated with the R-point, can be unambiguously attributed to sheets 9 and 10 (the details are presented in the supplementary information, section S5 sheets 9 and 10). Since the first referee raised essentially the same question, we reproduce below the relevant parts of the reply to the first referee for the sake of convenience:

“Regarding the main objectives of our study, notably demonstration of the existence of the symmetry-enforced topological nodal planes and their dependence on the magnetization, this mechanism is clearly proven by the properties of sheets 5/6 and

7/8. It is therefore sufficient to demonstrate that all sheets of the FS have been detected, which requires to unambiguously determine the origin of the frequencies in regime V.

This is possible due to the unique symmetry of the main branches in regime V, namely β , γ , δ , ε and μ , which proves unambiguously that they are due to sheets 9 and 10 along with a reasonable estimate of the size of the orbits even though there is no detailed match with theory!

Namely, the strength of the torque signal varies with $(1/f)(df/d\phi)$. In turn, the signal vanishes at high-symmetry directions where $f(\phi)$ is stationary. Inspecting Fig.3, all signals in regimes I to IV vanish at the [010] direction. This implies that they must be due to FS sheets centered at the Γ -point, i.e., FS sheets 1 to 8. In contrast, the branches associated with sheets 9 and 10 in regime V are not stationary around [010] and consequently the signal does not vanish around [010] as can be seen in Fig.3b. Instead, the branches are stationary around [-111] as shown in Extended Display Item Figure E5. Thus the FS sheets associated with the frequencies observed in regime V must be centered at the R-point, unambiguously connecting them with FS sheets 9 and 10!

With the bands underlying FS sheets 9 and 10 being very flat and extremely sensitive to the precise direction of the magnetic field in the vicinity of the R-point, the discrepancy of the experimental and theoretical frequencies of sheets 9 and 10 is due to limitations of the numerical calculations, being in addition exceptionally sensitive to the precise adjustment of the Fermi level. In this sense the clear separation of the frequency spectra in five regimes represents a major advantage of MnSi as compared to dHvA spectra observed in other metallic magnets.

While the calculations of FS sheets (9,10) are sensitive to details, and precise agreement with experiment cannot be reached, they clearly demonstrate the presence of symmetry-enforced crossing of sheets (9,10) at the intersection with the nodal planes. This underscores our assessment and provides further evidence for the mechanism we propose.“

An additional issue not explicitly raised by the second referee, which may also be of interest to her/him, concerns the seeming presence of fine structure in the dHvA spectra shown in Fig. 2(b) [Fig. 2(c) in the revised manuscript]. Marked and labelled in Fig. 2(b) [Fig. 2(c) in the revised manuscript] are only peaks that correspond unambiguously to dHvA frequencies. In contrast, some of the fine structure visible in Fig. 2(b) [Fig. 2(c) in the revised manuscript] represent side lobes arising from the FFT analysis in a finite reciprocal field window. We note that here are no dHvA frequencies that cannot be unambiguously attributed to any of the FS sheets.

An extensive discussion of the methods used to discriminate such artificial effects, is presented as part of the reply to the third referee below. The supplementary information and methods section have been amended to explain the analysis better.

We hope that the above considerations help to better understand the content of the data and the line of the arguments we present.

In the revised manuscript we have clarified our presentation regarding these aspect of our study.

Referee #2: The main experimental proof for the topological protectorates on the nodal planes given by the authors is the consistency between the theoretical and experimental dHvA spectra of the topological orbits which is comprised of different segments of FS sheets 5 and 6 (Fig. 4b1 and 4c). Although this result seems to support the claim of TP and NP, it is not a direct evidence. The complication of the FS, as well as the inconsistency between theoretical and experimental dHvA spectra shown in Fig. 3, puts additional concern on this experimental proof.

We reply: As discussed below the experimental proof is very direct: The orbits we observe as well as the orbits that are absent represent an immediate consequence of the existence of the TPs. Furthermore, it is important to emphasize that the evidence supporting topological nodal planes is equivalently observed for FS sheets (5,6) and (7,8). In both cases the spectra are easy to identify and the comparison with theory unambiguous and straightforward.

For lack of space, the details of sheet 5 and 6 are presented in the main text; the equivalent observations and arguments for sheets 7 and 8 are presented in the extended display items and supplement.

For both, sheets (5,6) and (7,8), the difference between cyclotron orbits with and without a crossing enforced by the intersection with the topological nodal planes can be resolved very well and unambiguously. Namely, the loop orbit shared between sheets 5 and 6 (and likewise between sheets 7 and 8) has been identified beyond any reasonable doubt. This orbit *must* pierce the Brillouin zone surface multiple times, because it is centered around U. The allowed trajectories are defined solely by the existence of the TPs.

In the absence of topological nodal planes magnetic break-down would be present. In this case, the DFT band structure implies an estimated FS splitting of ~ 5 meV, already resulting in a reduction of the weight of putative non-topological orbits by 50%. Such a strong effect could easily be resolved in our experiment. In fact, this would correspond to a k-space distance of about 1/500 of a reciprocal lattice vector, where the high resolution reflects the exponential dependence of the magnetic breakdown probability on the FS splitting. Thus, the experimental dHvA spectra directly prove the existence of the TPs with an unrivalled resolution.

As a comment on the side, the degree of information resolved here is much more direct than anything accessible by ARPES (provided ARPES could be used under applied magnetic field). State of the art ARPES could neither resolve the band crossing with a similar energy or momentum resolution, nor could it resolve whether the crossing is exactly located at the Fermi level. The latter is inherent in our study, since we probe directly the states on the Fermi surface. Moreover, measurements of Fermi arcs using ARPES provide also indirect evidence, since the crossings themselves are not probed. As a further comment, to demonstrate the presence of the topological nodal planes, it would be nice to track the Berry curvature, however, it is not essential as such.

We have revised the manuscript to communicate these points better.

Referee #2: The most direct way to prove such complicated nodal band crossings is band dispersion measurements by ARPES. I understand ARPES measurements under magnetic fields are impossible at present.

We reply: We agree that ARPES in recent years has been very widely established and highly successful in many studies. However, as summarized above ARPES is also subject to a large number of constraints. On this note, the observation of Fermi arcs represents inherently indirect evidence of topological band crossings, since the crossings are not detected directly!

Perhaps most importantly, dHvA measurements allow to resolve a tiny band splitting on the order of 5meV (and below) or 1/500 of a reciprocal lattice vector, thus permitting to discriminate between magnetic break-down and topologically enforced crossing points with meaningful accuracy. This resolution exceeds by far the state of the art achievable by ARPES. We therefore disagree strongly with the referee, that ARPES with present-day state-of-the-art resolution would be the better method, even if measurements could be performed under applied magnetic fields. For the specific problem we address in our study, quantum oscillations are superior as a band-specific, high-resolution probe of the quasiparticle spectra.

Referee #2: Given the nodal planes at the BZ boundaries carry large Berry curvature, it may lead to exotic transport properties such as large intrinsic anomalous Hall effect. Even though this is also not a direct probe for the nodal structure, the consequence of the topological nodal planes can be manifested by such measurements.

We reply: We wish to thank the referee for this suggestion. Several papers have reported magnetotransport properties of MnSi, revealing in particular a sizeable anomalous Hall effect in the spin-polarized state [e.g. M. Lee et al., Phys. Rev. B **75**, 172403 (2007)].

It transpires that some of us (A.B. and C.P.) were also involved in a study connecting the Hall effect of MnSi and $\text{Mn}_{1-x}\text{Fe}_x\text{Si}$ with brute force ab initio calculations [C. Franz et al., Phys. Rev. Lett. **112**, 186601 (2014)]. We apologize sincerely for not having cited this paper. As a main result this study established quantitative consistency of the electronic structure with normal, anomalous and topological Hall contributions in MnSi and the Fe-doped compounds. The Fermi surface calculated as part of this study is in excellent agreement with the Fermi surface determined experimentally in our dHvA measurements. The account of the anomalous Hall effect establishes strong Berry curvature contributions. While there is no doubt on the importance of large Berry curvatures, we agree with the referee, that this provides empirical evidence of topological nodal planes, but does not prove their existence as such.

To go beyond the work reported in this paper requires a numerical assessment of the transverse and longitudinal magnetoresistance, resolving the effects of intra- and interband transitions, as well as defect-related scattering [see also the discussion in section V(B) in R. Ritz et al. PRB **87**, 134424 (2013)]. This represents a formidable, independent research program well-beyond the work reported in our manuscript.

We have added a comment in the revised manuscript pointing out the empirical evidence for strong Berry curvatures associated with topological nodal planes, as observed in the large anomalous Hall effect.

Referee #2: In general, quasi-particles excited near the Dirac/Weyl points can be described as emergent relativistic fermions, which are characterized by small/zero effective mass, high mobility and Berry phase in cyclotron motions. Given the author showed MnSi features Weyl nodes, four-fold points and nodal planes, relativistic fermion behavior would be expected in this material. Nevertheless, I noted the effective masses m^* extracted from the temperature dependences of dHvA oscillation amplitudes {Fig. 2c} do not seem consistent with this expectation. m^* is $\sim 15m_e$ for FS sheets (5,6) and (7,8), far larger than that of free electron mass. Other FS sheets also do not show massless behavior though their masses are not as large as $15m_e$.

We reply: The referee states that the combination of a linear dispersion near Dirac and Weyl points with Berry phases in cyclotron motion as the main new aspect of topological crossing points. She/he claims that relativistic behavior, i.e., vanishingly small masses, would be expected in the dHvA spectra of MnSi. The remark of the referee suggests that she/he does not appreciate a fundamental aspect of our study and an important facet of the implications of our findings.

The experimental part of our study serves to demonstrate the existence of topological nodal planes in MnSi. This does neither require the observation of cyclotron orbits that feature a relativistic mass on the whole trajectory, nor does it require that orbits are fully located within these topological nodal planes. Rather, it requires to clearly resolve differences in the spectra with and without topological nodal planes. In fact, it is fundamentally not possible to obtain cyclotron orbits that are fully located within a topological plane in the present setting as time reversal symmetry is broken in the direction of the magnetic field, i.e., the orbits are always perpendicular to the nodal planes.

As the referee summarized correctly above, the topological nodal planes are located at the BZ boundaries perpendicular to the direction of the magnetization, whereas the field direction by force of nature is always normal to the cyclotron orbits in k-space.

Indeed, none of the orbits that can be observed runs *on* a plane of linear band crossings as shown in Fig. 4. The loop orbits of sheet 5 and 6 cut *through* the nodal planes on either two or four points on the orbit subject to the field direction. Thus, there are only a few nodal points on such a trajectory. Likewise, the orbits on sheet 7 and 8 demonstrate the presence of nodal planes perpendicular to the magnetization and the absence of the nodal planes when time reversal symmetry is broken. This is the reason why the corresponding orbits as a whole do not acquire a relativistic fermion cyclotron mass. The mass measured is loosely speaking an average around the orbit. More precisely, it scales with the integral over the reciprocal absolute group velocity along the closed trajectory. As a remark on the side, we note that we presented a comparison of the effective masses observed experimentally with the Sommerfeld coefficient of the specific heat to confirm that we detected all Fermi surface sheets.

While the cyclotron orbits in quantum oscillatory studies feature only points of the nodal planes, carefully conceived transport and bulk properties under magnetic field may be dominated by large sections of the Fermi surface supporting topological protectorates and thus relativistic masses. A simple example may be the observation of the large anomalous Hall effect as discussed above.

However, it may finally also be interesting to note that a combination of topologically charged and topologically trivial quasiparticle excitations is potentially very interesting in its own right. For instance, expanding a suggestion by Smith Phys. Rev. B **74**, 172403 (2006) they might ultimately be found to account for the extended non-Fermi liquid temperature dependence of the resistivity, observed in MnSi at high pressures. Another example concerns the possible formation of quantum disentangled liquids akin many-body localization in systems comprising quantum particles with very light and very heavy masses. Such exotic phases have been proposed theoretically by Grover and Fisher in J. Stat. Mech. P10010 (2014).

We have amended the presentation of our results and their putative impact in the revised manuscript.

Referee #2: In summary, the results reported in this manuscript is interesting and it may guide the search for new magnetic topological materials. The weakness is that the authors chose a material with an extremely complicated band structure to demonstrate the concept of symmetry-enforced nodal planes through dHvA measurements, such that the interpretation of the data is not quite convincing. I think this manuscript can be considered further if the concerns I summarized above can be addressed.

We reply: We hope that we managed to convince the second referee that her/his initial impressions of our manuscript were due to a lack of accuracy of our presentation and that all questions and comments can be answered satisfactorily. In turn, we hope that the second referee finds the revised manuscript much improved and suitable for communication to the wide audience of Nature.

Referee #3 (Remarks to the Author):

Marc Wilde and co-workers report on the observation of Magnetic-field-controlled topological protectorates of the Fermi surface in the ferromagnetic material MnSi. They employ a combination of theoretical methods, density functional theory and band topology calculations, together with de Haas-van Alphen (dHvA) quantum oscillation measurements to determine the complex Fermi surface of MnSi. It is a well-written manuscript, I congratulate the authors for this thorough piece of work.

The authors state that three key challenges in terms of material research concerning spintronic devices and quantum information technology are still unresolved:

- (a) Identify topological band degeneracies that are generically located at the Fermi level
- (b) the ability to easily control topological degeneracies
- (c) demonstrate the relevance of topological degeneracies in large, multi-sheeted Fermi surfaces

The theory section is very elaborate and highlights the importance of this work also for an experimental condensed matter physicist. The authors, however, need to be more specific what type of novel physics emerges from this paper and the specific study of MnSi that the manuscript can be considered in Nature.

We reply: We wish to thank the third referee for her/his efforts made to review our manuscript. We were delighted to read, that the third referee congratulates us on the thoroughness of our study. The detailed and very insightful questions and comments of the third referee demonstrate a deep appreciation of quantum oscillatory phenomena as well as the more subtle aspects of the results we present. They help us greatly to improve our presentation.

On this note we were also delighted to read, that the third referee did not express any concerns regarding the completeness of our data akin those articulated by the first and the second referee, who seem to be less familiar with quantum oscillatory studies. We feel that this underscores that we did not communicate and emphasize clearly enough our data for the first and the second referee to appreciate that our data answers all of the questions they raised.

In her/his conclusion the third referee encourages us to better communicate what type of novel physics emerges from the results of our study to better meet the requirements for publication in Nature. We wish to thank the referee for making us aware that we need to improve our manuscript in this respect.

In reply to the concluding recommendation of the third referee we would like to stress that the notion of topological crossings in the electronic structure of materials has generated a tremendous interest in and great hopes for applications. Yet, we trust that the third referee will agree with us that the required degree of control of such topological crossing has not been achieved beyond theoretical proposals and proof-of-concept demonstrations. We attribute the present state of the art precisely to the three key questions summarized by the referee, which motivated our study to start with.

Considered from a historical perspective, it was initially essential to identify topological crossing points that dominate the Fermi surface to convince the scientific community of their

existence and putative importance. However, it is not necessary for topological crossings to dominate the entire Fermi surface to be relevant for technological applications.

Rather, it is essential to be able to control the presence of such topological crossings by means of a simple and easy-to-use quantity. Choosing an appropriate probe that couples to the existence/non-existence of topological crossings will then enable the technological use of topological crossing. For instance, such a probe could be a carefully designed and micro-structured conduction channel placed on top of the material, or defects that couple microwaves or optical radiation specifically to the topological characteristic of the electronic structure. Here it is easy to imagine a cornucopia of different implementations. However, as the third referee will hopefully agree, this represents a full-blown research program in its own right, well beyond the scope of our study.

The main new physics we report may be summarized as the identification and experimental demonstration of a symmetry-enforced mechanism, that controls the formation of topological protectorates that are generically located at the Fermi level. In other words, the topological protectorates are insensitive to the details of the material and the band-filling, and may be controlled with the breaking of a specific symmetry. As the latter is purely a matter of field orientation in our case, a vanishingly small change of an external parameter that is easy to change allows to switch the topological protectorates on- and off, even though they are insensitive to details of the system affecting the band-filling.

All of this translates into a major effect of an unconventional property by means of a tiny change of parameter! Because the presence/absence of the topological crossing is enforced by the symmetry alone, all Fermi surface sheets intersecting the topological nodal planes are affected regardless how complex the Fermi surface may be. Thus, there is no longer a need to select materials with highly specialized Fermi surfaces (such as in TIs or Weyl or Dirac metals) to match the precise position of the Fermi level. Taken together, the generic positioning at the Fermi level, and the symmetry-enforced control of the topological protectorates provide enormous sensitivity, regardless of the complexity of the system. This represents the 'shift of paradigm' we report.

Referee #3: Point (a) and (b) and be identified and controlled in materials, where adjacent electron and hole pockets give rise to, for instance, magnetic breakdown phenomena. In other words, the field-tunability is inherent to the specific Fermi surface of the material under study.

We reply: We greatly appreciate this comment and question by the third referee, which alludes precisely to the main new aspect of the physics we report in our paper. The question also illustrates that we somehow did not manage to communicate this new aspect succinctly.

We agree entirely with the third referee, that Fermi surfaces in magnetic materials respond sensitively to an applied magnetic field. In fact, as the third referee probably knows there is a long history of magnetic-field controlled transitions of the Fermi surface topology, known as 'electronic topological transitions' (ETT) [e.g. Ya. M. Blanter et al. *Physics Reports* **245**,159 (1994)]. Such transitions are *not* concerned with the topological charge of Fermi surface sheets, but the topology of the Fermi surface as such. ETTs may include changes of topological charge. However, if they do, it is fortuitous at least on the level at which these studies are reported so far.

The magnetic breakdown phenomena the third referee mentions are somewhat similar. They might accidentally involve changes of topological charge, but not generically. Further, whether an ETT, magnetic breakdown, or specific transition of Fermi surface topology occurs under an applied magnetic field depends also on the field strength, representing a material-specific aspect just like the question whether the Fermi levels happens to reside precisely at the topological crossing, e.g., such as the Weyl points in a Weyl semimetal.

Indeed, magnetic break-down phenomena have been studied in great detail for many decades. In recent years exciting new insights have been gained theoretically and experimentally [e.g., O'Brien et al., PRL **116**, 236401 (2016); Alexandradinata and Glazman, PRL **119**, 256601 (2017), and van Delft et al., PRL **121**, 256602 (2019)]. Considering Weyl semimetals as a materials platform in which certain orbits are in close vicinity, they permit to explore exotic breakdown phenomena such as Klein tunneling. These effects differ, however, distinctly from the behavior we observe on FS sheets (5,6) and (7,8), where magnetic breakdown is absent due to the non-trivial topological properties of these FS sheets.

Taken together, in our study we identify a symmetry-enforced mechanism, namely non-symmorphic symmetries plus time-reversal symmetry, that are abundant in magnetic point groups. We show that they can be used to change the topological charge of Fermi surface sheets crossing topological nodal planes. Since all bands crossing the nodal plane are affected, the control is purely a matter of symmetry-breaking rather than absolute field strength or electron-filling (it is a generic on/off mechanism). In this sense it does not matter, if the material of interest has a hideously complex Fermi surface!

In the revised manuscript we have adapted the introduction, conclusions and the presentation of specific aspects of FS sheets (5,6) and (7,8) to better communicate the difference with classical field-tuned changes of the Fermi surface topology.

Referee #3: I have a few questions concerning the experiments. The authors perform dHvA experiments to determine the Fermi surface and the cyclotron masses at the Fermi energy. The spectrum in Figure 2b looks quite impressive and one does need some imagination to believe that all the observed frequencies revealed in Figure 2a originate from the raw data. I strongly encourage the authors to consider the following comments and questions.

We reply: We wish to thank the referee for her/his suggestions how to improve the presentation of our data. We have added clarifying information in the methods section, in Fig.2 and in section S4 in the supplementary information in the old version of the manuscript (in the revised manuscript the relevant section is S5). In addition, we inserted a new section S4 in the supplementary information to clarify the situation. In these amendments we present additional information concerning the points raised by the third referee in the following.

Referee #3: .

1) Technical questions: what is/are the window function(s) used for the FFT analysis? I do see quite some side lobes in the FFT spectra. Do the FFT spectra remain robust using different window functions?

We reply: We greatly appreciate this very insightful comment of the third referee. In our data analysis we applied a wide range of complementary methods and tests to delineate quantum oscillatory components of the torque signal from spurious artefacts due to the specific implementation of the Fast Fourier transforms (FFTs) and background subtraction. Notably, in the determination of the FFTs this included the use of different window functions and the use of different background subtraction schemes, as well as the calculation of FFTs from *synthesized* quantum oscillations using the Lifshitz-Kosevich (LK) formalism. In addition, we considered the angular evolution of dHvA branches to confirm that the dHvA frequencies were identified correctly.

Before turning to the technical facets of our analysis, we wish to note that the spectra shown in Fig. 2(b) [Fig. 2(c) in the revised manuscript], which triggered the question by the third referee, as well as the dHvA frequencies listed in table EII represent the essence of this very detailed technical analysis. In particular, Fig. 2(b) [Fig. 2(c) in the revised manuscript] was conceived to display in a single panel as many intrinsic spectral details over as wide a range as possible.

In our analysis several different FFT window functions were tested in order to distinguish side lobes from genuine dHvA orbits. The main representatives we used included Rectangular, Hann, Hamming, Blackman-Harris and Tukey windows.

The spectra shown in Fig. 2(b) [Fig. 2(c) in the revised manuscript] were determined with a rectangular window to display in a single panel as many intrinsic spectral details over as wide a range as possible. In other words, while a rectangular window exhibits strong spectral leakage, as the third referee points out correctly, it provides also the highest frequency resolution while losing no spectral power. For our study of MnSi this superior resolution is needed in order to resolve some of the dHvA frequencies, e.g., in regime III. In addition, FFTs were calculated with windows providing excellent side lobe suppression as a means to distinguish side lobes from real dHvA orbits.

For the convenience of all referees we present in Fig. R1 a comparison of the FFT of the 35mK data shown in Fig. 2(b) [new panel in revised manuscript] using a rectangular window and a Tukey window with $\alpha=0.5$ and $\alpha=1$ (the latter corresponds to a Hann window). All other parameters were kept identical. Moreover, no amplitude correction was performed for different windows in order to illustrate the loss of spectral power. For regime I to IV, the window extended from $1/14 \text{ T}^{-1}$ to $1/10 \text{ T}^{-1}$, where the large-amplitude low-frequency part of the oscillations and the ferromagnetic background were removed by subtracting a curve obtained by adjacent averaging with a moving interval of 10% of the full window. The data were then zero-padded, windowed and the FFT determined. For regime V, the window extended from $1/15 \text{ T}^{-1}$ to $1/4 \text{ T}^{-1}$. The field values given are the values before the internal field correction.

Fig. R1. Fast Fourier Transforms (FFTs) using three different Tukey windows corresponding to $\alpha=0$, $\alpha=1/2$ and $\alpha=1$. The general Tukey window corresponds to a rectangular window with $\alpha=0$, whereas the Hann window corresponds to $\alpha=1$.

Fig. R1 highlights that several distinct FFT peaks may be resolved for the rectangular window, e.g., between 2.5 kT and 3.2 kT in regime III, while they become increasingly smeared out for the other window functions. That is, only the rectangular window allows us to resolve these peaks. Vice versa, several tiny peak-like features on and near the slopes of large main peaks, e.g., between 2 kT and 2.5 kT below ξ_1 , are suppressed when using non-rectangular windows, revealing that these are spurious effects.

As an additional test to identify spurious effects due to the finite field range of our data we created FFT spectra with identical parameters using *synthesized* quantum oscillatory data generated by means of the LK formalism. That is, quantum oscillations were calculated using the LK formula with the measured frequencies and masses as input and the corresponding FFTs were compared with the FFTs of the experimental data. This method provides an additional impression of the side lobes, since it is known *a priori* which peaks correspond to actual dHvA orbits in the synthesized spectra.

We finally also used the angular evolution of the spectra to discriminate between dHvA orbits and spurious side lobes. For instance, close-by dHvA frequencies may display a very different angular evolution and even cross as a function of angle, while a side lobe tends to track the main peak as a function of angle.

Referee #3:

2) I advise the authors to show the 35 mK curve and graph where the high-frequency oscillations are better seen (by subtracting the background)

We reply: We thank the third referee for this suggestion which we took as an encouragement to give our experimental results additional credibility.

In the revised version of the manuscript we added a new panel to Fig. 2 [Fig.2(b) in the revised manuscript], showing torque versus reciprocal magnetic field at 35mK. To better bring out frequency components due to heavy masses, which display small oscillation amplitudes, we subtracted the background and the lowest frequencies featuring large amplitudes. For the subtraction we used a moving average with a window of ~ 0.0028 1/T. In order to improve the visibility of the individual oscillations, data in Fig. 2(b) are shown for a reduced range of reciprocal fields.

Referee #3:

3) The authors state that 'FFT's over the range 4 to 15 T (10 T to 15 T) were performed to evaluate frequency components below (above) $f = 350$ T.' Can the authors comment on the error bars of the carrier masses? Did the authors perform a more thorough range analysis to ensure that the carrier masses are not over and/or underestimated?

We reply: We wish to thank the third referee for enquiring about these very important aspects of our analysis. Indeed, we have considered all of these points and addressed them according to the state-of-the-art as explained in the following. We apologize for not having provided the full details in the first version of our manuscript. In the revised manuscript we added a new section S4 to the supplementary information, where we discuss these aspects. In addition, we added a remark in the methods section, referring the reader to section S4 for information on these questions.

Following convention, the effective masses were inferred from the temperature dependence of the FFT intensity using the LK-formalism (eq. (6) in the methods section), which is a function of T/B . The effective masses are hence subject to statistical uncertainties in the sample temperature and FFT intensity (e.g. due to numerical uncertainties), where we assume that uncertainties in the actual field values are vanishingly small. They are, further, subject to systematic variations due to the magnetic field range, the window of reciprocal fields analyzed, as well as the magnetic field dependence of the effective mass itself.

Starting with the error bars of the carrier masses reported in table EII, they represent the statistical uncertainty of the mass inferred from the Lifshitz-Kosevich fits of the temperature dependence observed experimentally with respect to the average magnetic field B_{av} of the FFT window. They are thus well-defined, reflecting uncertainties in the sample temperature and FFT amplitude.

We added a sentence in the caption of Table EII stating how the error bars were determined.

The systematic uncertainties are assumed to be dominated by FFT window boundaries as follows: The Lifshitz-Kosevich analysis yields accurate results, when the size of the FFT window in $1/B$ is small compared to $1/B_{av}$. It is thus important to assess the effect of the window size on the results. We tested our results for (i) different window sizes centered at the same $1/B_{av}$ and, (ii), smaller FFT windows centered at different $1/B_{av}$. This gives, in principle, an account of, both, the dependence on the window size and a possible B -dependence of the quasiparticle masses themselves.

For this type of analysis, we needed to narrow the FFT windows in comparison to the full-range rectangular window required to resolve most of the narrowly spaced frequency branches. In turn, a comprehensive analysis of the systematic dependence of the mass values on the window size was only possible for some branches. In particular, such an analysis was not possible for the low-frequency dHvA branches up to 350T which required the largest windows. Instead, the analysis was possible for FFT peaks that are well isolated on the frequency axis, such as the orbits κ and M . While narrowing the window did have an effect on the values of the mass of κ and M , it resulted also in much larger error bars. Within these larger error bars, however, there was no significant trend.

Keeping these limitations in mind, it has long been known that the quasiparticles in MnSi acquire a large part of their mass from coupling to the spectrum of spin fluctuations. This is

reflected in the magnetic field dependence of the Sommerfeld coefficient, which decreases by roughly 20% up to 14T. Such a reduction is known as 'quenching' of spin fluctuations (a mode stiffening under applied fields here without discernible spin-wave contributions) that is well-established in many d- and f-electron compounds, notably the class of heavy fermion materials. For the comparison of the effective masses, the Sommerfeld coefficient close to the average field value was used.

Referee #3: I agree and think that the authors did a great job in the assignment, see Table EII, of the orbits (comparison theory and experiment). Concerning magnetic breakdown and the different breakdown orbits, I wonder whether the authors could try to provide more experimental evidence concerning their assignment by the comparison of their cyclotron masses with the sum of the individual associated orbits for each breakdown orbit.

4) Can the authors provide an insight into the masses of the magnetic breakdown orbits by looking at the sum of the masses of the individual orbits?

We reply: Again, we would like to thank the third referee for this very insightful and clever remark, reflecting her/his very deep knowledge and appreciation of quantum oscillatory phenomena. Indeed, we have also carried out an analysis of the break-down orbits along the lines suggested by the third referee. This analysis was not mentioned in our manuscript for mere lack of volume. It served foremost as another sanity-check regarding the determination of the effective masses. In the revised manuscript we have now added a paragraph in Section S5 in the supplementary information on sheets 3 and 4, where we discuss the effective masses of the breakdown orbits. For the sake of convenience, we reproduce this paragraph in the following:

"We note that the five sets of breakdown branches defined above exhibit a systematic hierarchy of cyclotron masses that can be used to further corroborate the assignment. This is understood most easily by looking at Fig. E6 c2. When we neglect the SOC-induced avoided crossings at j_1 to j_8 for a moment, only two orbits would arise as shown in the inset. The dark red orbit, which originates from the cuboid sheet 3, and the dark green orbit, which originates from the octaeder-shaped sheet 4. These two have band masses of $1.5m_e$ and $3.2m_e$, respectively. Including SOC, avoided crossings at $j_1 - j_8$ lead to the five sets of breakdown branches.. Now, the highest branch of set 1 corresponds to the non-SOC sheet 3, while the lowest branch of set 5 corresponds to the non-SOC sheet 4. They have - of all branches - the lowest and the highest masses, respectively. The masses of the other branches are in between these two extremes, since the carriers travel partly on both non-SOC trajectories.

It follows, that the lowest branch in the first set has the highest mass in that set, since the carriers travel on the heavier sheet at all four corners, i.e., between $j_1 - j_2$ and $j_3 - j_4$ etc. The next higher branch of the same set is a bit lighter, since it incorporates only three of the corners etc. The effect is, however, small in the calculated band masses. The same argument applies to all five sets. Thus, the masses are expected to increase overall from the lowest frequency set 1 to the highest frequency set 5, but decrease with increasing frequency within each set."

This assignment matches the expected effective mass evolution of the branches discussed above: Overall the masses increase from set one to set five from $\sim 10m_e$ to $\sim 16m_e$. However, inside the sets, a decrease of masses with increasing frequency is observed within

experimental error, as can be seen for set one going from $10.9m_e$ (ρ) to $8.7 m_e$ (ν). The overall fading of the observed signal strength with increasing frequency is also mainly due to the mass evolution from set 1 to set 5.

We decided to discuss the effective masses of breakdown orbits in terms of the trajectories without SOC, since this appears to be most accessible for an interested general readership. However, since the third referee asked specifically about relating the magnetic breakdown orbit masses to the *sum* of the individual orbits we wish to point out the equivalence of the reasoning above to the general approach of attributing the orbits enclosing any of the areas defined in Figure E6 c2 with their cyclotron mass $m^*=(\hbar^2/2\pi)*d|A_{in+\Delta_1+\dots}|/dE$. Using the above relation on each of the orbits considered with their respective areas, one arrives at the masses as discussed above.

A standard procedure often discussed in the literature, e.g., in the context of simple networks of overlapping orbits with small Bragg gaps at the zone boundaries, is to derive the mass of a combination of orbits by summing up the masses of all orbits that contribute to the combination of orbits. In the specific situation we encounter in our study, orbits enclosing only a Δ_1 (or a Δ_2) are semi-classically forbidden: Both sheets are hole sheets and both are centered around Γ and anticross inside the bulk of the BZ, i.e., charge carriers travelling on them have the same sense of rotation. This leads to parts of a Δ_1 -only-trajectory which would have to be followed in a direction opposing the Lorentz force. Hence, such orbits are expected to be absent from the dHvA spectrum and were, accordingly, also not observed in our study. We would thus need to introduce them artificially and assign appropriate masses to them for a discussion in terms of a „sum of masses of individual orbits“, which would be counterintuitive to start with.

Referee #3:

5) There is a typo on page 58: Lidshitz-Kosevich

We reply: We have corrected the typo; We thank the referee for pointing this out.

Referee #3: To conclude, I do not see enough new physics in this manuscript to recommend immediate publication in Nature. This manuscript, taking into account my comments and suggestions, may be suitable for another journal within the nature family ([REDACTED] or [REDACTED]).

We reply: We greatly appreciate the encouragement of the third referee to better communicate the main new physics of our results and the large number of very insightful questions on technical aspects of our study. We hope that the third referee finds the revised manuscript much improved and recommends publication in Nature.

Reviewer Reports on the First Revision:

Ref #1

Some of my concerns are addressed by the authors in the modified manuscript. The authors clarified that the de Haas – van Alphen effect is a feasible experimental method to study the symmetry-enforced nodal planes. Their experimental results strongly imply the existence of the symmetry-enforced nodal planes in MnSi. However, since there are so many complex trivial Fermi Surfaces in MnSi, I feel it is not a suitable material to reveal the physics of nodal plane. More important, the authors do not show any other physical properties induced by the topological nodal planes. Therefore, I cannot recommend its publication in Nature.

Ref #2

The revised manuscript has been significantly improved and the authors addressed most of the issues raised in my first report satisfactorily. I now find that the comparison of theoretical and experimental dHvA spectra for Fermi surface sheets (5,6) and (7,8) does give compelling evidence for the argument of symmetry-enforced nodal planes and topological protectorates (TPs) at the Fermi surface. The authors have also shown a number of magnetic materials have the necessary symmetries that enforce nodal planes and TPs. Since TPs are at the Fermi level and sensitive to the direction of magnetization, it may enable tuning of exotic properties by the control of band topology. I think this work has broad interest and can possibly generate a great deal of interest. I recommend publishing this manuscript in Nature.

However, I think the current version of the manuscript is still difficult to follow for general readers for the following reasons:

- 1) In Fig. 1f, the authors present the entire Fermi surface which includes 10 different sheets. While sheets (1,2) and (3,4) can be seen clearly, it is very difficult to distinguish sheets (5,6) from sheets (7,8) due to their similarity.
- 2) Fig. 4a is extremely confusing; sheet 5 and 6 cannot be resolved by color at all. Without clear visualization of sheets 5 and 6, it is difficult to see how the loop orbits around U (Fig. 4b1 and 4b2) is comprised of different segments of sheets 5 and 6. I noted sheets 5 and 6 as well as the loop orbits around U shown in Figure E6 d1 are much clear. I suggest the authors to replace the current Fig.4a with Figure E6 d1.
- 3) In Fig. 4e, the authors present the intensity map of dHvA spectra in the regime of the neck-type orbits around the Y1 point. It would be more convincing to readers if a similar dHvA map is given for the neck-type orbit around the X-point, for which only a single frequency is expected. I think these problems should be fixed before acceptance.

Ref #3

The authors have made a considerable effort to improve their work. The overall broad impact of this manuscript is much better highlighted and justified now to my opinion (also described in detail in their response in part A). The answers to the questions/comments of the referees are very detailed and clear.

I can be very brief here and recommend publication of the revised manuscript in Nature.

Author Rebuttals to First Revision:

Referee #1: Some of my concerns are addressed by the authors in the modified manuscript. The authors clarified that the de Haas – van Alphen effect is a feasible experimental method to study the symmetry-enforced nodal planes. Their experimental results strongly imply the existence of the symmetry-enforced nodal planes in MnSi. However, since there are so many complex trivial Fermi Surfaces in MnSi, I feel it is not a suitable material to reveal the physics of nodal plane. More important, the authors do not show any other physical properties induced by the topological nodal planes. Therefore, I cannot recommend its publication in Nature.

We reply: We wish to thank the first referee for acknowledging the evidence we present of topological nodal planes in MnSi. We regret to learn that the first referee expects in addition evidence for unusual physical properties due to the topological nodal planes. We feel that this goes well beyond the scope of our study and regret that the first referee does not recommend publication in Nature.

Referee #2: The revised manuscript has been significantly improved and the authors addressed most of the issues raised in my first report satisfactorily. I now find that the comparison of theoretical and experimental dHvA spectra for Fermi surface sheets (5,6) and (7,8) does give compelling evidence for the argument of symmetry-enforced nodal planes and topological protectorates (TPs) at the Fermi surface. The authors have also shown a number of magnetic materials have the necessary symmetries that enforce nodal planes and TPs. Since TPs are at the Fermi level and sensitive to the direction of magnetization, it may enable tuning of exotic properties by the control of band topology. I think this work has broad interest and can possibly generate a great deal of interest. I recommend publishing this manuscript in Nature.

However, I think the current version of the manuscript is still difficult to follow for general readers for the following reasons:

1) In Fig. 1f, the authors present the entire Fermi surface which includes 10 different sheets. While sheets (1,2) and (3,4) can be seen clearly, it is very difficult to distinguish sheets (5,6) from sheets (7,8) due to their similarity.

2) Fig. 4a is extremely confusing; sheet 5 and 6 cannot be resolved by color at all. Without clear visualization of sheets 5 and 6, it is difficult to see how the loop orbits around U (Fig. 4b1 and 4b2) is comprised of different segments of sheets 5 and 6. I noted sheets 5 and 6 as well as the loop orbits around U shown in Figure E6 d1 are much clear. I suggest the authors to replace the current Fig.4a with Figure E6 d1.

3) In Fig. 4e, the authors present the intensity map of dHvA spectra in the regime of the neck-type orbits around the Y1 point. It would be more convincing to readers if a similar dHvA map is given for the neck-type orbit around the X-point, for which only a single frequency is expected.

I think these problems should be fixed before acceptance.

We reply: We were delighted to learn that second referee welcomes the main messages of our study and appreciates the revised version of our manuscript. We wish to thank the second referee for recommending publication of the manuscript.

We wish to thank the second referee for her/his recommendation on Fig. 4a. Following careful consideration of the suggestion by the second referee we favored to stay with the present version of Fig. 4a and refer the reader in the figure caption to the alternative color shading shown in the Extended Data 6d1.

Referee #3: The authors have made a considerable effort to improve their work. The overall broad impact of this manuscript is much better highlighted and justified now to my opinion (also described in detail in their response in part A). The answers to the questions/comments of the referees are very detailed and clear.

I can be very brief here and recommend publication of the revised manuscript in Nature.

We reply: We were very pleased to learn that the third referee welcomes the revised manuscript as much improved, recommending its publication.